# Alpha-ketoglutarate mitigates insulin resistance and metabolic inflexibility in a mouse model of Ataxia-Telangiectasia

Jacquelyne Ka-Li Sun[1], Ronald P. Hart [2], Karl Herrup [3], Amy Zexuan Peng[1], Genper Chi-Ngai Wong[1], Deng Wu[1], Kin-Ming Kwan[1,4,5] & Kim Hei-Man Chow [1,6] ✉

The maintenance of metabolic homeostasis relies on the ability to flexibly transit between catabolic and anabolic states in response to insulin signaling. Here we show insulin-activated ATM is a critical mediator of this process, facilitating the swift transition between catabolic-and-anabolic fates of glucose by regulating the functional status of PKM2 and HIF1α. In Ataxia-Telangiectasia (A-T), these mechanisms are disrupted, resulting in intrinsic insulin resistance and glucose intolerance. Consequently, cells exhibit a compensatory dependence on glutamine as an alternative metabolite for energy metabolism. Cerebellar degeneration, a hallmark of A-T, is characterized by the pronounced vulnerability of Purkinje cells, attributed to their unexpected sensitivity to insulin. Supplementation with α-ketoglutarate, the α-keto acid backbone of glutamine, has demonstrated potentials in alleviating glutamine dependence and attenuating Purkinje cell degeneration. These findings suggest that peripheral metabolic deficiencies may contribute to sustained neurodegenerative changes in A-T, underscoring the importance of screening, monitoring and addressing these metabolic disruptions in patients.

Ataxia-telangiectasia (A-T) is a rare autosomal recessive disorder caused by loss-of-function mutations in the ataxia-telangiectasia mutated (*ATM*) gene. While cerebellar ataxia is the hallmark clinical feature of A-T, a significant proportion of patients also exhibit a broad spectrum of chronic metabolic syndromes[1,2]. This included growth retardation, muscular atrophy and unexpected weight loss[3]. Among aging A-T patients, additional metabolic comorbidities, including dyslipidemia, glucose intolerance, nonalcoholic fatty liver diseases and diabetes, have also been reported[4–9]. Notably, a family case study has shown that insulin resistance is associated with the worsening of cerebellar and motor function[7]. More broadly, cerebellar ataxia is also a common clinical feature observed in over 150 inherited metabolic disorders[10].

Building upon the growing understanding of the interaction between the brain and the periphery, a fundamental question that remains unresolved is whether metabolic disruptions originating from the periphery can exert lasting effects on brain function via altering the profiles of endocrine signals derived from the peripheral system[11]. The metabolic manifestations observed in individuals with A-T suggest a plausible role for endocrine dysregulation in these processes[12]. Among the endocrine signals frequently disrupted in A-T, insulin stands out, with numerous reports documenting diminished insulin sensitivity or insulin resistance in A-T patients[7,8,13]. Under normal physiological conditions, insulin serves as a key anabolic hormone, promoting the biosynthesis and storage of macronutrients such as glycogen, protein,

[1]School of Life Sciences, Faculty of Science, The Chinese University of Hong Kong, Hong Kong, Hong Kong. [2]Department of Cell Biology and Neuroscience, Rutgers University, Piscataway, NJ, USA. [3]Department of Neurobiology, School of Medicine, University of Pittsburgh, Pittsburgh, PA, USA. [4]State Key Laboratory of Agrobiotechnology, The Chinese University of Hong Kong, Hong Kong, Hong Kong. [5]Centre of Cell and Developmental Biology, The Chinese University of Hong Kong, Hong Kong, Hong Kong. [6]Gerald Choa Neuroscience Institute, The Chinese University of Hong Kong, Hong Kong, Hong Kong. ✉e-mail: heimanchow@cuhk.edu.hk

and lipids. Simultaneously, insulin inhibits the breakdown of these macromolecules, thereby preventing any futile metabolic cycles[14]. Previous studies have identified the ATM kinase as a downstream mediator of insulin signaling, facilitating insulin-stimulated glucose metabolism[15–17]. However, several critical questions remain unanswered: the precise mechanism by which insulin activates ATM, whether ATM plays a role in coordinating the dynamic transitions between catabolic and anabolic states in insulin-sensitive cells, and why these metabolic disturbances primarily contribute to ataxic symptoms in A-T, but not cognitive and psychiatric conditions that are also commonly related to cerebellar damage[18,19].

Here, we show that deficiencies in peripheral metabolism play a significant role in the development of sustained cerebellar changes in individuals with A-T. Specifically, our findings suggest that the activation of ATM by insulin triggers a unique phosphoproteome network that is crucial for transitioning the glucose metabolic fate from catabolic to anabolic. When ATM is lacking, extensive metabolic reprogramming occurs, leading to a compensatory reliance on amino acids, especially glutamine, as an alternative fuel source for mitochondria. These alterations make insulin-sensitive Purkinje cells, particularly those expressing Zebrin-II/ALDOC which are anatomically abundant in the posterior and flocculonodular lobes of the vermal cerebellum, susceptible to the metabolic changes associated with ATM deficiency. Such an anatomical-specific vulnerability thereby contributes to the development of ataxic and motor-related symptoms in A-T. Our study demonstrates that supplementing with α-ketoglutarate, the α-keto acid of glutamine, can serve as an alternative metabolite to mitigate systemic glutamine dependence, glucose intolerance and simultaneously address concerns related to ammonia neurotoxicity.

## Results

### ATM deficiency leads to systemic insulin resistance and metabolic inflexibility

To comprehend the impact of ATM deficiency on whole-body metabolic status, a comprehensive metabolic analysis was conducted in *Atm*-knockout (KO) mice at ages comparable to adolescence (3 months old) and early adulthood (6 months old) in humans[20]. While discernible differences in physical parameters were absent among the younger mice, evident stunted growth and reduced body weight were observed in the 6-month-old *Atm*-KOs (Fig. 1a, b, Supplementary Fig. 1a). Post-weaning mice are typically fed on carbohydrate-rich diets, akin to common human diets, indicating that carbohydrate is a primary source of carbon for biomass synthesis and growth[21]. The efficiency of assimilating digested and absorbed carbohydrates, i.e., glucose, was evaluated by Kraft test, which simultaneously evaluates glucose tolerance (intravenous glucose tolerance test, IGTT) and insulin response[22]. While the 3-month-old mice exhibited nearly normal glucose tolerance, elevated plasma insulin profiles however, suggested pre-diabetic insulin resistance (Fig. 1c, d). In contrast, majority of 6-month-old *Atm*-KO mice were glucose intolerant complicated with compromised insulin responses, hinting a progression towards type 2 diabetes (Fig. 1c, d). The HOMA-IR (Homeostatic Model Assessment for Insulin Resistance) index calculation further confirmed that *Atm*-KO mice, irrespective of age, were insulin resistant (HOMA-IR > 2.2)[23] (Fig. 1d). These observations therefore hinted an incompetence of carbohydrate utilization. Further elaboration by the indirect calorimetry test revealed that while the average food intake and locomotor activities remained similar among all the groups, persistent, non-fluctuating respiratory quotient (RQ) values ranging between 0.8 and 0.9 (indicative of protein catabolism) in both light and dark cycles were found among the 6-month-old *Atm*-KO mice (Fig. 1e). This contrasted with the profound ability to switch from protein/lipid metabolism to carbohydrate catabolism (RQ = 0.9–1) during the dark cycles observed in other experimental groups (Fig. 1e). Such a lack of metabolic flexibility due to insulin resistance and diabetic complication

gradually depleted the adipose reserves (Fig. 1f) and skeletal muscle masses (Fig. 1g). Notably, abnormal accumulation of oil droplets in the liver (Fig. 1h) and an elevated fed-stage serum triglyceride (TG) levels (i.e., hyperlipidaemia) were found (Supplementary Fig. 1b). These observations suggested a gradual development of cachexic-like phenotype[24] among the diabetic 6-month-old *Atm*-KO mice.

Previous study revealed ATM mediates insulin action on enhancing glucose uptake and assimilation[16], this prompted us to investigate whether the metabolic characteristics observed in *Atm*-KO mice are contributed by defective insulin actions. An artificial model of insulin deficiency induced by streptozotocin (STZ) was employed in 1-month old young mice (Fig. 1i)[25], an age well precede any obvious metabolic disturbances in concern had emerged (Fig. 1a–h). Subsequently, these mice were administered daily equal dosages of either long-lasting insulin (glargine) or a control vehicle for the following two weeks (Fig. 1i). Indirect calorimetry results revealed that among the WT mice, glargine administration restored systemic carbohydrate utilization, primarily during the dark cycle when food was consumed (Fig. 1j, Supplementary Fig. 1c, d). This treatment also averted the onset of glucose intolerance, elevation of serum triglyceride levels, and accumulation of oil droplets in liver emerged in the WT mice in vehicle treatment (Fig. 1k-m, Supplementary Fig. 1e–g). In contrast, the same glargine administration program failed to improve these metabolic abnormalities in *Atm*-KO mice, indicating that ATM is crucial for insulin's systemic metabolic effect and the loss of ATM leads to these metabolic abnormalities.

### Cerebellar Purkinje cells are insulin sensitive and functionally associated with peripheral insulin resistance

We next investigated if metabolic and endocrine disruptions related to insulin-ATM signaling originating from the periphery may contribute to cerebellar degeneration and ataxia in A-T via the altered endocrine signals of insulin. To confirm the exceptional relevance of such signaling pathway in this unique brain region, we first examined the spatiotemporal transcriptomic profiles of various brain regions throughout the human lifespan. Gene expression levels of *ATM* in the cerebellar cortex consistently champed over other brain regions during both prenatal and postnatal periods (Supplementary Fig. 2a)[26]. Notably, a similar expression pattern was also found for insulin receptor (*INSR*) (Supplementary Fig. 2a). Subsequent immunohistochemical analyses with antibodies targeting insulin receptor isoforms (INSRα/β) and insulin receptor substrates (IRS1/2) in 6-month-old *Atm*-WT mice unveiled a notable presence of INSRβ and IRS1 signals within Purkinje cells (IP3R1 + ) (Supplementary Fig. 2b), hinting they are responsive to insulin originated from the peripheral circulation that gained entry into the central nervous system via an active transport mechanism[27,28]. In the in vivo settings, temporal dynamics of the insulin response following a surge in blood glucose introduced through IGTT was recorded (Fig. 2a). As typically expected, the duration of heightened plasma insulin levels lasted for approximately 2 h before returning to near the baseline. This therefore, also led to the detection of insulin in the cerebrospinal fluid (CSF). Notably, elevated basal and saturated CSF insulin concentrations were detected in all *Atm*-KO mice before and at 2 h after the initiation of IGTT (Fig. 2a) which mirrors a phenomenon found in other pre-diabetic models[29]. With reference to this physiological time duration of a systemic insulin response, 2-hour treatment of glargine to ex vivo cerebellar tissue cultures was conducted, which further validated the responsiveness to insulin in this brain region (Fig. 2b). Functionally, behavioral evaluations on the motor and coordination functions, alongside the corresponding HOMA-IR index, demonstrated inverse correlations (Fig. 2c). This prompted us to examine the cerebellar tissues harvested at 2 h post-IGTT from 6-month-old mice. Those from the *Atm*-KO group revealed signs of insulin resistance, characterized by diminished tyrosine autophosphorylation of INSRβ in conjunction with heightened serine

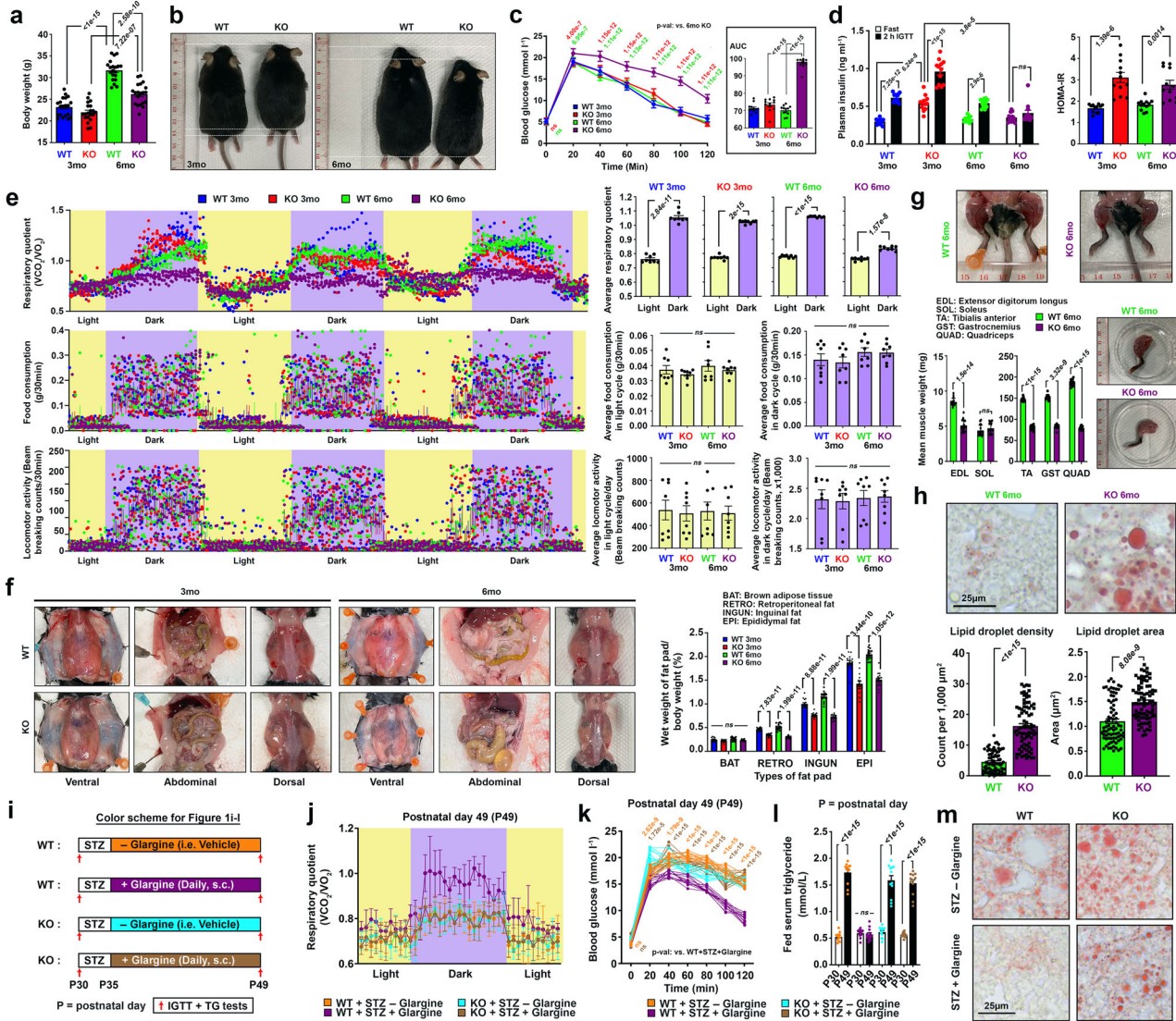

**Fig. 1 | ATM deficiency gives rise to a hypercatabolic phenotype. a** Summary on body weight differences ($n = 20$, one-way ANOVA). **b** Representative images of mice showing body length differences ($n = 18$), quantification presented in Supplementary Fig. 1a. **c** Left: results of IGTTs ($n = 12$, two-way ANOVA). Right: area under the curves (AUCs) ($n = 12$, one-way ANOVA). **d** Left: plasma insulin levels were assessed during fasting and 2 h following the IGTT ($n = 12$; two-way ANOVA). Right: HOMA-IR index data ($n = 12$, one-way ANOVA). **e** Left: Chronological variations in indirect calorimetry measurements depicting alterations in respiratory quotient, food consumption, and locomotor activity. Right: Changes observed between light-and-dark cycles ($n = 8$; two-tailed unpaired t-test for RQ; one-way ANOVA for the rest). **f** Left: visual representation of mouse truncal fat deposits. Right: Percentage wet weight of fat pads over total body weight ($n = 16$, one-way ANOVA). **g** Images of hindlimbs and wet weights for major skeletal muscles [$n = 16$, two tailed unpaired t-test for all except for GST where two-tailed Mann Whitney test was applied].

**h** Representative images of liver oil red O staining. Alterations in lipid droplet density and area are presented ($n = 80$ images, 16 mice per group, two-tailed Mann-Whitney test). **i** Schematic diagram detailing the daily treatment regimen of streptozotocin (STZ) (40 mg/kg body weight, i.p.) and glargine (50 Units/kg body weight, s.c.). **j** Temporal variations in the indirect calorimetry readings on alterations in respiratory quotient ($n = 6$). Quantification presented in Supplementary Fig. 1d. **k** Outcomes of IGTTs following the STZ treatment paradigm depicted in Fig. 1i ($n = 12$, two-way ANOVA). AUCs presented in Supplementary Fig. 1f. **l** Fed serum triglyceride levels assessed before and after the STZ treatment regimen ($n = 12$, two-way ANOVA). **m** Representative images of liver oil red O staining following the STZ treatment regimen. Quantification presented in Supplementary Fig. 1g. $N$ presents biological replicates. Values represent the mean ± s.d. Source data are provided as a Source Data file.

phosphorylation on IRS1 (Fig. 2d, Supplementary Fig. 2c). Previously, insulin was found to promptly activate ATM in cellular models[16], such response was also replicated in our ex vivo cerebellar tissues model (Fig. 2e), via an unexpected mechanism associated with oxidative stress and dimerization of ATM (Fig. 2e, Supplementary Fig. 2d, e). Such effect was effectively abolished by ethyl pyruvate, a stable sca-venger of hydrogen peroxide ($H_2O_2$) (Fig. 2f)[30]. While the tyrosyl phosphorylation cascade on the insulin receptor traditionally med-iates the insulin signaling, emerging evidence indicates that the

hormone can also initiate an alternative $H_2O_2$-dependent response via the actions of NAD(P)H oxidase (NOX)[31,32]. Notably expressed in the cerebellum (Supplementary Fig. 2f), NOX4 has been linked to A-T, although its association with insulin signaling was previously unestablished[33]. In the current investigation, the inhibition of NOX4 by its specific antagonist GLX7013114 or shRNA effectively impeded insulin-induced dimerization and activation of ATM (Fig. 2f, Supple-mentary Fig. 2g), confirming the involvement of an oxidative stress-related mechanism in this process.

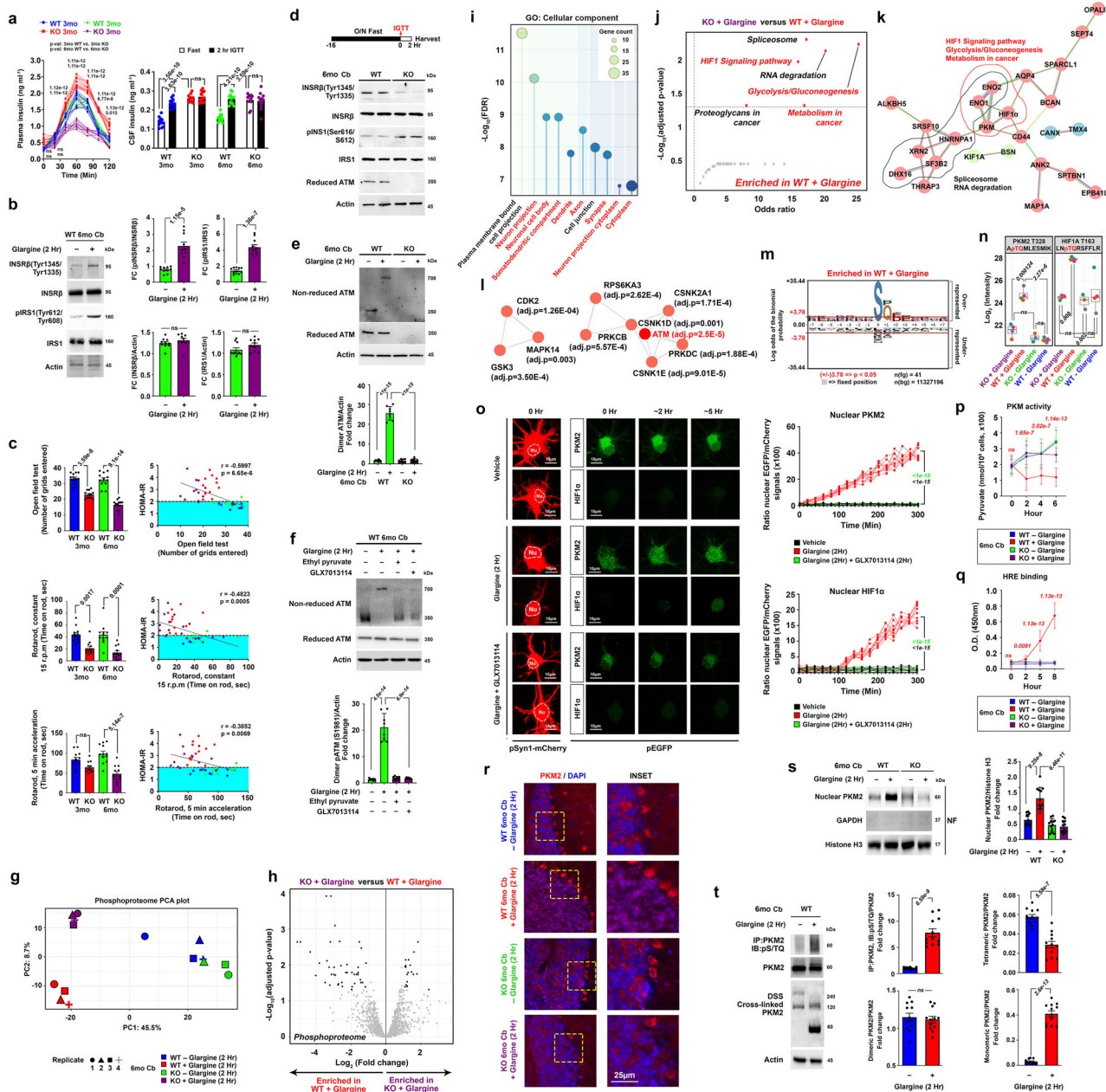

**Fig. 2 | Insulin-activated ATM modulates key regulators of aerobic glycolysis in the cerebellum. a** Plasma and cerebrospinal fluid insulin levels assessed during the IGTT ($n = 12$; two-way ANOVA). **b** Representative immunoblot images on key insulin signaling components. Quantification presented on the right ($n = 10$; two-tailed unpaired t-test). **c** Summary on behavioral performances in the open-field and rotarod paradigms ($n = 12$, one-way ANOVA for all except for Rotarod, constant 15 r.p.m test where Kruskal-Wallis test was used). Correlations between HOMA-IR index and performances ($n = 48$, Pearson's correlation). **d** Representative immunoblot images on key insulin signaling components ($n = 16$). Quantification presented in Supplementary Fig. 2c. **e, f** Representative immunoblots on ATM status ($n = 8$, one-way ANOVA). **g** PCA plot of phosphoproteome profiles ($n = 4$ for all except $n = 3$ for KO-Glargine group). **h** Statistical significance versus changes in magnitude of phosphopeptides ( | log2 fold change | >1) (Limma with Benjamini-Hochberg correction) is shown. **i** Cell component analysis and **j.** functional pathway enrichment of phosphopeptides enriched in "WT+Glargine" group referencing the STRING database (Benjamini–Hochberg correction) and KEGG database on the Enrichr[172] (Fisher exact test with correction), respectively. **k** Protein-protein interaction network analysis and **l** kinase enrichment analysis of phosphopeptides

enriched in "WT + Glargine" group referencing the STRING database (Benjamini–Hochberg correction) and kinase enrichment database on the Enrichr[13] (Fisher exact test with correction), respectively. **m** Visualization of phosphorylation motif of phosphopeptides detected in "WT + Glargine" group via the pLogo platform (Binominal probability with Bonferroni correction)[152]. **n** Relative intensities of pPKM2(T328) and pHIF1A(T163) phosphopeptides ($n = 4$ for all except "KO-Glargine" where $n = 3$, one-way ANOVA). **o** Representative time-lapse images of GFP-tagged PKM2 and HIF1α transient dynamics in cortical neurons pre-transfected with AAV-Synapsin-mCherry-C1-WPRE and pEGFP-C1-PKM2/pg-HIF-1alpha-EGFP. Quantifications presented on the right ($n = 10$, two-way ANOVA).

**p, q** Measurements of (**p**). PKM activity and (**q**). degree of HRE binding ($n = 16$, two-way ANOVA). **r** Representative images of PKM2 ($n = 8$). **s, t** Representative immunoblots on **s**, nuclear localization ($n = 12$, one-way ANOVA) and **t** structural status ($n = 12$, two-tailed unpaired t-test) of PKM2. Unless otherwise specified, all ex vivo glargine and TEPP-46 treatments were performed in 100 nM for 2 h. Values represent the mean ± s.d. $N$ presents biological replicates. Source data are provided as a Source Data file.

## Insulin-activated ATM modulates key regulators of aerobic glycolysis in the cerebellum

ATM, a serine/threonine kinase conventionally associated with nuclear DNA damage responses, showcases more than 700 phosphorylation targets in response to DNA damage stressors[34]. In contrast, when serving as a cytosolic stress sensor, ATM engages a distinct array of substrates[35]. To explore the cellular consequences downstream of insulin-activated ATM, quantitative label-free phosphoproteomics analysis was performed in cerebellar tissues collected from both *Atm*-WT and *Atm*-KO mice, which identified a total of 2320 phosphosites that could be reproducibly quantified (Supplementary Dataset 1). Among which, 983 (i.e., 42.3%) phosphosites differ significantly among all the experimental groups (Supplementary Fig. 3a–f, Supplementary Dataset 1). This change contrasted with the total proteomic analysis conducted in parallel, which only 7.5% (i.e., 236/3150) of the total proteins detected were different significantly (Supplementary Fig. 4, Supplementary Dataset 2). Such huge difference indicated that most changes detected at the phosphoproteome level were independent from the total protein level changes. Focusing on the phosphoproteome profiles, principal component analysis (PCA) and hierarchical clustering analysis revealed that glargine-treated groups were distinctly separated, in contrast to the vehicle-treated groups (Fig. 2g, Supplementary Fig. 3a). The data quality was validated, which revealed that despite these observations, the number of phosphosites quantified (except 1 sample) (Supplementary Fig. 3b), so as the sample coefficients of variation among all treatment groups (Supplementary Fig. 3c) were highly similar. In the subsequent differential expression analysis, while the vehicle-treated groups (i.e., WT-Glargine vs KO-Glargine) failed to reveal any meaningful differences in their phosphoproteome profiles (Supplementary Fig. 3f), comparisons made between insulin-stimulated versus unstimulated groups in either *Atm*-WT or *Atm*-KO samples revealed obvious differences in the phosphoproteome profiles (Supplementary Fig. 3d-e). Specifically, among the *Atm*-WT samples, pathway analyses with the proteins derived from unique upregulated phosphosites in glargine-treated group were implicated widely in a set of central carbon metabolic pathways (i.e., glycolysis/gluconeogenesis, pentose phosphate pathway, fructose and mannose metabolism and central carbon metabolism in cancer) (Supplementary Fig. 3d). Such pattern was however not observed among the *Atm*-KO samples (Supplementary Fig. 3e). Notably, a number of conventional protein phosphorylation events known to be triggered by insulin were detected in both glargine-treated *Atm-WT* and *Atm-KO* samples groups, which included pTSC2 at S939[36]; pARAF at S580[37]; pPFKL at S775[38]; pPDHA1 at S293[39]; pACLY at S455[38] and pMDH1 at S241[38] (Supplementary Fig. 3g). These hinted that ATM is likely regulating certain downstream actions of insulin, but not all.

Based on these initial findings, we then directly compared the phosphoproteome profiles between the two insulin-treated groups (i.e., *Atm*-WT + Glargine vs. *Atm*-KO + Glargine), which revealed a total of 79 differentially changed phosphopeptides (Fig. 2h). Cellular component analysis with the 54 phosphopeptides belonged to 40 proteins differentially enriched in insulin-treated *Atm*-WT group revealed their predominant localization in various neuronal compartments (Fig. 2i). This finding not only supports the notion that phosphorylation events primarily occur in neurons within the cerebellum but also suggests a preference for subcellular enrichment in "neuronal cell body", "somatodendritic compartment" and "neuron projection cytoplasm," indicating that the cytoplasm is a key site for phosphorylation events where dimeric activated ATM can be found[40]. Functionally, KEGG pathway enrichment analysis revealed their potential roles in aerobic glycolysis (i.e., "glycolysis/gluconeogenesis", "metabolism in cancer", and "HIF-1 signaling pathway") and RNA splicing (i.e., "Spliceosome" and "RNA degradation") (Fig. 2j). Despite the apparent lack of overlap in biological functions between these two sets of pathways, a more in-depth examination of the protein-protein association network using

STRING has revealed interconnectedness among proteins enriched in them (Fig. 2k). Of particular note is the presence of the protein HNRNPA1 (Heterogeneous nuclear ribonucleoprotein A1) in the network, a key RNA-binding protein that regulates the alternative splicing of PKM pre-mRNA, thereby favoring the formation of PKM2 mRNA over that of PKM1[41,42]. This suggests that the enrichment of phosphoproteins related to RNA splicing may further regulate the effectiveness and capacity of aerobic glycolysis in *Atm*-WT neurons exposed to glargine (i.e., insulin)—a metabolic status that supplies metabolites for synaptic formation and neurite outgrowth during brain development[43]. In the contrary, phosphopeptides enriched in insulin-treated *Atm*-KO group (i.e., 20 phosphopeptides belonged to 19 proteins) failed to be yielded into any biological function (Supplementary Fig. 3h).

The above observation suggested that such changes are unique to an activated insulin-ATM signaling network. This notion was supported further by the kinase enrichment analysis[44]. With the phosphopeptides enriched in the insulin-treated *Atm*-WT group, ATM emerged the top kinases predicted, displaying as well an interrelating network with a group of kinases associated with insulin signaling (CSNK2A1[45], PRKCB[46], CSNK1D[47], PRKDC[48], CSNK1E[49]) (Fig. 2l). Such pattern was not observed with phosphopeptides enriched from the insulin-treated *Atm*-KO group (Supplementary Fig. 3i). Moreover, the total number of kinases enriched in the *Atm*-WT group exceeded that in the *Atm*-KO, with the majority (18/23) of uniquely identified kinases in the former playing roles in insulin signaling (CSNK2A1[45], PRKCB[46], MAPK14[50], MAPK12[51], MAPK10[52], RPS6KA3[53], MAPK9[54], MAPK8[55], MAPK7[56], MKNK1[57], MAPK1[58], SGK1[59], PRKACB[60]). (Supplementary Fig. 3j). At the substrate level, a substantial proportion (i.e., 21/54) of phosphopeptides enriched in insulin-treated *Atm*-WT group were modulated at SQ/TQ residues (Fig. 2m)—the substrate targeting motif of the ATM kinase[61]. Notably, these hits included PKM2 and HIF1α (Fig. 2n)—the common regulators of aerobic glycolysis effect[62,63]. The necessity of ATM in mediating the insulin effect on PKM2 and HIF1α SQ/TQ phosphorylation was further validated by a gene knockdown analysis (Supplementary Fig. 2h). In the presence of ATM, acute glargine exposure resulted in robust nuclear translocation of PKM2 (Figs. 2o, r, s), in the expenses of its cytosolic role as a metabolic enzyme (Fig. 2p). Such changes were likely attributed to the loss in its tetramer quaternary structure upon SQ/TQ site phosphorylation (Fig. 2t, Supplementary Fig. 2h). Similarly, a delayed effect on nuclear localization and DNA binding effect of HIF1α was observed (Fig. 2o, q, Supplementary Fig. 2h). Such a slower response could be in part due the inherent instability of HIF1α in the cytoplasm under normoxic condition[64], and additional time is needed for the protein to build up in the system. Complementing these phenomena, in silico docking simulations further supported that phosphorylation of the conserved T163 residue on HIF1α is predicted to enhance its interaction with the R281 residue on its nuclear binding partner ARNT (Aryl hydrocarbon receptor nuclear translocator) (Supplementary Fig. 2i); whereas for PKM2, docking studies using a PKM2 monomer structure demonstrated that phosphorylation at T328 may also enhance its interactions with its partner nuclear importing factor importin-α3 (Supplementary Fig. 2j). Together, this observation demonstrated that insulin-activated ATM modulates the protein stability as well as nuclear activities of these key regulators of aerobic glycolysis in the cerebellum.

## Defective insulin-ATM signaling results in glycolytic insufficiency and a shift towards glutamine dependence in the cerebellum

With these findings, it is then crucial to comprehend the molecular details of how ATM mediates the metabolic impact of insulin and the repercussions on the cerebellar metabolic balance in the absence of it. Global metabolic profiling analysis of mouse cerebellar tissues collected at 3 and 6 months of age unveiled a common set of perturbed

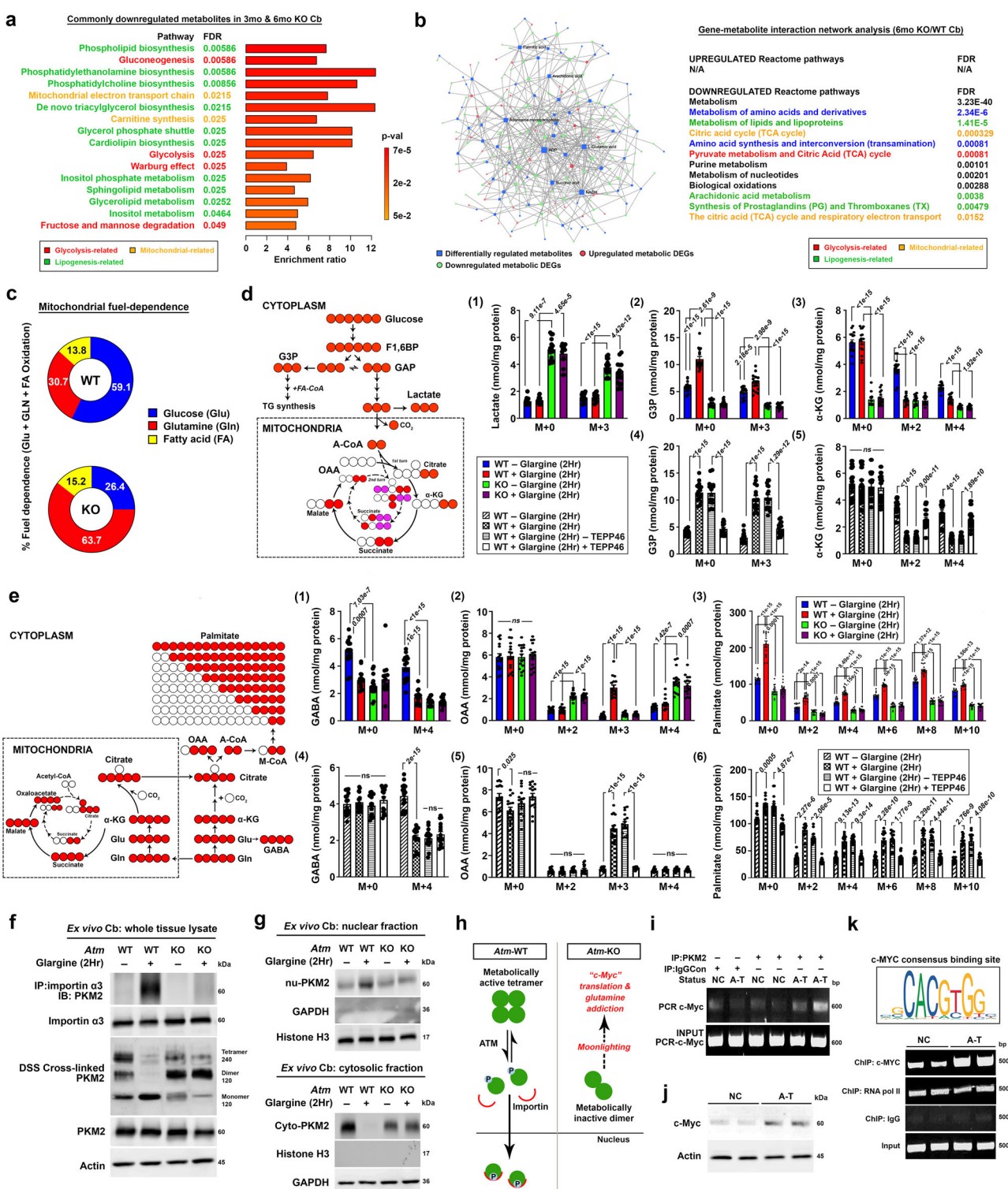

metabolites (Supplementary Fig. 5a, b). Those related to "methionine metabolism" and "glycine and serine metabolism" were consistently elevated in the absence of ATM (Supplementary Fig. 5c), while the commonly downregulated metabolites were implicated mostly in the central glycolytic network (e.g., Warburg effect, gluconeogenesis, glycolysis, fructose and mannose degradation), suggestive of a reduced glycolytic activity (Fig. 3a). Also, metabolites belonged to multiple lipid biosynthesis pathways (e.g., phospholipid biosynthesis, de novo triacylglycerol biosynthesis, inositol phosphate metabolism, sphingolipid metabolism, glycerolipid metabolism, glycerol

phosphate shuttle, cardiolipin biosynthesis, phosphatidylcholine biosynthesis, phosphatidylethanolamine biosynthesis) which reflects the anabolic capacity of lipid macromolecules, were diminished as well (Fig. 3a). For those take part in the mitochondrial bioenergetic networks (e.g., carnitine synthesis and mitochondrial electron transport chain) were also reduced in the absence of ATM (Fig. 3a). Matching with these findings, bulk transcriptomic profiling of cerebellar tissues harvested from 6-month-old mice revealed that differentially downregulated genes were highly involved in glycolysis, adipogenesis (i.e., lipid anabolism), and myogenesis (i.e., protein anabolism)

**Fig. 3 | ATM deficiency disrupts glycolytic capacity and reprograms to glutamine dependence in the cerebellum. a** Metabolite set enrichment analysis of commonly downregulated metabolites enriched in cerebellar tissues of 3- and 6-mo *Atm-KO* mice (*n* = 6, Globaltest with Bonferroni correction[173]). **b** Integrated gene-metabolite network analysis in cerebellar tissues harvested from 6-mo *Atm*-WT and KO mice. Significantly changed KEGG metabolic genes and metabolites were mapped to Reactome pathways (*n* = 4 for transcriptome; *n* = 6 for metabolome, Globaltest with Bonferroni correction[173]). **c** Pie chart illustrating the changes in proportions of glucose, glutamine and fatty-acid dependence in acute ex vivo cerebellar culture (*n* = 3). Original data is shown in Supplementary Fig. 5g. **d** Left: schematics of $^{13}C_6$-U-glucose carbon flow. Right: mass isotopologue analysis of lactate, glycerol-3-phosphate (G3P), α-ketoglutarate (α-KG) in ex vivo cerebellar cultures [*n* = 16, one-way ANOVA for all except for (1) "M + 0 Lactate" where Kruskal-Wallis test was used]. **e** Left: schematics of $^{13}C_5$-U-glutamine carbon flow. Right: mass isotopologue analysis of γ-Aminobutyric acid (GABA), oxaloacetate (OAA), palmitate in ex vivo cerebellar cultures [*n* = 16, one-way ANOVA for all except for (1) "M + 0 GABA", (2) "M + 4 OAA", (4) "M + 0 GABA" and (6) "M + 0 and M + 2

Palmitate" where Kruskal-Wallis test was used]. **f** Representative immunoblots on importin α3-PKM2 interaction and PKM2 protein quaternary structure status (*n* = 16). Quantifications presented in Supplementary Fig. 5i. **g** Representative immunoblot images on subcellular localization of PKM2 (*n* = 16). Quantification presented in Supplementary Fig. 5j. **h** Schematics of possible fates of PKM2 cytosolic dimer/monomers. **i** Representative blot images of RNA-immunoprecipitation assay for the interaction between PKM2 and the IRES region of the *c-Myc* mRNA harvested from human patient fibroblasts (*n* = 10). Quantification presented in Supplementary Fig. 6b. **j** Representative immunoblots showing levels of c-Myc protein in human patient fibroblasts (*n* = 12). Quantification presented in Supplementary Fig. 6c. **k** Schematics of human *c-Myc* consensus binding site sequences. Representative ChIP-qPCR blots showing levels of *c-Myc* binding to glutaminase-2 (*GLS2*) promoter in human patient fibroblasts (*n* = 10). Quantification presented in Supplementary Fig. 7d. Unless otherwise specified, all ex vivo glargine and TEPP-46 treatments were performed in 100 nM for 2 h. Values represent the mean ± s.d. *N* presents biological replicates. Source data are provided as a Source Data file.

(Supplementary Fig. 5d–f, Supplementary Dataset 3). Subsequently, an integrated gene-metabolite interaction network analysis consistently underscored such metabolic disturbances in the cerebellum under ATM deficiency (Fig. 3b).

These similar findings, together with the physiological observations at systemic level (Fig. 1), collectively highlighted an inherent incompetence in utilizing glucose as a source of fuel for energy and carbons for macromolecule biosynthesis. To validate these notions, mitochondrial fuel-dependence analysis was conducted in ex vivo cerebellar cultures, which revealed a prominent shift from glucose to glutamine as the primary source of respiratory fuel (Fig. 3c, Supplementary Fig. 5g). Further isotopologue tracing analysis of heavily labelled glucose ($^{13}C_6$-U-glucose) demonstrated that even under unstimulated condition, glucose flux towards the formation of glycerol-3-phosphate (G3P, M + 3) and α-ketoglutarate (α-KG) (M + 2, M + 4 from 2nd round TCA) were already compromised in *Atm*-KO; except that towards lactate production (M + 3) [Fig. 3d(1-3)]. Upon a short 2-hour exposure to glargine, *Atm*-WT group promptly redirected glucose carbons towards various anabolic fates [i.e., G3P (M + 3), riboluse-5-phosphate (M + 5), serine (M + 3), sphinganine (M + 2) and glycine biosynthesis (M + 2) [Fig. 3d(2), Supplementary Fig. 5h]] while reducing the relative commitment to the formation of catabolic products such as α-KG (M + 2, M + 4) [Fig. 3d(3)]. Notably, this response was partially counteracted by TEPP-46, a tetrameric PKM2 stabilizer [Fig. 3d(4-5), Supplementary Fig. 5h]. In contrast, glargine had little effect on glucose fate in *Atm*-KO group [Fig. 3d(1-3)]. With regard to the heightened glutamine dependence instead, fate tracing by heavily labelled glutamine ($^{13}C_5$-U-glutamine) revealed *Atm*-KO exhibited inherently reduced its contributions to GABA biosynthesis [Fig. 3e(1)], whereas its oxidative fate in mitochondria, as indicated by elevated levels of M + 2 and M + 4 oxaloacetic acid (OAA), was enhanced under all treatment conditions as compared to the *Atm*-WT counterparts [Fig. 3e(2)]. Contrary to this, in *Atm*-WTs, baseline glutamine-mediated GABA synthesis was high but reduced sharply upon glargine stimulation [Fig. 3e(1)]. This change was attributed to the shift of glutamine fate towards its reductive anabolic fate as evident by the increased levels of cytosolic M + 3 OAA and heavily labelled palmitate [Fig. 3e(2-3)]. Such anabolic effects of glargine were effectively prohibited by TEPP-46 [Fig. 3e(4-6)].

These findings hinted that the decrease in the active tetrameric configuration of PKM2, brought on by the insulin-ATM signaling, is a downstream effector response regulating the catabolic versus anabolic fates of glucose and glutamine in cerebellum. Further supporting this notion, immunoblotting confirmed an enhanced PKM2-importin-α3 binding in *Atm-WT* following glargine treatment, leading to predominant signals of monomeric and nuclear-localized PKM2 forms (Fig. 3f, g, Supplementary Fig. 5i, j). Conversely, in *Atm*-KOs, PKM2

existed predominantly as cytosolic, low pyruvate kinase metabolic activity dimers instead (Fig. 3f, g, Supplementary Fig. 5i-j). In this status, dimeric PKM2 may directly facilitate internal ribosome entry site (IRES)-dependent translation of c-Myc (Supplementary Fig. 6a)[65]—a pivotal regulator of glutamine dependence (Fig. 3h)[66]. In human A-T patient fibroblasts, RNA-immunoprecipitation assays unveiled an increased interaction between PKM2 and the *c-Myc* mRNA IRES region, hence an elevated c-Myc protein level was detected (Fig. 3i, j, Supplementary Fig. 6b, c). Glutaminase-2 (GLS2), the enzyme crucially induced in A-T cerebellar tissues[67], promotes glutamine dependence as the major bioenergetic fuel in the mitochondira[68]. Notably, common c-Myc-binding sites were identified at both mouse and human *GLS2* promoters (Supplementary Fig. 7a–c). Chromatin immunoprecipitation (ChIP)-qPCR analysis focusing on the -1930 to -1401 base pair region upstream of the human *GLS2* promoter revealed augmented c-Myc binding in A-T cells (Fig. 3k, Supplementary Fig. 7d). Collectively, these data not only signify a significant metabolic reconfiguration, but also a diminished insulin-driven anabolic switching effect when ATM is lost. At the molecular level, these metabolic shifts were orchestrated by the predominant accumulation of dimeric PKM2 in the cytosolic compartment.

## Zebrin-II/ALDOC-positive Purkinje cells, reliant on glycolysis, are more susceptible to metabolic challenges imposed by ATM deficiency

On the top of exhibiting a high demand for energy, neurons in the maturing brain, in general, also relies on essential macromolecules like lipid- and amino acid- derivatives for proper growth and maintenance of synaptic functions[69]. Extended from the observation of reduced glycolytic-derived central carbon metabolites and neurotransmitter-related metabolites (e.g., GABA, serine, choline-reaction intermediates) (Fig. 3), targeted lipidomic analysis in cerebellar tissues revealed that the abundance of several species of fatty acids, phospholipids and sphingolipids was reduced in *Atm*-KO as well (Fig. 4a), hinting potential impacts to tissue and cell physiologies. Immunoblotting analyses further supported this notion, revealing signal reductions in markers of Purkinje cells (calbindin) but not those of granule neurons (NeuN) (Fig. 4b, Supplementary Fig. 8a). Further immunohistology examinations confirmed specific losses in IP3R1+ Purkinje cell number, soma diameter, and climbing fiber connections (VGlut2) (Fig. 4c–e, Supplementary Fig. 8b), suggesting an impaired dynamics of synaptogenesis and synaptic stabilization process in *Atm*-KO[70]. Notably, these changes were more prominently found within vermal posterior [i.e., Lobule (L) 6-L9] and flocculonodular lobes (i.e., L10) (Fig. 4c–e, Supplementary Fig. 8b), regions within the cerebellum that concern with whole-body posture, locomotion and balance[71,72].

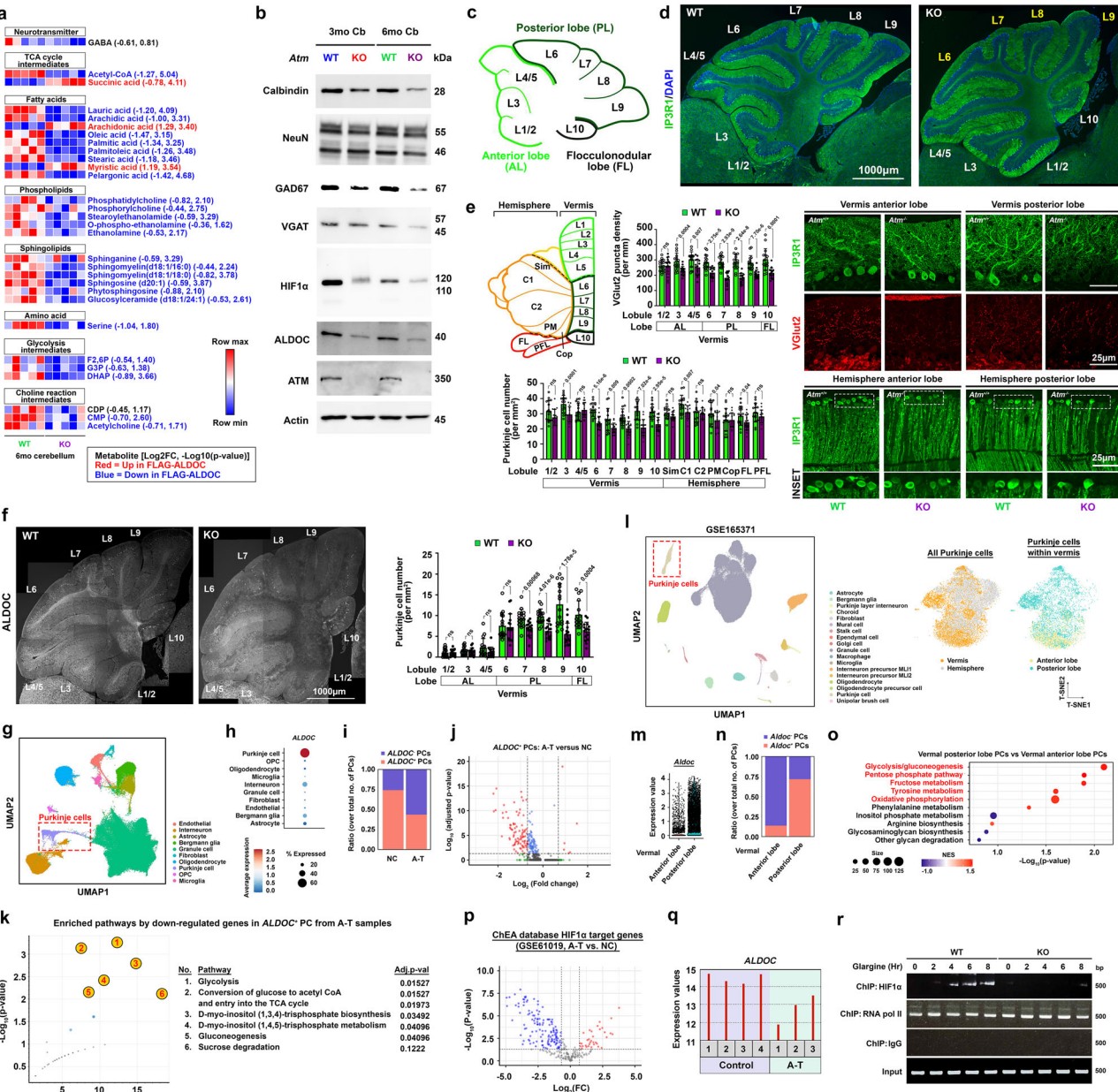

**Fig. 4 | ALDOC-positive Purkinje cells reliant on glycolysis are selectively more vulnerable to metabolic reprogramming due to ATM deficiency. a** Alterations in lipid-related metabolites (n = 5, two-tailed unpaired t-test). **b** Representative immunoblots of Purkinje cell (Calbindin, ALDOC) and granule neurons (NeuN) and GABAergic synapses (GAD67, VGAT) markers (n = 12). Quantifications presented in Supplementary Fig. 8a. **c** Schematics unveiling the characteristic foliation arrangement of cerebellar vermis. **d** Representative images depicting specific alterations in Purkinje cell density (n = 12). **e** Schematics of the gross anterior-posterior anatomy of the cerebellum. Key regions include SIM (simple lobule), C1 (crus 1), C2 (crus 2), PM (paramedian lobule), FL (flocculus), and PFL (paraflocculus). Quantifications and representative images demonstrating alterations in Purkinje cell (IP3R1 +) density and the extent of climbing fiber connections (VGlut2) are shown (n = 16, two-tailed unpaired t-test). **f** Representative images and quantifications of ALDOC+ signals across distinct lobes (n = 16, two-tailed unpaired t-test). **g–k** Referencing a published dataset[76], **g** UMAP plot of and **h** relative expression levels of ALDOC different cell types is shown. **i** Relative cell ratios of ALDOC+ and

ALDOC- Purkinje cells. **j** DEG profiles in ALDOC+ Purkinje cells (Wilcoxon rank-sum test with Bonferroni correction). **k** Pathway enrichment of down-regulated DEGs in ALDOC+ Purkinje cells conducted on Enrichr[172] (Fisher exact test with correction). **l–o** Referencing a published dataset[163], **l** UMAP plot and t-SNE plots reveal the relative abundance of Purkinje cells in different cerebellar regions. **m** Relative expression level of ALDOC and **n**, cell ratio of ALDOC+ and ALDOC- Purkinje cells. **o** Gene set enrichment analysis of KEGG metabolic genes expressed in Purkinje cells found in different lobe regions (Weighted Kolmogorov–Smirnov test with Benjamini-Hochberg correction). **p, q** Referencing a published dataset[162], **p** differential expression levels of HIF1α target genes and **q** the relative expression level of ALDOC in A-T versus NC samples (Limma with Benjamini-Hochberg correction). **r** Representative ChIP-qPCR blots illustrating the levels of HIF1α bound to Aldoc promoter in the acute ex vivo cerebellar culture exposed to 100 nM glargine (n = 8). Quantifications presented in Supplementary Fig. S11f. Values represent the mean ± s.d. N presents biological replicates. Source data are provided as a Source Data file.

Indeed, the innate properties of Purkinje cells are not uniform throughout different cerebellar lobules and can be sub-classified based on their Zebrin-II/ALDOC expression patterns[73]. Levels of ALDOC, a brain-specific isozyme of glycolytic aldolase highly expressed in a subpopulation of Purkinje cells located within the vermal posterior and flocculonodular lobes[74], were more profoundly reduced in Atm-KO (Fig. 4c-e). This change was further validated by a similar pattern of losses in numbers of ALDOC+ Purkinje cells located

at lobule L7-L10 (Fig. 4f, Supplementary Fig. 9). Intriguingly, as also a glycolytic enzyme[75], these changes in ALDOC correlated with the pattern of loss in pIRS1 (Y612) signals, a marker of active insulin response (Supplementary Fig. S10). The human relevance of such ALDOC changes was alternatively validated in an A-T cerebellar single-cell transcriptomic dataset[76] (Fig. 4g–k, Supplementary Fig. 11d, e). Further comparison of single-cell transcriptome profiles of mouse Purkinje cells extracted from anterior (AL) and posterior/flocculonodular lobes (PL) of the vermal region not only validated that Purkinje cells within the PL were predominantly ALDOC+ (Fig. 4m, n). Additional investigations unveiled that this subset of Purkinje cells situated within the vermal PL region exhibited relatively heightened expression of genes associated with "glycolysis/gluconeogenesis," "pentose phosphate pathway," "fructose metabolism," and "oxidative phosphorylation," suggesting these cells by default have relatively greater activities and reliance in the central carbon metabolism network (Fig. 4o).

The aldolase reaction is canonically situated at the center of the glycolytic pathway, implicated in enhancing lipid and cholesterol biosynthesis[77]. Here, by ectopic expression of *ALDOC* in HT22 neuronal cells, this not only enhanced their glycolytic capabilities (Supplementary Fig. 11a-b), but also the production of glucose-derived triglyceride precursors like DHAP and G3P, as well as various species of fatty acids, phospholipids and sphingolipids (Supplementary Fig. 11c), thus validating the interrelationships among ALDOC level, glycolysis and lipid anabolism. Previous studies indeed revealed that *Aldoc* is a downstream target of HIF1α[78–80]. This suggests that the losses of chronic insulin-activated ATM effects on HIF1α stabilization and subsequent nuclear localization in *Atm*-KO may compromise the expression of *Aldoc* as well. Accordingly, targeted gene expression analysis of 314 HIF1α targets extracted from the ChEA database revealed predominant reductions in A-T cerebellar tissues, including that of *ALDOC* as well (Fig. 4p-q). Further in silico promoter analysis revealed a conserved HIF1α binding region in both the mouse and human *ALDOC* gene (Supplementary Fig. 12). ChIP-qPCR then revealed a progressive increase in the association of HIF1α with the evolutionarily conserved hypoxia response element (HRE) positioned at bases -1308 to -1299 within the *Aldoc* gene promoter, only in *Atm*-WT acute cerebellar tissue cultures after a prolonged exposure to glargine (>4 h) (Fig. 4r, Supplementary Fig. 11f). These findings imply that the levels of ALDOC and its impact on glycolytic capabilities are contingent upon a more chronic effect of insulin that is known to be crucial for normal growth and development during childhood and early adolescence[81,82]—life stages that the maturation of the cerebellum is also the most robust[83]. These together explain how Zebrin-II/ALDOC-positive Purkinje cells are selectively more vulnerable under ATM deficiency, and how specific anatomical degeneration due to the loss of these cells results in motor-related phenotypes in A-T.

## Nuclear-localized PKM2 co-activates HIF1α the aerobic glycolysis regulator to reshape a metabolic landscape that favors glucose anabolic fate towards lipid biogenesis

Our data so far revealed that in the absence of ATM, insulin-sensitive cells fail to translate the hormonal signal into an instantaneous metabolic transition that fosters anabolic reactions that are particularly important for general growth and development during early childhood to early puberty[84]. During these life stages, hyperinsulinemia is considered as part of the physiological responses against reduced insulin sensitivity elicited by growth hormones[85–87]. Under other circumstances, however, hyperinsulinemia is considered as pathological and it is a hallmark of prediabetes[88], correlated to elevated risks of overweight and obesity in ordinary individuals[89,90]. To investigate whether this pathophysiological outcome is otherwise stemmed from the sustained activation of ATM, six-month-old *Atm*-WT and *Atm*-KO mice were administered with daily doses of glargine for a duration of four weeks (in vivo half-life = 12–13.5 h)[91,92] to simulate a persistent

hyperinsulinemia condition. This was either conducted alone or in combination with the blood-brain barrier-permeable PKM2 activator TEPP-46[93] (Fig. 5a). After completing the treatment paradigm, majority of WT mice administered with glargine alone or in conjunction with TEPP-46 exhibited only slight indications of impaired glucose disposal (Supplementary Fig. 13a). However, indications of compensatory hyperinsulinemia and reduced insulin sensitivity were much evident and obvious, denoted by elevated levels of fasting plasma insulin and HOMA-IR values exceeding 2.2 after the IGTT (Supplementary Fig. 13b-c). Importantly, the marked increases in body weight observed in mice after receiving glargine alone were alleviated in those co-administered the TEPP-46 (Fig. 5b), suggesting that in the presence of a metabolically functional PKM2 tetramer, the pro-anabolic effect of chronic insulin is abolished. Conversely, given that the lack of ATM can directly nullify the anabolic effect of insulin as well (Figs. 2–3), *Atm*-KO mice of this age were inherently insulin-insensitive and could even consider pre-diabetic or even diabetic prior to the commencement of this treatment regimen [Fig. 1c, d, Supplementary Fig. 13a–c (*Atm*-KO + vehicle group)]. Therefore, subsequent and sustained exposure to chronic glargine, whether administered alone or in conjunction with TEPP-46 over the four-week period, did not result in discernible variances in glucose disposal, fasting plasma insulin levels, HOMA-IR values, or body weight, as compared to their corresponding vehicle-treatment group (Fig. 5b and Supplementary Fig. 13b, c). Likewise, indirect calorimetry assessment revealed that in WT mice, extended exposure to glargine alone in the absence of TEPP-46 impeded their ability to efficiently adjust the utilization of metabolic substrates between light and dark cycles (Fig. 5c). The resulting prolonged carbohydrate utilization during both cycles (indicated by a RQ = 0.9–1) was likely sustained by increased standard laboratory chow intake (62% carbohydrate)[94] (Supplementary Fig. 13d). This phenomenon further explains how more rapid increase in body weight was observed in these mice (Fig. 5b), as was linked to increased adipose tissue deposits (Fig. 5d) and the accumulation of large-sized oil droplets in the liver (Fig. 5e). Despite these changes, no obvious differences in the majority of the skeletal muscle mass were found (Supplementary Fig. 13e). In contrast to these findings in *Atm*-WT mice, *Atm*-KO mice failed to respond to glargine or when co-treated in combination with TEPP-46 (Fig. 5b-e). Their inherent insulin resistance and glucose intolerance results in sustained reliance on lipid and protein utilization (indicated by RQ = 0.7–0.8) throughout both light and dark cycles (Fig. 5c). Despite these mice also consumed a comparable amount of standard laboratory chow (62% carbohydrate)[94] as the *Atm*-WT mice on vehicle (Supplementary Fig. 13d), the absence of the anabolic impact of insulin resulted in inefficient carbohydrate utilization, as evidenced by the lack of biologically meaningful changes in respiratory quotient measurements at times when food was heavily consumed during the dark cycles (Fig. 5c), so as the status of glucose intolerance reflected by the IGTT (Supplementary Fig. 13a). This inefficiency extended to limited lipogenic capacities (Fig. 5d) while heightening lipolysis instead, as indicated by their significantly lower adipose tissue mass and accumulation of relatively small-sized oil droplets in the liver[95] (Fig. 5d, e). These findings underscore the essential role of ATM in enabling the transition between catabolic and anabolic metabolic states at the whole-body systemic level.

To underscore if these systemic peripheral metabolic alterations would also be reflected in the cerebellum, cerebellar tissues were also collected from these animals for further analysis. As already shown previously (Fig. 2), following extended insulin exposure, activated ATM phosphorylates and promotes nuclear localization of both PKM2 and HIF1α. The combined effects also enabled nuclear-localized PKM2 to engage with and function as a co-activator of HIF1α[96] in *Atm*-WT cells (Fig. 5f–h, Supplementary Fig. 13f-g). Remarkably, in specimens collected from *Atm*-WT subjects receiving TEPP-46 co-treatment, PKM2 stabilization in its tetrameric quaternary structure and

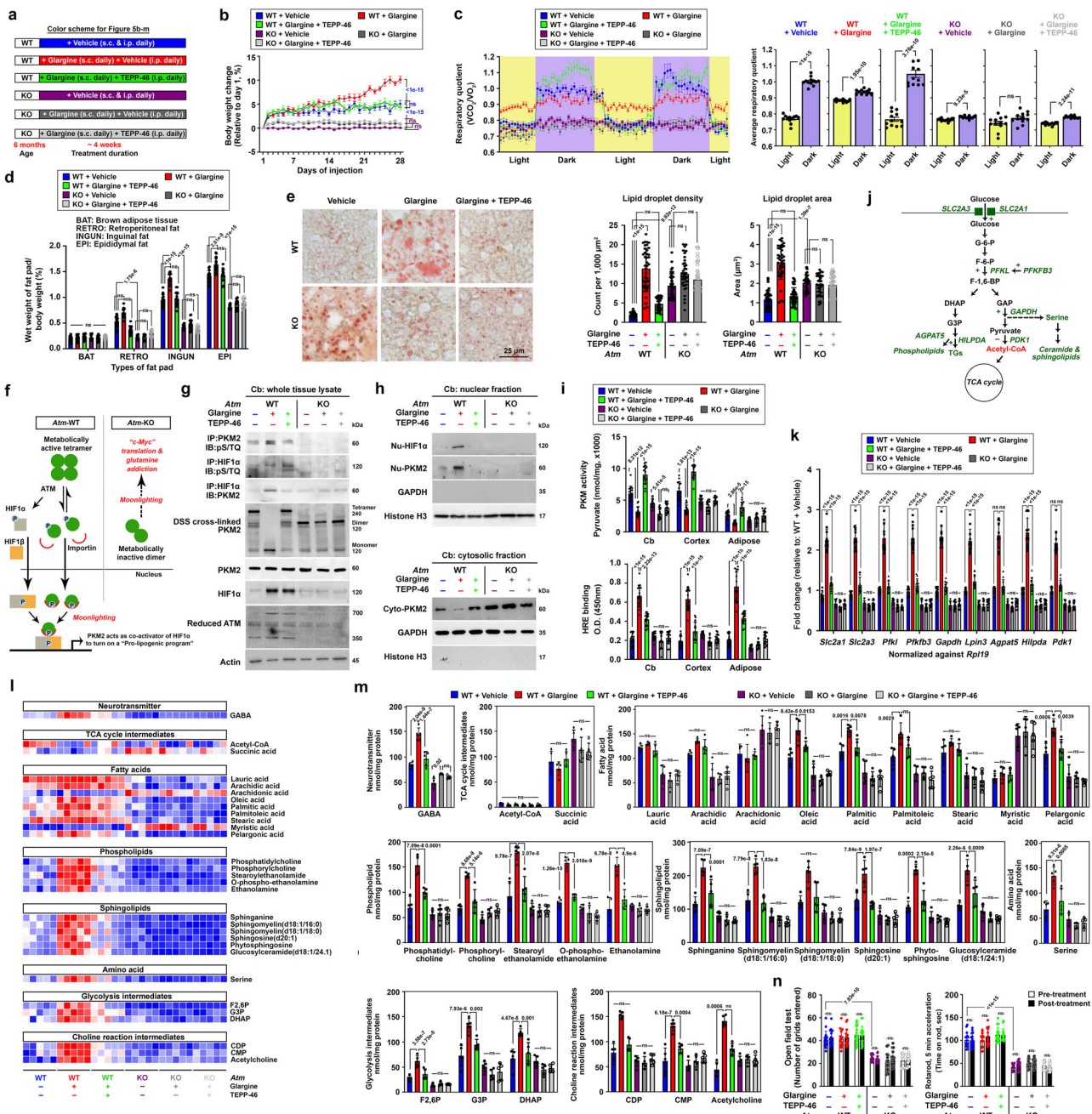

**Fig. 5 | Chronic activation of ATM upon sustained exposure to insulin promotes glucose anabolic flux towards lipid biogenesis. a** Schematic illustrates the daily regimen of glargine (50 Unit/kg body weight, subcutaneous) ± TEPP-46 (50 mg/kg body weight, intraperitoneal) administration over a 4-week period. **b** Changes in mean body weight throughout the treatment paradigm ($n$ = 16, two-way ANOVA). **c** Chronological variations in indirect calorimetry assessments highlighting alterations in respiratory quotient. Summary of changes observed between light and dark cycles ($n$ = 12; two-tailed unpaired t-test). **d** Changes in percentage of wet weight in various fat pads ($n$ = 16, one-way ANOVA for all except for "BAT" and "RETRO" groups where Kruskal-Wallis test was used). **e** Representative images of liver oil red O staining and variations in lipid droplet densities and areas ($n$ = 40 images, Kruskal-Wallis test). **f** Schematics illustrate the transition between PKM2 tetramers, dimers, and monomers, and potential outcomes for nuclear monomers. **g**, **h** Representative immunoblots depicting the protein dynamics of PKM2 and HIF1α in **g** whole cell and **h** subcellular fractionated lysates harvested from

cerebellar tissues ($n$ = 12). Quantification presented in Supplementary Fig. 13f, g. **i** Assessment of HRE binding and PKM metabolic activities in tissues collected after the drug treatment paradigm ($n$ = 16, One-way ANOVA). **j** Schematics illustrate HIF1α targets (i.e., green highlights) with roles as pivotal regulators of the glycolytic pathways. **k** Expression levels of HIF1α targets, as indicated in **j**, in cerebellar tissues ($n$ = 12, one-way ANOVA for all except for "*Agpat5*" and "*Pdk1*" where Kruskal-Wallis test was used). **l**, **m** Alterations in **l**, targeted lipid-related metabolite in harvested cerebellar tissues and **m** statistical analyses of levels of each metabolite ($n$ = 5, one way ANOVA for all except for "acetyl-CoA", "Sphingomyelin(d18:1/18:0)", "CDP" and "acetylcholine" where Kruskal-Wallis test was used). **n** Summary of longitudinal behavioral performances in the open-field test and rotarod tasks before and after the administration of drug treatments ($n$ = 16, two-way ANOVA). Values represent the mean ± s.d. *N* presents biological replicates. Source data are provided as a Source Data file.

metabolic activities were anticipated (Fig. 5g–i). This stabilization effect also counteracted ATM-mediated SQ/TQ phosphorylation on PKM2, likely an outcome of stearic hinderance caused by forced tetramerization. Subsequently, TEPP-46 also abolished glargine-stimulated nuclear translocation of PKM2 and its interaction with HIF1α (Fig. 5g, h, Supplementary Fig. 13f-g). Consistent with the observation in peripheral tissues such as the adipose tissue, chronic glargine-induced nuclear localization of HIF1α in conjunction with nuclear monomeric PKM2, which facilitated a robust nuclear HRE binding in the central nervous system (i.e., cerebellum and cerebral cortex) (Fig. 5i). Consequently, the expression levels of HIF1α target genes, which encompass genes encoding proteins and enzymes that facilitate the utilization of glucose carbons for fatty acids, phospholipids and sphingolipids biosynthesis, were also upregulated (Fig. 5j, k and Supplementary Fig. 14). The lipogenic impacts in the cerebellum were corroborated through targeted lipidomic assessment in the Atm-WT samples (Fig. 5l-m). Significantly, in the context of human A-T, lipogenic pathways were consistently compromised, as reflected from the bulk transcriptome data of human cerebellar samples (Supplementary Fig. 15). Specifically, this set of lipogenic pathways were also relatively more compromised ALDOC+ but not ALDOC- Purkinje cells in A-T brains (Fig. 4j–k, Supplementary Fig. 11d, e). Among the Atm-KOs, the absence of molecular responses suggested that the administration of glargine or TEPP-46 did not yield in any beneficial or detrimental effects on their behavioral performance (Fig. 5n). These findings together align with the earlier work indicating that defective lipid dyshomeostasis is a prevailing pathological mechanism in inherited cerebellar ataxia[97].

## Supplementation of α-ketoglutarate (α-KG), the α-keto acid of glutamine, mitigates metabolic challenges associated with ATM deficiency

Our in vivo and in vitro analyses suggest that in ATM-deficient conditions, glycolytic insufficiency triggers a compensatory dependence on amino acids, particularly glutamine, as an alternative mitochondrial carbon source (Fig. 3c–e). While the major source of glutamine is skeletal muscles[98], constant breakdown of proteins to sustain its increased demand as a fuel by the body can result in a reduction in lean muscle mass, further heightening the risk and progression of insulin resistance and other metabolic complications[99]. To mitigate this hypercatabolic effect, one straightforward strategy may involve the dietary supplementation of glutamine. While short-term supplementation over a period of 2 weeks showed initial promise[100], concerns over ammonia ($NH_3$) neurotoxicity[101], particularly if such interventions are prolonged for the management of genetic disorders, may limit its potential.

α-KG, the α-ketoacid derived from glutamine/glutamate, plays a pivotal role as the nexus between cellular carbon and nitrogen metabolism[102]. As a linear keto acid, this metabolite can be directly absorbed in the stomach and small intestine[103,104]. α-KG in circulation could then traverse the blood-brain barrier through both carrier-mediated and passive diffusion mechanisms[105,106]. At the target cell surface, the transmembrane transportation of a conjugated form of α-KG as divalent anions can be efficiently facilitated by sodium-dependent dicarboxylate transporters[107]—NaDC1/SLC13A2 and NaDC3/SLC13A3. Indeed, gene expression levels of them were induced in cerebellar tissues of A-T (Fig. 6a). Once inside the cell, α-KG has the capacity to transform into glutamate through a transamination reaction facilitated by glutamic-oxaloacetic transaminase 1-like 1 (GOT1L1), even in scenarios where the major GOT1 enzyme is suppressed (Fig. 6a). This conversion also generates cytosolic OAA, which may enter the mitochondria via specific carriers encoded by SLC25A34 and SLC25A35[108] to bolster the TCA cycle. Concurrently, the glutamate produced during this process can undergo two metabolic fates: either it could be transformed into GABA by the cytosolic glutamate

decarboxylase-1 (GAD1) or enter the mitochondria via glutamate carrier-2 (GC2/SLC25A18) where expression of these two genes both trended upward in A-T cerebellar tissues as well (Fig. 6a). Once inside the mitochondria, glutamate may deaminate back to α-KG by glutamate dehydrogenase (GLUD2) and enter the TCA cycle (Fig. 6a). The ammonia liberated from this deamination reaction can be utilized by the cytosolic glutamate-ammonia ligase (GLUL), which converts glutamate into glutamine to support GABA synthesis (Fig. 6a). In this sequence, exogenous α-KG could effectively serve as a substitute for endogenous muscle-derived glutamine in generating not only a carbon source for sustaining the TCA cycle (Fig. 6a), but also a balanced molarity of ammonia for GABA synthesis, as supported further by subsequent isotopologue tracing experiments (Fig. 6b-c). In ex vivo cerebellar cultures derived from Atm-KO mice, the introduction of heavily labelled disodium-conjugated 1,2,3,4-$^{13}C_4$-α-KG effectively competed against the involvement of heavily labelled $^{13}C_5$-U-glutamine as mitochondrial fuel (Fig. 6c), suggesting the potential capability of α-KG to surrogate the exhaustion of endogenous glutamine derived from skeletal muscles or neurotransmitters. Significantly, tissue ammonia level was reduced as well (Fig. 6d), affirming that α-KG represents a less hazardous but a more efficient substitute for metabolic supplementation.

In the in vivo context, a small pilot study utilized young Atm-KO mice at postnatal day 25 (P25) was conducted to first evaluate the safety and tolerability of 2% calcium α-KG (CaAKG), an alternate form of α-KG[109,110]. This pilot study encompassed a 60-day period during which 2% CaAKG was either included in soften food pellets or administered intraperitoneally (10 mg/kg/day) (Supplementary Fig. 16a). Initial results from the IGTT conducted at P25 indicated no significant differences in baseline blood glucose clearance abilities before the treatment (Supplementary Fig. 16b). However, by P85, following the completion of the treatment regimen, Atm-KO mice treated with the vehicle exhibited notably impaired blood glucose tolerance as compared to vehicle-treated Atm-WT mice (Supplementary Fig. 16b). Comparative assessments revealed that mice receiving 2% CaAKG through dietary incorporation over the same duration demonstrated more superior blood glucose control outcomes as compared to those received intraperitoneal injections (Supplementary Fig. 16b). Moreover, metabolic profiling conducted at P85 also revealed a more prominent improvement in the switching fuel dependence among the mice undergone dietary supplementation with 2% CaAKG, as evidenced by the bigger variance in RQ values between the light and dark cycle s (Supplementary Fig. 16c). Noteworthy, parameters such as food intake (Supplementary Fig. 16d) and locomotor activity levels remained relatively consistent across all experimental groups (Supplementary Fig. 16e). Based on the outcome of this pilot, subsequent experiments which extended the treatment duration to 185 days via the dietary supplementation approach was implemented (Fig. 6e). Analysis of survival rates revealed a significant reduction in mortality among Atm-KO mice receiving 2% CaAKG from the diet (Fig. 6f), an improvement likely associated with enhanced overall growth, as evidenced by increased body length (Fig. 6g) and greater preservation of lean skeletal muscle mass[111] (Fig. 6h). As an α-keto acid (RQ > 1), CaAKG overcame the challenges related to inefficient utilization of carbohydrates (i.e., absence of RQ switching to 0.9–1.0) (Fig. 6i) via serving as a biological fuel itself. This treatment concurrently alleviated the reliance on protein (RQ = 0.8), particularly when CaAKG was consumed along food at multiple times within a day to sustain this effect (Fig. 6i, Supplementary Fig. 16f-g). The consequent preservation in lean muscle mass, a largest tissue in the body, which is also sensitive to insulin, not only improved the overall glucose tolerance (Fig. 6j), fasting hyperinsulinemia status (Fig. 6k), but also the HOMA-IR status in comparison to the vehicle-treated Atm-KO counterparts (Fig. 6k). These endocrine profile improvements also significantly alleviated the degree of deterioration of coordination and motor function in these mice (Fig. 6l-m).

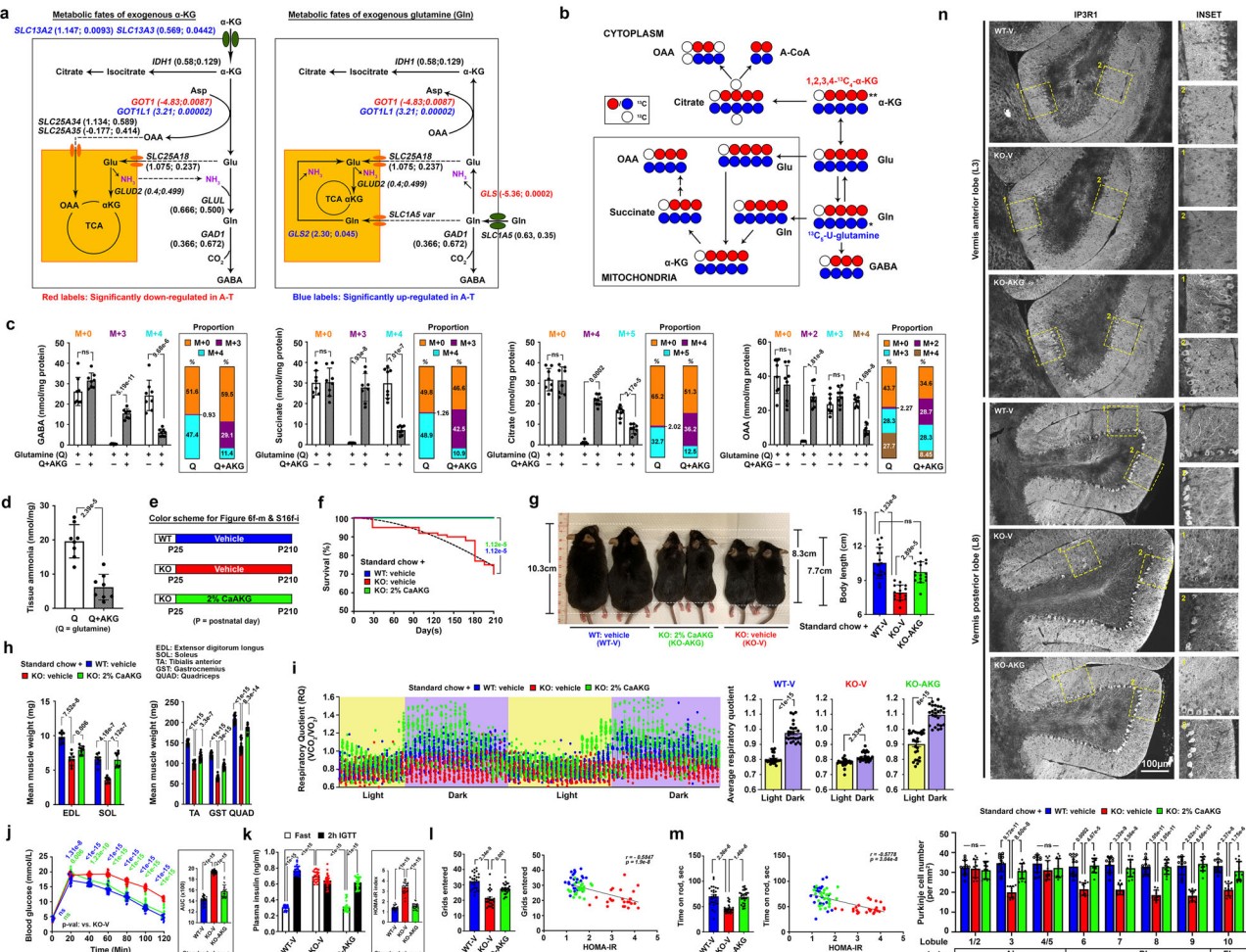

**Fig. 6 | α-Ketoglutarate supplementation mitigates endogenous glutamine wastage and insulin resistance in *Atm*-KO mice. a** Diagram illustrates the metabolic fates of α-KG or glutamine. Expression level changes in genes involved in the key metabolic reactions in human A-T cerebellar tissues (from GSE61019) compared to non-diseased controls. Values presented in brackets represent $Log_2$(fold change) followed by $Log_{10}$(adjusted p-values). **b** Schematics of $^{13}C_5$-U-glutamine and 1,2,3,4-$^{13}C_4$-α-KG carbon flow. **c** Isotopologue analysis of GABA, succinate, citrate, and OAA in acute ex vivo cerebellar cultures. The relative distributions of heavy isotopologues over the total amount of the respective metabolite are depicted ($n = 8$, two-tailed unpaired t-test for all except for "M + 4 citrate" where two-tailed Mann-Whitney test was used). **d** Tissue ammonia levels in acute ex vivo cerebellar cultures ($n = 8$, two-tailed unpaired t-test). **e** Schematics of a dietary regimen involving CaAKG supplementation (2% w/w) or vehicle alongside a standard laboratory diet. **f** Survival analysis throughout the treatment paradigm ($n = 20$, two-way ANOVA). **g** Representative mouse images depicting body length differences and their quantifications after the treatment paradigm ($n = 16$, one-way

ANOVA). **h** Summary of the average wet weights of major skeletal muscles ($n = 8$, one-way ANOVA). **i** Chronological variations in indirect calorimetry readings demonstrating alterations in respiratory quotient. Overview of changes between light and dark cycles ($n = 26$, two-tailed Mann-Whitney test for all except for WT-V where two-tailed unpaired t-test was used). **j** Outcomes of IGTT ($n = 26$, two-way ANOVA) and calculations of the AUCs ($n = 26$, one-way ANOVA). **k** Assessment of plasma insulin levels after overnight fast and at 2 h after the IGTT ($n = 26$, two-way ANOVA). Calculation of the HOMA-IR index shown on the right ($n = 26$, one-way ANOVA). **l, m** Overview of behavioral performances in the **l** open-field and **m** rotarod paradigms ($n = 26$, Kruskal-Wallis test). Correlations between the HOMA-IR index and performances ($n = 78$, Pearson's correlation). **n** Representative images of cerebellar vermis anterior (L3) and posterior (L8) lobes. Quantifications of IP3R1+ Purkinje cell numbers per mm² area is shown ($n = 12$, one-way ANOVA for all except for L6 where Kruskal-Wallis test was used). Values represent the mean ± s.d. N presents biological replicates. Source data are provided as a Source Data file.

The systematic enhancements in metabolic health that initiates from the peripheral, coupled with the direct impact of CaAKG on glutamine dependence, better preserved the number of insulin-sensing Purkinje cells (Fig. 6n, Supplementary Fig. 17) and their synaptic connections in the posterior and flocculonodular lobes of the vermal cerebellum (Supplementary Fig. 18).

## Discussion

Mammalian cells, unlike unicellular organisms, lack the intrinsic ability to independently uptake extracellular nutrients but instead depend on the presence of extracellular growth factor signals to regulate nutrient absorption and subsequent metabolic processes[112,113]. Insulin, a crucial signal rapidly initiated in response to elevated postprandial blood

sugar levels, plays a significant role in this regulatory network[114]. ATM is traditionally recognized as a pivotal component in the DNA damage response pathway[115], yet recent investigations from our laboratory and others have also underscored the critical involvement of the kinase in the central carbon metabolism[116]. This dual role suggests that ATM may serve as a significant regulator that instigates metabolic reprogramming in response to physiological stressors[67,117,118]. Our work here highlights the pivotal involvement of ATM in translating acute and chronic insulin hormonal signals into appropriate anabolic responses by initiating a specific phospho-protein network. Importantly, the linkage between insulin and ATM also facilitates an important crosstalk between the peripheral and central nervous system. Noteworthy downstream components of this network include PKM2 and HIF1α, key

regulators of aerobic glycolsis[62,114] that allows the diversion of glycolytic intermediates to facilitate the biosynthesis of various macromolecules[119].

The enzyme PKM assumes a pivotal role in the enzymatic conversion of phosphoenolpyruvate to pyruvic acid at the terminal step of glycolysis, thereby exercising critical regulation over the fate of glucose carbons[120]. Our investigations have unveiled that in individuals subjected to physiological duration of insulin stimulation, activation of ATM leads to instant phosphorylation and destabilization of PKM2 tetramers. Consequently, this protein quaternary structure loss results in diminished pyruvate kinase activity, prompting a diversion of glucose carbons towards biomass generation in the cytoplasm instead of committing to mitochondrial oxidation, a phenomenon also reported by others as well[121]. In the context of chronic insulin exposure, however, the sustained activation of ATM further influences the behavior of PKM2. In conjunction with a delayed accumulation and nuclear translocation of ATM-phosphorylated HIF1α, PKM2 co-activates HIF1α[96] to instigate a metabolic program favoring lipid biogenesis. While this phenomenon could be physiologically meaningful during mid-childhood[84] to early puberty[85–87] for speeding overall growth and brain maturation, it is more commonly associated with prediabetes during other life stages[88], contributing to body weight gain and excessive adipogenesis. Conversely, in the context of A-T, the deficiency of ATM resulted in PKM2 existing as a metabolically low-activity dimer in the cytoplasm, rather than the high-activity tetramer. This could be attributed to inherently depleted levels of its allosteric effectors, such as fructose-1,6-bisphosphate and serine, in these cells, which is also a side-effect of glycolytic insufficiency[122,123]. The lack of ATM also impedes both the immediate and protracted anabolic responses evoked by insulin. This deficit potentially accounts for the manifestation of inherent insulin insensitivity, stunt and even neurodegeneration in individuals afflicted with the disease.

Our findings show that HIF1α is a target of ATM resulting from prolonged, rather than short-term, insulin exposure. Therefore, under normal insulin response durations, we do not anticipate the stabilization and nuclear activities of HIF1α. However, when insulin response is extended—either as part of the normal growth and development process typically seen from mid-childhood to early puberty[84] or in pathological conditions like pre-diabetes—the stabilization and nuclear activities of HIF1α emerge. This supports a transcriptomic landscape that further facilitates aerobic glycolysis to stimulate lipogenesis, aiding in the biosynthesis of membrane-related phospholipids or triglycerides for storage purposes. At molecular level, HIF1α is a transcription factor being continuously produced but swiftly degraded under normoxic conditions unless subjected to post-translational modifications[124]. The degradation process involves hydroxylation and destabilization of HIF1α by α-KG-dependent prolyl hydroxylases (PHDs) or asparaginyl hydroxylases such as factor inhibiting HIF-1 (FIH-1)[125]. Therefore, it is possible as well the supplementation of α-KG could instead engender a heightened risk of further diminishing the residual HIF1α activity and exacerbate the metabolic defects in ATM deficiency. Noteworthy, in addition to α-KG, the activities of PHDs/FIH1 are also dependent on the co-existence of other co-factors, including Fe (II) ions, oxygen, and ascorbate[126]. Moreover, it is also salient to note the distinctive capacity of PHD3 to augment instead of impeding PKM2 function as a coactivator of HIF1α[96]. This knowledge collectively suggests the presence of a complex regulatory framework[127], which emphasize that the regulation of PHD activities within the intracellular environment is reliant on a diverse range of factors, rather than being exclusively governed by α-KG. Indeed, HIF1α levels and nuclear activities in tissue lysates harvested from the test animals after α-KG supplement failed to reveal any notable differences as compared to the vehicle-treated animals (Supplementary Fig. 16h-i). A comparable observation was also noted in fibroblasts from A-T patients following chronic α-KG supplement (Supplementary Fig. 16i). These indeed were likely attributable to the active integration of exogenous α-KG into the cellular metabolic network, rather than being channeled to activate the PHDs that support HIF1α degradation. This notion becomes evident when both isocitrate dehydrogenase-1 (IDH1) (Inhibitor: Compound 13/IDH1i)[128] and aspartate aminotransferase (GOT1) (Inhibitor: Compound 2c)[129]—the 2 pivotal cytosolic enzymes that facilitate α-KG participation into the metabolic network—were inhibited, which resulted in reductions of HIF1α nuclear activities (Supplementary Fig. 16i).

In A-T, the progressive degeneration of cerebellar Purkinje neurons and the consequent manifestation of ataxia delineate the most severe facets of the pathological condition. Purkinje cells are distinguished by their substantial physical dimensions, GABAergic properties, and intricate synaptic interconnections, which all these factors attribute to their elevated metabolic requisites relative to other neuronal populations within the brain[130]. Consequently, they are notably predisposed to perturbations in fuel metabolism[67]. Consistent with prior research[131], the examination of spatial and temporal expression levels of *ATM* and *INSR* throughout the entire human lifespan has revealed their heightened levels within the cerebellar cortex as compared to other brain regions, starting from the late embryonic stage (i.e., 200 days postnatally). Our current study underscores the insulin sensitivity of Purkinje neurons and highlights that ALDOC-positive Purkinje cells, primarily localized anatomically in the posterior and flocculonodular regions of the vermal cerebellum, exhibit increased vulnerability in the absence of ATM. Damage to these specific cerebellar areas is commonly associated with postural ataxia and a range of impairments in visual tracking, oculomotor control, spatial cognition, and language skills frequently reported in A-T patients[132]. ALDOC, a member of the fructose-bisphosphate aldolase enzyme family akin to the common ALDOA and ALDOB[133], is linked to enhanced glycolytic capacity and dependence upon ectopic overexpression, which is consistent with the transcriptome profiles of ALDOC-positive Purkinje cells. Furthermore, our findings suggest that chronic and sustained actions of insulin, a physiological endocrine change that occurs during mid-childhood and puberty[83], could induce ALDOC expression via HIF1α, suggesting it could be an important contributor to cerebellar maturation and development during these life stages. A malfunctioning insulin-dependent metabolic network in ALDOC-positive Purkinje cells may therefore play a role in amplifying their susceptibility. This vulnerability in selective cerebellar regions in part elucidates how the ataxic phenotype but not cognitive or psychiatric impairments, emerges as a dominant symptom stemming from cerebellar degeneration in A-T.

These brain-specific observations, alongside their association with the broader systemic changes, suggest that a metabolic-cantered strategy could potentially ameliorate or prevent certain symptoms, contingent upon the timing of intervention. Our investigation supports that the early-life introduction of dietary supplementation with α-KG as an innovative approach to enhance growth, bolster lean muscle mass, and uphold insulin sensitivity in the context of ATM deficiency. α-KG, situated at the nexus between carbon and nitrogen metabolism within the TCA cycle, targets the primary sites of metabolic dysregulation. Given the convergence of α-KG metabolism and that of glutamine, as well as the heightened presence of cell surface receptors for α-KG in A-T suggest a potential preference for α-KG in *Atm*-deficient cells and tissues. Moreover, the metabolism of α-KG also circumvents the requirement for glutamine deamination, a process that may result in neurotoxic ammonium accumulation in ATM-deficient cells. Instead, α-KG metabolism likely achieves nitrogen equilibrium, via acting as a principal amino-group receptor in the transamination reactions, the first catabolic step of most amino acids[134]. With reference to previous reports showing its capacity to enhance longevity in aged mice and mitigate systemic inflammation while devoid of documented adverse outcomes[109], α-KG emerges as a promising metabolite with potential therapeutic implications.

Prolonged α-KG supplementation has exhibited favorable outcomes on insulin sensitivity in obese rats through the augmentation of nitric oxide production in endothelial cells and the inhibition of hepatic gluconeogenesis via diverse pathways[110,135], which may explain how glucose intolerance is improved in A-T mice after α-KG supplementation. Future investigations should delve into whether these effects of α-KG can mitigate the peripheral metabolic complications linked to ATM deficiency and conceivably in other conditions characterized by similar metabolic perturbations.

Collectively, our findings emphasize the central importance of ATM activation as a previously unrecognized key regulator of both cellular catabolism and anabolism. ATM functions as a finely tuned switch that facilitates the communication between the endocrine insulin cues and the intricate intracellular metabolic framework. This work provides novel mechanistic understandings regarding how disturbances in homeostatic insulin action could result in either over- or under-active ATM responses, both of which may trigger persistent metabolic irregularities. Moreover, it also suggests that peripheral metabolic deficiencies is an overlooked contributing risk factor accelerating sustained neurodegenerative changes and ataxia in A-T. Our results also indicate that α-KG could be considered as a potential therapeutic target for managing human A-T and other disorders characterized by similar metabolic challenges. It is also noted that while the widespread metabolic perturbations inherent to the disease were consistently observed across various tissues sampled from the *Atm*-KO mouse model, it is crucial to recognize that such analyses may have limitations in fully elucidating possible metabolic differences that could have pre-existed among distinct cell subtypes (e.g., neurons versus glia) located within different regions of the tissue. Furthermore, the clarity on whether a dosage-dependent effect exists, particularly in patients with mutations leading to residual ATM proteins, remains uncertain[136]. Addressing these nuances will necessitate dedicated investigations in future studies.

## Methods
### Reagents, antibodies, open reading frame plasmids
Unless otherwise specified, all chemicals and reagents were purchased from Sigma-Aldrich. Details of antibodies, special reagents, assay kits, sequence-based reagents and analytical software, so as sequences for oligos and a list of unique reagents are provided in Supplementary Table 1. Unique reagents generated in this study will be made available upon reasonable request to the lead contact with a completed Institutional Materials Transfer Agreement.

### Animal maintenance, tissue harvest and primary cortical neuronal culture
**Animal maintenance and sample characteristics.** For all experiments, unless otherwise stated, no inclusion or exclusion criteria were applied other than genotype, HOMR-IR status and age. Animals were housed under a 12 h light/dark cycle at room temperature (22 ± 2 °C) and constant humidity levels at around 50–70%, with food and water provided ad libitum in a specific pathogen free (SPF) environment. Within these criteria, were randomly chosen among the available colonies of both sexes. B6;129S4-Atmtm1Bal/J (*Atm*$^{+/+}$, *Atm* +/− or *Atm*$^{-/-}$) mice were obtained from The Jackson Laboratory. They were maintained and bred in the Laboratory Animal Service Centre of the Chinese University of Hong Kong. Heterozygous B6;129S4-Atmtm1Bal/J were crossbred to produce *Atm*$^{-/-}$ (i.e., *Atm*-KO) for the study. The *Atm*$^{+/+}$ (i.e., *Atm*-WT) littermates were used as controls for all experiments. All in vivo physiological and behavioral experiments and ex vivo tissue culture experiments involved subjects from both sexes to ensure that any potential existence of sex-biased effect would be observed. An exception arose in the context of the study involving STZ injection. It is noteworthy that STZ, a pharmacological agent utilized to induce diabetes in laboratory mice, as documented in the literature[25], exhibits differential sensitivity in female mice due to potential protective effects conferred by estrogen[137]. Consequently, owing to the inherent complexities introduced by the estrous cycle in fertile female mice and the disparate STZ dosages required to elicit pancreatic β-cell toxicity (with female subjects typically necessitating higher STZ doses than their male counterparts), historical precedents have favored the utilization of male animals in STZ-induced diabetic mouse studies[138,139]. In alignment with this sex-biased experimental paradigm, we also adopted a similar approach for this assay to ensure consistency and comparability with existing literature and methodologies. For the in vitro primary cortical neuronal-based analysis, cells were mixed cultured from tissues of both sexes. All protocols were approved by the Animal Ethics Experimentation Committee (AEEC) at CUHK (and their care was in accordance with the institutional and Hong Kong guidelines) (Ref.: 20-012-GRF; 20-104-NSF; 23-064-NSF).

**Brain tissue harvesting.** Brain tissue isolation was performed in adult mice unless otherwise specified. Mice were first anesthetized by an intraperitoneal administration of 1.25% (vol/vol) avertin at a dosage of 30 ml per kg body weight. The heart of each mouse was then exposed, the left chamber was catheterized, and the right atrium was opened. Chilled physiological saline was perfused transcardially for 3 min to remove blood from the body. After perfusion, the cranial bones were opened, cerebral cortex and cerebellar tissues of the whole brain were collected. Other non-brain tissues, including multiple fat pads, hindlimb skeletal muscles and liver were also collected, snap-frozen in liquid nitrogen and stored at −80 °C before use.

**Cortical neuronal culture.** Embryonic day 16 embryos of both sexes were collected in ice-cold PBS-glucose, and the cortical lobes were dissected out. Meninges were removed under a dissection microscope, and the cortices were placed in 1× trypsin solution and placed in DMEM in 10% (vol/vol) fetal bovine serum to inactivate the trypsin, followed by transfer to neurobasal media supplemented with B-27, penicillin–streptomycin (1×) and L-glutamine (2 mM; GlutaMAX, Invitrogen). Tissue was triturated ten times through a wide-opening 5 ml pipette and allowed to settle to the bottom of a 15 ml conical tube. Cells that remained afloat in solution were retained, while pellets at the bottom were removed. Surviving cells were counted with trypan blue to identify dead cells and were plated on poly-L-lysine-coated (0.05 mg ml−1) glass coverslips. Unless otherwise specified, cells were plated in 24-well plates at 50,000 cells per well and allowed to mature for over 7–10 days in vitro before transfection or lentiviral transduction, or over DIV14 before drug treatment experiments. Samples were randomly chosen for different treatments.

### Human patient fibroblast culture
Lines of primary human skin fibroblasts from A-T patients and their respective controls were obtained from the NIA Aging Cell Repository of the Coriell Institute for Medical Research and maintained in DMEM/10% FBS.

### Lentivirus production and transduction
Human embryonic kidney 293FT cells (Invitrogen) were transfected using Lipofectamine 2000 (Invitrogen) with the expression of two helper plasmids: psPAX2 and pMD2.G. 10 μg transfer vector, 5 μg pMD2.G, and 5 μg psPAX2 of DNA were used per 10 cm plate. 48 h after transfection, the supernatants of four plates were pooled, centrifuged at 780 g for 5 min, filtered through a 0.45 μm pore size filter, and further centrifuged at 76,000 × g for 2 h. The resulting pellet was resuspended in 100 μl of PBS. Lentivirus titration was performed with a p24 ELISA (Clontech).

### IGTT and plasma insulin test

Mice were fasted for 16 h before intraperitoneal injection with 2 mg per kg body weight of glucose. Blood samples were taken from the tail vein at 0, 20, 40, 60, 80, 100 and 120 min. Glucose levels were measured using an Accu-Chek glucose meter (Roche Diagnostics). Plasma insulin levels were measured via ELISA assays using an ultrasensitive mouse insulin kit (Crystal Chem). The HOMA-IR index was calculated using the following formula: fasting plasma glucose (mmol l$^{-1}$) × fasting insulin (mIU l$^{-1}$)/22.5.

### Collection of CSF from the cisterna magna followed by CSF insulin test

CSF was collected from the cisterna magna of mice before and after the IGTT. Glass capillary tubes (Sutter Instrument; borosilicate glass B100-75-10) were prepared on a Sutter P-87 flaming micropipette puller (heat box set at 300 and the pressure index set at 330) and trimmed so that the inner diameters of the tapered tips were about 0.5 mm.

Mice were anesthetized by intraperitoneal administration of 1.25% (vol/vol) avertin at a dosage of 30 ml per kg body weight. For each individual mouse, the skin near the neck was first shaved, and then the body was placed prone on the stereotaxic instrument with direct contact to a heating pad. Once the head was secured with the head adaptors, the surgical site was swabbed with 10% povidone iodine, followed by 70% ethanol, and a sagittal incision of the skin was made inferior to the occiput. Under a dissection microscope, subcutaneous tissue and muscles (m. biventer cervicis and m. rectus capitis dorsalis major) were separated by blunt dissection with forceps. Then, the mouse was laid down so that the head was at a nearly 135° angle to the body. Under the dissection microscope, the dura mater of the cisterna was blotted dry with a sterile cotton swab and penetrated with a capillary tube to reach the cisterna magna. When a notable change in the resistance to the capillary tube occurred following insertion, the CSF was collected into the capillary tube.

The capillary tube was carefully removed, and CSF was ejected from the capillary tube with a syringe into a 1.5-ml tube, and frozen immediately on dry ice until further assays. After CSF sampling, muscles were realigned, and the skin was sutured. About 1 ml of 0.9% NaCl was injected to prevent dehydration. The mice were kept in a 37 °C incubator until full recovery. At 3 h after full recovery, mice were subjected to the IGTT as described above, and an additional round of CSF sampling was performed immediately after the IGTT. Once CSF samples were collected, mice were immediately sacrificed for collection of other tissue samples. Finally, CSF insulin levels were measured by ELISA using an ultrasensitive mouse insulin kit following the manufacturer's protocol (Crystal Chem).

### Behavioral tests

All mice were individually housed. All behavioral tests were performed during the dark phase of the circadian cycle between 19:00 and 23:00. All behavioral testing began by allowing the mice to habituate in the test rooms for 2 h before any tests. Experiments were performed blinded to the age and the HOMA-IR status. Using an overhead camera, experimental mice were also subjected to an open-field test and a rotarod test.

**Open-field test.** The open-field test was used for evaluation of anxiety and locomotion. Rodents show distinct aversions to large, brightly lit, open and unknown environments. It is assumed that they have been phylogenetically conditioned to see these types of environments as dangerous. In the experiments, mice were placed in the center of an open-field arena 50 cm (length) × 50 cm (width) × 38 cm (height) that was made from white high-density and non-porous plastic. Free and uninterrupted movement of the mouse was allowed for 5 min and movements were video taped. Locomotor activity was measured using the number of crossed grids, while exploratory activity was measured

using the number of rearing on the hind feet. The total travel distance and time spent in the outer versus the inner zone areas of the field were computed using a Smart 3.0 video tracking system (Panlab, Harvard Apparatus).

**Rotarod test.** To assess the acquisition of skilled behavior in mice, we first modified the standard rotarod test to emphasize the learning aspect of the test and minimize other factors. A rotarod machine with automatic timers and falling sensors (MK-660D, Muromachi-Kikai) was used. The mouse was placed on a drum. Before training sessions, the mice were habituated for 1 min immediately before the session. Animals were given two training trials (inter-trial interval: 2 h) with the rotarod adjusted to accelerate from 4 r.p.m. to 40 r.p.m. over a 5 min period each day. Latency to fall was measured. After a week of training, mice were tested using the rotarod adjusted to maintain a constant speed of 15 r.p.m. for the entire 5 min test period. The latency of the mice to fall off the rod or take one revolution was measured. Trials were repeated four times with inter-trial intervals of 30 min over a single day.

### In vivo drug treatment paradigms

**In vivo streptozotocin (STZ) ± glargine treatment paradigm.** At around 5 days prior to initiating the experiment, four independent cohorts of male littermate *Atm*-WT or *Atm*-KO mice were obtained from the breeding program as mentioned above and they were kept on standard laboratory diet (Teklad 2918) briefly during the post-weaning period until reaching 1 month old. A repeated (5x) low doses of STZ paradigm was then introduced. Before all STZ injections, food was refrained from animals for 4 h. At time prior to injection, STZ was freshly prepared in 50 mM sodium citrate buffer (pH 4.5) to a final concentration of 4 mg/ml. For each mouse, STZ working solution was injected intraperitoneally at 40 mg/kg (1.0 ml/100 g) in all experimental animals for 5 consecutive days. After each injection, mice were allowed to free access food and 10% sucrose water for recovery; except on day 6 and onwards, 10% sucrose water was replaced back as regular water. On Day 6, glargine (50 Unit/kg body weight, s.c. injection, once a day) was performed for the upcoming 14 days until P49[140,141]. Both IGTT and serum triglycerides were evaluated at time before STZ administration (P19) as well as on P49 after the entire drug treatment paradigm. Indirect calorimetry assessment was performed for 24 h after the dose of glargine injection on P49 prior tissue harvesting for other experiments.

**In vivo Glargine ± TEPP-46 treatment paradigm.** This part was performed in accordance with a previous published study with slight modifications[140,141]. Independent cohorts of littermate *Atm*-WT or *Atm*-KO mice of both sexes were obtained from the breeding program as mentioned above and they were kept on a standard laboratory diet (Teklad 2918) until 6 months old. At this age, mice were treated with either saline or a long- and slow-acting insulin reagent, glargine (50 Unit/kg body weight, s.c. injection, once a day), for 4 weeks. One glargine-treated WT group was co-administered with TEPP-46 (50 mg/kg body weight, i.p. injection, once a day) as well[93]. Mice were housed individually at the regular 20–22 °C, allowed to feed on the regular chow diet (Teklad 2918) *ad libitum*, and kept at 12 h light/dark cycles. Before the treatment starts, IGTT test on blood glucose level and plasma insulin measurement was performed after an overnight (16 h) fast with a glucose meter (Abbott Diabetes Care, Inc.) as to ensure similar baseline values in subjects within the *Atm*-WT or *Atm*-KO group. All mice were inspected daily to monitor their general health conditions, body weight change and food consumption.

**In vivo 2% (w/w) CaAKG feeding paradigm.** Independent cohorts of littermate *Atm*-WT or *Atm*-KO mice were obtained from the breeding program as mentioned above until weaning at post-natal day 25 (P25).

All mice were housed on a 12 h light/dark cycle and kept at 20-22°C. Treatment was started right after weaning.

CaAKG-treated animals were subjected to a 2-month (pilot study) or 6-month long 2% (w/w) CaAKG supplement on the regular mouse diet (Teklad 2918 Irradiated 18% protein and 6% fat diet) while the control groups were kept on the standard 2918 diet. Pure calcium 2-oxoglutarate (Carbosynth Company) was homogeneously mixed during manufacturing of the 2918 diet prior to irradiation and pelleting. The exact starting sample size could be found in the survival data. Mice were housed individually for food consumption recording and to prevent fighting and injuries. All lifespan and health span-related experiments were initiated at around 7 months of age in the 6-month long study. Baseline parameters such as body weight, sizes and glucose tolerance were evaluated, prior a fair partitioning of mice was done into different groups, i.e. for any given mouse in any given group, there are similar mice of both sexes in all other groups. This allows any outcome of the study to be more related to experiments or the treatment rather than the inherent property of a group. Mice were then inspected daily to monitor their general health conditions, body weight change and food consumption.

**In vivo intraperitoneal (i.p.) administration of 2% (w/w) CaAKG treatment paradigm.** The volume of injection was the lowest volume possible and not exceeded the current recommended guidelines [i.e., Needle Gauge: 25–27 g; Volume: <10 ml/kg (for a 25 g mouse, maximum volume would be 0.25 ml)]. Animals were gently removed from the cage and restrain appropriately in the head-down position. A daily injection (10 mg/kg/day) was performed at the lower right quadrant of the abdomen to avoid damage to the urinary bladder, cecum and other abdominal organs. Since the entire injection program had lasted for 60 days, injection was performed on alterative side of the lower abdomen differed from the side injected on the day before. Once after injection, all animals were placed back into their corresponding cages and observe for any complications (e.g. bleeding at injection site).

### Indirect calorimetry

Mice after undergoing through various in vivo treatment paradigms at various ages were individually placed in the registration chambers of the calorimetry system and allowed to adapt for 24 h prior the recording. After adaptation, the volume of carbon dioxide production ($VCO_2$) and the volume of oxygen consumption ($VO_2$) were recorded for at least 24 h using the Oxylet calorimeter system (Pan Lab/Harvard Instruments). The analysis of the respiratory exchange ratio (RER) was performed using the Metabolism software (Pan Lab/Harvard Instruments)[142].

### Acute cerebellar brain slice culture

Mouse cerebellar brain slices were prepared in coronal orientation as previously reported with slight modifications[143]. With freshly harvested brain tissues from mice at different ages (For details, please refer to different figure panels), tissues were trimmed and that the top part of the cerebellum was discarded. Both hemispheres were then glued on the chuck with the median side facing up. Slices were then cut (0.4 mm) using a Vibroslice (VT1000S, Leica) in an ice-cold solution containing 64 mM NaCl, 2.5 mM KCl, 1.25 mM $NaH_2PO_4$, 10 mM $MgSO_4$, 0.5 mM $CaCl_2$, 26 mM $NaHCO_3$, 10 mM glucose, and 120 mM sucrose. Slices were allowed to recover for 60 min prior initiation of acute drug treatments for whereas time points in artificial cerebrospinal fluid (aCSF) containing 126 mM NaCl, 2.5 mM KCl, 1.25 mM $NaH_2PO_4$, 2 mM $MgSO_4$, 2 mM $CaCl_2$, 26 mM $NaHCO_3$, and 10 mM glucose. All solutions were saturated with 95% $O_2$/5% $CO_2$ (volume/volume). For experiments where glargine and or TEPP-46 were treated to the slices, 100 nM glargine and 100 nM TEPP-46 was used (For details, please refer to different figure panels).

### Frozen sections preparation, Immunofluorescence histochemistry and Oil Red O staining

10 μm cryo-sections of frozen mouse brains or livers were used.

**Immunohistochemistry.** This was performed in accordance to a previously published study with slight modifications[144]. Samples were fixed with fresh 4% (wt/vol) paraformaldehyde for 10 min, washed and followed by permeabilization with 0.3% Triton-X100 in PBS for 10 min. After blocking with 5% (wt/vol) BSA in PBS for 1 h, primary antibodies were added and incubated overnight at 4 °C. The following day, coverslips were washed three times (10 min each) with PBS. After rinsing, secondary antibodies were applied for 1 h at room temperature followed by three additional washes with PBS. The coverslips were then inverted and mounted on glass slides with ProLong Gold Antifade Reagent (Life Technologies). All samples were examined and imaged on a upright fluorescent microscope (Nikon Ni-U) equipped with 10x and 20x objectives (CFI Plan Apochromat Lambda series) and Nikon DS-QI2 camera. Image acquisition and analysis were performed on the NIS-Elements software (Nikon). For visualizing the whole cerebellar structure, sequential images were stitched pairwise using the linear blending algorithm in Image J[145].

**Oil red O staining.** 0.7% Oil Red O stock solution (m/v in absolute isopropanol) was prepared by diluting the stock solution with distilled water to obtain the working solution. Cryo-sections of frozen mouse livers were washed with 60% isopropanol (v/v in PBS) and stained in Oil Red O working solution for 10 min. Subsequently, stained sections were washed with 60% isopropanol (v/v in PBS) to clear the background noise signals. All samples were examined and imaged on the brightfield function of the Nikon Ni-U upright fluorescent microscope equipped with 20x objective (CFI Plan Apochromat Lambda series) and Nikon DS-QI2 camera. The images were first bandpass-filtered to sharpen positive droplet-like signals for easier analysis, followed by automatic image thresholding and particle analysis in Image J.

### Subcellular protein fractionation, co-immunoprecipitation, SDS−PAGE and western blotting

For nuclear and cytoplasmic fractionation, freshly harvested brain tissues or cell platelets were used. A subcellular protein fractionation kit for tissues (ThermoFisher) was used following the manufacturer's protocol. For whole cell lysates, frozen or freshly harvested samples were homogenized in RIPA buffer (Millipore) with 1× complete protease inhibitor mixture (Roche) and 1× PhosSTOP phosphatase inhibitor mixture (Roche) on ice and centrifuged for 10 min at 18,400 × g to remove large debris. The protein concentration of the supernatant was determined using a Bradford assay (Bio-Rad). For co-immunoprecipitation, 1 mg of the total cell lysate was first incubated with control IgG (Santa Cruz Biotechnology) for 30 min, precleared with 50 μl of Dynabeads Protein G (Invitrogen) and then incubated with various antibodies according to the suggested dilutions on the product datasheets overnight at 4 °C. Beads bound with immune complexes were collected using DynaMag-2 (Life Technologies) and washed three times before elution in 90 μl of buffer containing 0.2 M glycine-HCl, pH 2.5, which was neutralized with 10 μl of neutralization buffer (1 M Tris-HCl, pH 9.0). The eluates were subjected to 9–15% SDS−PAGE and western blot analysis.

For western blotting, 100 μg of proteins derived of total cell lysates or subcellular fractionated lysates were separated by SDS−PAGE and transferred to polyvinylidene difluoride membranes. For the cross-linking experiments, cells were washed with ice-cold PBS three times and treated with 5 mM disuccinimidyl suberate (DSS, A39267, ThermoScientific, Waltham, MA) for 30 min at room temperature. The cross-linking reaction was stopped by adding the quenching solution (1 M Tris, PH 7.5) to the final concentration of 20 mM for 15 min[146]. Then, cell lysates were used for WB. Following

blocking, membranes were probed with various primary antibodies to determine different levels of protein expression. Immunoreactive antibody–antigen complexes were visualized using the SuperSignal West Femto Chemiluminescent Substrate (ThermoFisher). For the detection of dimerized ATM, proteins were denatured by the addition of 2% SDS without boiling and separated under nonreducing conditions at 4 °C[147]. Full scan blots can be found in Source Data file.

## Pyruvate kinase colorimetry assay

This assay was performed according to the manufacturer's protocol (Abcam). In brief, fresh tissues or cells were first extracted with 4 volumes of assay buffer, centrifuged to obtain clear extract and assayed immediately. For the standard curve, pyruvate standard was diluted to 1 nmol/μl and a serial dilution was prepared. For reaction mix preparation, the assay buffer, substrate mix, enzyme mix and OxiRed™ Probe were mixed and added to standard- or sample-containing wells, followed by colorimetry measurement at $OD_{570nm}$ for multiple time points. Pyruvate kinase (PK) activity was calculated as:

$$PK\ activity = (B \times Sample\ Dilution\ Factor)/[(T2 − T1) \times V]$$
$$(in\ nmol/min/ml = mU/mL)$$

Where:

B is the pyruvate amount from pyruvate standard curve (in nmol).
$T_1$ is the time of the first reading ($A_1$) (in min)
$T_2$ is the time of the second reading ($A_2$) (in min)
V is the sample volume added into the reaction well (in mL).

Unit definition: One unit of Pyruvate Kinase is the amount of enzyme that will transfer a phosphate group of PEP to ADP; yielding 1.0 μmol of pyruvate per minute at 25°C.

## HIF1α transcription factor DNA binding activity assay

This enzyme linked immunosorbent assay (ELISA) served as a replacement of the radioactive electrophoretic mobility shift assay (EMSA) and that the assay was performed by strictly adhering to the manufacturer's protocol (Abcam).

## Tissue ammonia colorimetry assay

Ammonia Assay Kit purchased from Abcam was used for the estimation of ammonia content in cerebellar tissues collected. The assay was performed by strictly adhering to the manufacturer's protocol.

## Triglyceride (TG) ELISA assay

Triglyceride (TG) ELISA Kit purchased from Sigma was used for the in vitro quantitative measurement of triglyceride concentrations in serum collected. The assay was performed by strictly adhering to the manufacturer's protocol.

## Live imaging of cytoplasmic redox status with Cyto-roGFP or CellROX dye

Cyto-roGFP obtained from Addgene (plasmid no. 49435) senses redox changes in a cell[148]. The Cyto-roGFP biosensor was transfected into cortical neurons on DIV 9, and 24 h after transfection, neurons were exposed to different treatment conditions in neurobasal culture medium. Cultures were then imaged for 24 h at hourly intervals in a 95% air/5% CO2-gassed incubator using a Leica TCS SP8 confocal laser scanning platform, equipped with Leica HyD hybrid detector and visualized through a HC PL APO CS2 63× (1.40 NA) oil-immersion objective. Image acquisition was controlled by LAS X. The redox-sensitive protein reporter has excitation maximum at 400 ± 15 nm and 484 ± 15 nm and an emission maximum at 525 ± 15 nm. The relative amplitudes of these peaks depend on the state of oxidation. With increased oxidation the 400 ± 15-nm excitation peak increases, while the 484 ± 15-nm peak decreases[149]. Importantly, the ratiometric nature of the analysis renders the results independent of the expression levels

of the plasmid in any one cell. Data were collected with the Leica Application Suite X Microscope Software. The fluorescence excitation ratios were obtained by dividing integrated intensities obtained from manually selected portions of the imaged regions of intact whole cells collected using 400 ± 15-nm and 480 ± 15 nm excitation filters after appropriate background correction. Background correction was performed by subtracting the intensity of a nearby cell-free region from the signal of the imaged cell. Alternatively cytoplasmic redox state was determined by the CellROX dye, which was used according to the manufacturer's protocol.

## Chromatin immunoprecipitation (ChIP)-PCR assay

A total of $5 \times 10^6$ A-T patient fibroblasts (Fig. 3k) or acute cerebellar slice culture treated with or without 100 nM glargine for various time courses were cross-linked with 3.7% formaldehyde (Sigma) at room temperature for 10 min. Samples were incubated with 0.125 M glycine to terminate cross-linking, washed twice with PBS and lysed with SDS nuclear lysis buffer (1% SDS, 10 mM EDTA and 50 mM Tris-HCl, pH 8.1) for 10 min on ice. Sonicated lysates were diluted in ChIP dilution buffer (0.01% SDS, 1.1% Triton X-100, 1.2 mM EDTA, 167 mM NaCl and 16.7 mM Tris-HCl, pH 8.1) and incubated with 10 μg of rabbit IgG (Santa Cruz Biotechnology) and Dynabeads Protein G (Invitrogen) overnight at 4 °C with gentle shaking. The cleared supernatants were mixed with either 20 μl of c-MYC antibody for Fig. 3k, anti-HIF1α antibody for Fig. 4r or with pre-immune rabbit IgG as negative controls for both overnight at 4 °C. Antibody–protein–DNA complexes were co-precipitated with Dynabeads Protein G. Protein–DNA conjugates were eluted from the bead complexes with elution buffer (100 mM NaHCO3 and 1% SDS) for 30 min. Cross-links were reversed in 5 M NaCl. RNA and protein were removed by incubation first with 10 μg DNase-free RNase-A at 37 °C for 1 h, and then with 20 μg proteinase K (Sigma) at 50 °C for 4 h. DNA was recovered by phenol–chloroform extraction and ethanol precipitation. A DNA fragment encompassing the c-MYC binding region of the human *GLS2* promoter or HIF1α binding region of mouse *Aldoc* promotor was amplified using 35 cycles of PCR at 94 °C for 30 s, 55 °C for 30 s and 72 °C for 30 s. All amplified products were resolved on a 1.5% agarose gel. Primers are listed in Supplementary Table 1. Full scan blots can be found in Source Data file.

## RNA immunoprecipitation (RIP)

RIP was performed using the Magna RIP™ RNA-binding protein immunoprecipitation kit (Millipore) following strictly the manufacturer's guidelines. Human patient fibroblasts were treated with ice-cold PBS and Pierce™ IP lysis buffer (ThermoFisher) supplemented with RNase inhibitors and a protease inhibitor cocktail. The supernatants were incubated with the anti-PKM2 antibody or control rabbit IgG with rotation at 4 °C overnight before further incubation with protein G bead for another night at the same temperature. After that protein-RNA precipitates were enriched and bounded RNAs were extracted by the proteinase K-chloroform method. Enrichment of the internal ribosome entry site region of the *MYC* mRNA was performed by reverse transcription followed by RT-PCR.

## qPCR with reverse transcription analysis

Total cellular RNA was purified from cerebellar tissues or cultured cells using a RNeasy mini kit (Qiagen) following the manufacturer's protocol. For real-time qPCR, RNA was reverse-transcribed using a High-Capacity cDNA Reverse Transcription kit (Applied Biosystems) according to the manufacturer's instructions. The resulting complementary DNA was analysed by qPCR with reverse transcription using SYBR Green PCR Master Mix (Applied Biosystems). All reactions were run on a LightCycler 480 Instrument II (Roche Diagnostics) with a 15 min hot start at 95 °C followed by 40 cycles of a 3-step thermocycling program: denaturation step of 15 s at 94 °C; annealing step of 30 s at 55 °C; and extension step of 30 s at 70 °C. Melting curve analysis

was performed at the end of every run to ensure that a single PCR product of the expected melting temperature was produced at a given well. A total of seven to nine biological replicates with four technical replicates were performed for each treatment group. Data were analysed using the comparative Ct method (ΔΔCt method). The primers (5′ to 3′) used were designed on Primer-Blast and are listed in Supplementary Table 1.

### Conventional cell line and patient fibroblast culture, transfection and live imaging

Lines of primary human skin fibroblasts from A-T patients and their respective controls were obtained from the NIA Aging Cell Repository of the Coriell Institute for Medical Research and maintained in DMEM/ 10% FBS. Immortalized mouse hippocampal cells (HT22 cell line, ThermoFisher Scientific) were cultured in DMEM/10% (vol/vol) fetal bovine serum/penicillin–streptomycin medium (Gibco). Primary cortical neurons were isolated from E16 embryos of C57BL/6 J or Atm + /− (Atm-KO) pregnant mice and cultured as described above. DNA constructs were transfected with Lipofectamine 2000 or Lipofectamine LTX with Plus Reagent into HT22 cells and primary cortical neuronal cultures, respectively. Following the manufacturer's protocol, 6 h after transfection, cells were refreshed with culture medium and further incubated for 48–72 h to allow recovery and ectopic expression prior harvesting or any drug treatments.

For primary cortical neurons that were transfected with either pEGFP-C1-PKM2 (A gift from Axel Ullrich; Addgene plasmid #64698) or pg-HIF-1alpha-EGFP (A gift from Violaine Sée; Addgene plasmid # 87204); time lapse confocal microscopy was performed afterwards in various treatment conditions, as indicated in the text. Cells were incubated on the microscope state at 37 °C, 5% CO$_2$ or 20% O$_2$ and observed by confocal microscope using a Leica Sp8 with a 63 × 1.3 NA oil immersion objective. Excitation of EGFP was performed using an argon ion laser at 488 nm. Emitted light was detected through a 505–550 nm bandpass filter from a 545 nm dichroic mirror. Excitation of the empty AAV-Synapsin-mCherry-C1-WPRE (A gift from Michael Courtney; Addgene plasmid # 159956) used as a control was performed using a green helium-neon laser (543 nm) and detected through both a 545-nm dichroic mirror and a 560-nm long pass filter. Data capture was carried out with the LAS X Life Science Microscope software. These experiments were performed three times and ~45 cells were analyzed for each construct.

### Label-free phosphoproteome analysis

**Protein extraction, digestion and clean up.** Following the specified treatment protocols and duration, cerebellar tissues were promptly collected and flash-frozen in liquid nitrogen. All specimens were subsequently stored at −80 °C prior to cryo-pulverization. The EasyPep Mini MS Sample Prep Kit by ThermoFisher was utilized for the ensuing protein extraction processes, albeit with minor adaptations. To elucidate, 5 mg of tissue samples underwent homogenization in 500 μl of lysis solution containing 1 μl of universal nucleases using Dounce homogenizers. This was followed by centrifugation at 16,000 g for 10 min, resulting in the retrieval of the supernatant for protein quantification. Subsequently, 500 μg of protein lysates were adjusted to 500 μl with lysis solution and then combined with 250 μl of reduction solution and 250 μl of alkylation solution at 95 °C for 30 min. Upon the samples' cooling, 250 μl of reconstituted enzyme solution, comprising trypsin/lysyl endopeptidase (LysC), was introduced to the reduced and alkylated protein samples. The mixture was then subjected to overnight incubation with agitation at 300 r.p.m. at 37 °C to facilitate protein digestion. Upon completion of the incubation period, 250 μl of Digestion Stop Solution was added to the sample, followed by gentle mixing. After the completion of the digestion process, 300 μl of the digested proteins were subsequently introduced to a dry Peptide Clean-up column, followed by centrifugation at 1500 g for 2 min, with

the subsequent discarding of the flowthrough. This procedure was iterated until all the lysates had been loaded onto the column. Subsequently, the column underwent a washing process involving washing buffers A and B to remove impurities, before the enriched proteins were eluted. The eluted peptides were then subjected to drying using a SpeedVac system (ThermoScientific) and subsequently stored at -80 °C until further processing, or for phosphopeptide enrichment.

**Phosphopeptide enrichment.** Phosphopeptides were extracted from the digested peptides using the High-Select TiO2 Phosphopeptide Enrichment Kit (ThermoFisher), with slight modifications. Initially, the lyophilized peptides were fully dissolved in 150 μl of binding/equilibration buffer (pH<3). The suspended peptide solution was then transferred to a TiO2 spin tip, with a maximum of 150 μl loaded per transfer, and centrifuged at 1000 g for 5 min. This loading process was repeated until all the suspended peptides had been transferred to the spin tip. Following this, the spin tip underwent a series of washing steps, involving the binding/equilibration buffer, wash buffer, and LC-MS grade water, to eliminate impurities and non-phosphorylated peptides, thereby concentrating the phosphopeptides for subsequent analyses. Subsequently, the phosphorylated peptides were eluted, speed dried using the SpeedVac system (ThermoScientific) and then stored at −80 °C until further investigations were conducted.

**Liquid chromatography-mass spectrometry analysis.** Both the complete tryptic-digested peptides and the enriched phosphopeptides were introduced into a Vanquish Neo LC system (ThermoFisher) for the purpose of peptide separation. The peptides underwent separation employing a trap-and-elute methodology utilizing a 20 mm × 75 μm trap column (ThermoFisher) and an Aurura Ultimate XT 25 cm × 75 μm C18 column (Ionopticks). The temperature of the separation column was maintained at 50 °C using an XT compatible heater controller (Ionopticks, Australia). Solvent A and B were constituted of 0.1% formic acid in milli-Q water and 0.1% formic acid in acetonitrile, respectively. The gradient settings, with a flow rate of 300 nl/min, were programmed as follows: a linear increase from 2% B to 6% B over 0–2 min, followed by a gradient from 6% B to 30% B within 2–77 min, and subsequently from 30% B to 90% B during 79–82 min, maintaining this composition until 87 min. The gradient then decreased from 90% B to 2% B at 87.1 min and was sustained until 90 min.

Subsequent to elution, peptides were directly introduced into an Orbitrap Fusion Lumos Mass Spectrometer (ThermoFisher), where mass spectra data were obtained employing a data-dependent acquisition (DDA) approach within a scan range of 400–1500 m/z. The spray voltage was maintained at 2 kV, accompanied by an ion transfer tube temperature of 300 °C. For MS1 analysis, the resolution was set at 60,000, with an AGC target of 4.00E5 and a maximum injection time of 50 ms. Following this, MS2 spectra were acquired utilizing a 3 s cycling time and a 40 s dynamic exclusion period. The orbitrap resolution for MS2 was set at 15,000, with HCD collision energy of 30%, an AGC target of 5.00E4, and a maximum injection time of 22 ms.

**MaxQuant quantification.** After the experimental procedures, mass spectrometric data analysis was performed using MaxQuant, a specialized quantitative proteomics software tailored for the analysis of extensive datasets obtained through high-resolution MS, following the previously published workflow and procedures[150]. This software accommodates various labeling methodologies and supports label-free quantification techniques. MaxQuant, freely accessible for download from the specified source, incorporates the andromeda search engine for peptide identification, quantification, and the viewer application for the examination of raw data[150]. The raw data files were analysed using MaxQuant to obtain phosphosite identifications and their respective label-free quantification values using the following

parameters. Raw data were analyzed using MaxQuant's (version 2.6.6.0) (Cox & Mann, 2008) Andromeda search engine in reversed decoy mode based on a Mus musculus reference proteome (Uniprot-FASTA, UP000000589, downloaded November 2024) with a false discovery rate (FDR) of 0.01 at both peptide and protein levels, and identification of post-translational modifications were specified to identify site-specific protein phosphorylation in the corresponding samples. Digestion parameters were set to specific digestion with trypsin with a maximum number of 2 missed cleavage sites and a minimum peptide length of 7. Oxidation of methionine, amino-terminal acetylation and phosphorylation sites at serine, threonine and tyrosine (STY) were set as variable modifications; carbamido-methylation of cysteine was set as fixed modification, with a maximum number of 5 modifications per peptide. The tolerance window was set to 20 ppm (first search) and to 4.5 ppm (main search). Label-free quantification was set to a minimum ratio count of 2, re-quantification and match-between-runs was selected and 4 biological replicates per condition were analyzed. Resulting raw output protein group files generated from MaxQuant were processed using Phospho-Analyst to perform differential expression analysis to visualize the results preprocessed with MaxQuant[151]. The original data is deposited on the PRIDE Archive under the project accession number PXD062018.

**Identification of significant phosphosites using phospho-analyst.** Subsequently, MaxQuant output tables, which contain information about quantification of proteins and PTMs, were used as input to Phospho-Analyst online platform for quality control and post-hoc analyses (https://analyst-suites.org/apps/phospho-analyst/)[151]. For the phosphosite report, the input data were normalized based on the assumption that the majority of phosphosites do not change between the different conditions. Contaminant phosphosites, reverse sequences and phosphosites identified "only by site" were filtered out. In addition, phosphosites with localization probability < 0.75 have been removed as well. The phosphosite data was converted to log2 scale, samples were grouped by conditions and missing values were imputed using "Miss not At Random" (MNAR) method, which uses random draws from a left-shifted Gaussian distribution of 1.8 standard deviation apart with a width of 0.3. Protein-wise linear models combined with empirical Bayes statistics were used for the differential expression analyses. The limma package from R Bioconductor via the software was used to generate a list of differentially expressed phosphosites for each pair-wise comparison. A cutoff of adjusted p-value of 0.05 (Benjamini-Hochberg method) along with a |log2 fold change| of 1 has been applied to determine significantly regulated phosphosites in each pairwise comparison. Visualization of phosphorylation motif was performed on the pLogo platform (https://plogo.uconn.edu/)[152].

For the proteinGroup report, the data from MaxQuant were normalized based on the assumption that the majority of proteins do not change between different conditions. Contaminant proteins, reverse sequences and proteins identified "only by site" were filtered out. In addition, proteins that have been only identified by a single peptide and proteins not identified/quantified consistently in the same condition have been removed as well. The LFQ data was converted to log2 scale, samples were grouped by conditions and missing values were imputed using the "Miss not At Random" (MNAR) method, as described above. Protein-wise linear models combined with empirical Bayes statistics were used for the differential expression analyses. The limma package from R Bioconductor via the software was used to generate a list of differentially expressed proteins for each pair-wise comparison. A cutoff of adjusted p-value of 0.05 (Benjamini-Hochberg method) along with a |log2 fold change| of 1 has been applied to determine significantly regulated phosphosites in each pairwise comparison.

**In silico docking.** The mouse heterodimeric HIF1α: ARNT (HIF1β) complex structure was obtained from the Protein Data Bank (PDB: 4ZPR)[153]. According to that, the human HIF1α and HIF1β structures were modelled with SWISS-MODEL algorithm[154]. For the protein structures of human apoenzyme PKM2 (PDB: 3BJT)[155] and importin-α3 (PDB: 6BVZ)[156], they were directly extracted from the Protein Data Bank accordingly. To simulate the phosphorylation sites predicted form the SILAC experiment, targeted phosphate groups were added to the HIF1α and PKM2 using PyTMs plugin of the PyMOL software[157]. With the structures ready, HADDOCK2.4 was used to simulate the strengths of protein-protein interactions and distance between interacting residues[158].

To achieve a more definitive and accurate sampling of protein-protein interaction but not a random selection on binding structures based on conformation energies that fails to fit previous experimental findings, the following docking parameters were set prior the docking simulation. To drive the docking, ambiguous interaction restraints (i.e. the active site residues) derived from previous protein-protein inter-action analysis of experimentally resolved structures were referenced. Residues were regarded as accessible if they fulfil the minimum 15% of relative solvent accessibility, whereas passive residues were defined if they are located within the 6.50 Å radius of active site residues. On the other hand, non-polar hydrogens were eliminated during the docking processes; and that the semi-flexible regions were automatically defined.

During the simulation, a three-step docking procedure using the software default setting was conducted, characterized by an initial rigid docking, followed by semi-flexible docking and lastly a structure refinement step in the presence of water molecules. The resulting HADDOCK scores and predicted binding energies of the top three clusters were considered and compared. Cut-off distances of proton-acceptor (hydrogen bonds) and carbon-carbon (hydrophobic contacts) were set at 2.5 Å and 3.9 Å, respectively. Interactions between amino acid residues located on the docking proteins were subsequently analysed by the Protein-Ligand Interaction Profiler (PLIP)[159]. The representative images were created on the PyMOL software.

## In silico gene promotor analysis and targeted transcription factor binding site identification

The bindings of c-MYC and HIF1α transcription factors respectively on *GLS2* and *ALDOC* genes were first analysed on the ChIP-Atlas and the gross genome binding regions were visualized on the IGV genome browser[160,161]. Upon such confirmation, detailed mapping of the binding sites was performed with immediate promoter sequences extracted from the NCBI Genome Data Viewer using the using the JASPAR database, with a score threshold > 85 %. Conserved transcription factor binding sites found on the human and mouse homolog genes promotor were extracted for the downstream chromatin immunoprecipitation (ChIP)-PCR experiment, as described above.

## Human microarray data mining

A human expression array dataset (Gene Expression Omnibus accession no. GSE61019), comparing samples of A-T and control cerebella, was analysed with GEO2R[162]. Raw data from each of the samples were extracted. All transcripts that reflected significant changes (adjusted $P \leq 0.05$) by the GEO2R analyser were further extracted and analysed using GSEA with the KEGG pathway groupings of genes. Lipid metabolism-related pathway which were significant (Nom $p$-value < 0.05) and enriched in normal controls were selected and shown in Supplementary Fig. 15.

## Single-nucleus RNA sequencing data mining and analysis

The original and raw single-nuclei RNA sequencing data of human cerebellar cortex was downloaded from the Single Cell Portal (SCP1300)[76]. Similarly, the original and raw single-nuclei RNA

sequencing data of mouse cerebellar cortex were downloaded from the GEO database (GSE165371).

DropletUtils was then used as the first filter to remove any cells with extremely low expression in each of the samples, with the false positive rate set as 0.01. Filtered cells was then merged into as an integrated dataset which was then analysed by the Seurat 4.0.5. Genes expressed in at least three nuclei were retained, and those outlier nuclei with a high ratio of mitochondrial encoded transcripts (i.e., >20%, < 200 UMI) relative to the total RNA, and those potential doublets ( > 6000 UMI) were discarded in the subsequent analysis. Top 30 dimensions and resolution with 0.5 as input were used to build the UMAP graphs, tSNE graphs and cell clusters. Major cerebellar cell types were annotated according to the published study[163]. For the identification of differentially expressed genes of the targeted Purkinje cell clusters located at the anterior and posterior lobes was calculated using the FindMarkers function in the Seurat 4.0.5 package with the cut off value set at $p$-value < 0.01 and min.pct = 0.25. Those enriched in target clusters were then clustered for pathway analysis using the EnrichR software (https://maayanlab.cloud/Enrichr/).

## Mouse cerebellar tissue bulk RNA sequencing

Frozen cerebellar tissues from the transgenic mice were harvested and sent to the Novogene for total RNA extraction and RNA sequencing with the Illumina HiSeq X Ten platform. FASTQ files obtained were subsequently analysed by the FastQC (V. 0.11.9) of Babraham Bioinformatics, serving as a quality control assessment tool for these data. After assuring the quality of the data files, genome indexes were generated using the STAR software using. These indexes were generated with the mouse annotation and mouse genome assembly provided by the Gencode Release 28 (GRCm39, released in May 2021). The cleaned dataset was then aligned with the genome indexes using the STAR alignment software. FeatureCounts was then used to quantify the sequencing data. Data extraction, matrix construction and differential gene expression analysis were then performed using the DESeq2 package on R. Pathway analysis was performed with Enrich R. The original FASTQ files were deposited on GEO Omnibus (GSE222655).

## Untargeted metabolome analysis by capillary electrophoresis time-of-flight mass spectrometry (CE-TOFMS) and liquid chromatography (LC)-TOFMS

Metabolome analyses were performed in mouse cerebellar tissues using CE-TOFMS for both cationic and anionic metabolites on the basis of service purchased from Human Metabolome Technologies' standard library. Samples were sent to HMT where their weights were first measured. For CE-TOFMS preparation, samples were mixed with 1500 μl of 50% acetonitrile in water (v/v) containing internal standards (10 μM) and homogenized by a homogenizer (1500 r.p.m., 120 sec x 1 times). The supernatant (400 μl) was then filtrated through 5-kDa cut off filter (ULTRAFREE-MC-PLHCC, HMT) to remove macromolecules. The filtrate was centrifugally concentrated and resuspended in 50 μl of ultrapure water immediately before measurement. Whereas for LC-TOFMS preparation, weighted samples were mixed with 300 μl of 1% formic acid in acetonitrile (v/v) containing internal standards (10 μM) and homogenized by a homogenizer (1500 r.p.m., 120 s × 2 times). The mixture was yet again homogenized after adding 100 μl of Milli-Q water and then centrifuged (2300 × g, 4 °C, 5 min). After the supernatant was collected, 300 μl of 1 % formic acid in acetonitrile (v/v) and 100 μl of MilliQ-water were added to the precipitate. The homogenization and centrifugation were performed as described above, and the supernatant was mixed with previously collected one. The mixed supernatant was filtrated through 3-kDA cut-off filter (NANOCEP 3 K OMEGA, PALL Corporation, Michigan, USA) to remove proteins and further filtrated through column (Hybrid SPE phospholipid 55261-U, Supelco, Bellefonte, PA, USA) to remove phospholipids. The filtrate

was desiccated and resuspended in 200 μl of 50% isopropanol in Milli-Q water (v/v) immediately before the measurement.

**CE-TOFMS measurement.** The compounds were measured in the Cation and Anion modes of CE-TOFMS based metabolome analysis in the following conditions. Samples were diluted in 2 folds for measurement, to improve analysis qualities of the CE-MS analysis.

The parameters for the detection of cationic metabolites were established as follows: An Agilent CE-TOFMS system (Agilent Technologies Inc., Machine No. 12) equipped with a fused silica capillary (inner diameter: 50 μm, length: 80 cm) was utilized. For the analytical conditions, the Cation Buffer Solution (product number: H3301-1001) was employed as both the running and rinse buffer. Sample injection was performed under a pressure of 50 mbar for 5 s. The capillary electrophoresis (CE) voltage was set to +30 kV. In the mass spectrometry phase, ionization was conducted using the ESI (electrospray ionization) positive mode, with a capillary voltage of 4000 V. The mass spectrometer was operated with a scan range of m/z 50−1000. The sheath liquid used was HMT Sheath Liquid (product number: H3301-1020).

The parameters for the detection of anionic metabolites were established as follows: An Agilent CE-TOFMS system (Agilent Technologies Inc., Machine No. 5) equipped with a fused silica capillary (inner diameter: 50 μm, length: 80 cm) was utilized. For the analytical conditions, the Anion Buffer Solution (product number: I3302-1023) was employed as both the running and rinse buffer. Sample injection was performed under a pressure of 50 mbar for 22 s. The capillary electrophoresis (CE) voltage was set to +30 kV. In the mass spectrometry phase, ionization was conducted using the ESI (electrospray ionization) negative mode, with a capillary voltage of 3500 V. The mass spectrometer was operated with a scan range of m/z 50−1000. The sheath liquid used was HMT Sheath Liquid (product number: H3301-1020).

**LC-TOFMS measurement.** The compounds were analyzed in both Positive and Negative modes using LC-TOFMS-based metabolome analysis under optimized conditions. To enhance the quality of the CE-MS analysis, samples were diluted 1-fold prior to measurement.

The analysis for cationic metabolites in Positive mode was conducted using an Agilent 1200 series RRLC system SL (Agilent Technologies Inc.) coupled with an ODS column (2 × 50 mm, 2 μm) and an Agilent LS/MSD TOF mass spectrometry system (Machine No. 9). The column temperature was maintained at 40 °C, and the mobile phase consisted of H2O with 0.1% formic acid (Mobile Phase A) and a mixture of isopropanol, acetonitrile, and water (65:30:5) containing 0.1% formic acid and 2 mM ammonium formate (Mobile Phase B). The flow rate was set at 0.3 mL/min, with a total run time of 20 min followed by a post-run time of 7.5 min. The gradient conditions included Mobile Phase B starting at 1% for the first 0.5 min, increasing linearly to 100% over 13.5 min, and held at 100% for the remaining 6.5 min. Mass spectrometry was performed using ESI in Positive ionization mode, with a nebulizer pressure of 40 psi, a dry gas flow rate of 10 L/min, and a dry gas temperature of 350 °C. The capillary voltage was set to 4000 V, and the scan range was m/z 100−1700. A sample injection volume of 1 μL was used for the analysis.

The analysis of anionic metabolites in Negative mode was performed using an Agilent 1200 series RRLC system SL (Agilent Technologies Inc.) coupled with an ODS column (2 × 50 mm, 2 μm) and an Agilent LS/MSD TOF mass spectrometry system (Machine No. 9). The column temperature was maintained at 40 °C, with Mobile Phase A consisting of H2O containing 0.1% formic acid and Mobile Phase B comprising a mixture of isopropanol, acetonitrile, and water (65:30:5) with 0.1% formic acid and 2 mM ammonium formate. The flow rate was set at 0.3 mL/min, with a total run time of 20 min and a post-run time of 7.5 min. The gradient conditions began with Mobile Phase B at 1% for

the first 0.5 min, followed by a linear increase to 100% over 13.5 min, and held at 100% for the remaining 6.5 min. Mass spectrometry was conducted in ESI Negative ionization mode, with a nebulizer pressure of 40 psi, dry gas flow rate of 10 L/min, and dry gas temperature of 350 °C. The capillary voltage was set at 3500 V, and the scan range covered m/z 100–1700. A sample injection volume of 1 μL was used for the analysis.

Peaks detected in both CE-TOFMS and LC-TOFMS were extracted using automatic integration software (MasterHands ver. 2.17.1.11 developed at Keio University) in order to obtain peak information including m/z, migration time (MT) in CE, retention time (RT) in LC, and peak area. The peak area was then converted to relative peak area by the following equation. the limit of peak detection was determined by a signal-noise ratio of at least 3.

#### Relative peak area = metabolite peak area/internal standard peak area x sample amount

Putative metabolites were then assigned from HMT's standard library and Known-Unknown peak library on the basis of m/z and MT or RT. The tolerance was ± 0.5 min in MT and ± 0.3 min in RT, ± 10 ppm (CE-TOFMS) and ± 25 ppm (LC-TOFMS) in m/z. If several peaks were assigned the same candidate, the candidate was given the branch number.

#### Mass error (ppm) = (measured value − theoretical value)/measured value x 10$^6$

Subsequent absolute quantification was performed in target metabolites. All the metabolite concentrations were calculated by normalizing the peak area of each metabolite with respect to the area of the internal standard and by using standard curves, which were obtained by single-point (100 μM or 50 μM) calibrations. Significantly changed metabolites (with Log$_2$FC ± 0.5; $P < 0.05$) were enriched and analyzed by the Metabolite Set Enrichment Analysis (MSEA) or the Joint Pathway Analysis module (with KEGG metabolic gene expression data) on MetaboAnalyst (https://www.metaboanalyst.ca/MetaboAnalyst/ModuleView.xhtml).

#### Lipid extraction and selected lipidomics analysis

The following procedures were performed based on a published protocol with slight modification[164]. Lipid extraction procedures were performed as reported previously with minor modifications[165]. Briefly, 10 mg of frozen cerebellar tissues was deactivated by 900 μL mixture of chloroform: methanol (1:2) with 10% deionized water. Then the mixture was homogenized by OMNI Bead Ruptor (OMNI, USA). After homologenization at 4 °C for 1 h, 400 μL deionized water and 300 μL chloroform were added to the mixture, which the mixture was vortexed for 10 min followed by centrifugation at 9021 × g, 4 °C for 15 min. The resulting under layer organic phase material was then transferred to a sterile tube. Any unextracted residuals were again subjected to an repeated extraction step with another 500 μL chloroform. By combining the products extracted from the two extraction steps, the samples were then frozen dried in a SpeedVac (Genevac). Dried samples were stored at −80 °C until further analysis.

Lipidomics approach was conducted by Exion UPLC-QTRAP 6500PLUS (Sxiex) via electrospray ionization (ESI) iron source under conditions as follows: curtain gas = 20, ion spray voltage = 5500 V, temperature = 400 °C, ion source gas 1 = 35, ion source gas 2 = 35. Specifically, Triacylglycerol (TAG) and diacylglycerol (DAG) were detected using a reverse-phase high-performance liquid chromatography (HPLC)-electrospray ionization- mass spectrometry (MS) with Phenomenex Kinetex 2.6 μm C18 column (inner diameter 4.6 × 100 mm). The glyceride lipids were separated by isocratic elution mode with chloroform, methanol, 0.1 mol/L ammonium acetate (100: 100: 4) at a flow rate of 160 μL/min for 20 min. Quantitative analyses of TAG and DAG were conducted by applying the Neutral Loss MS/MS technology, referencing the internal standards. Similarly, free

cholesterol, sterols and corresponding esters were analyzed using HPLC tandem MS analysis through atmospheric pressure chemical ionization (APCI) mode and were quantified by referencing internal standards.

Lipids with different polarities including phospholipids and sphingolipids, as well as free fatty acids were separated by NP-HPLC according to previously reported[166] using Phenomenex Luna 3 μm silica column (inner diameter 150 × 2.0 mm). HPLC condition was set as follows: mobile phase A (chloroform: methanol: ammonium hydroxide, 89.5:10:0.5), B (chloroform: methanol: ammonium hydroxide: water, 55: 39: 0.5: 5.5). 95 % A was run for 5 min which was then linearly reduced to 60% in 7 min. After continuing for another 5 min, the condition was changed to 30% and was maintained for 15 min, then was changed back to initial gradient and kept for 5 min. Individual polar lipid species were quantified using multiple reaction monitoring transitions by referencing their internal standards.

#### Stable isotope labelled glucose, glutamine and competitive alpha-ketoglutarate (α-KG) metabolite tracing

Metabolic fate and catabolic flux of labelled glucose ($^{13}$C$_6$-U-glucose) and glutamine ($^{13}$C$_5$-U-glutamine) was performed as previously reported[167]. In primary cortical neurons, $^{13}$C$_6$-U-glucose or $^{13}$C$_5$-U-glutamine tracing followed by capillary electrophoresis-time of flight mass spectrometer (CE-TOF/MS) (Agilent Technologies) were performed. DIV14 primary cortical neurons were incubated in the glucose-free medium supplemented with 10 mM ($^{13}$C$_6$-U-glucose) (Cambridge Isotope, CLM-1396) or glutamine-free medium supplemented with 2 mM $^{13}$C$_5$-U-glutamine (605166, Sigma-Aldrich) and collected at 2 h of incubation at isotopic-steady state for both glycolysis and TCA cycle investigations[168]. Similarly, metabolic fate and competitive metabolic flux analyses of glutamine and α-KG were performed as well. In acute cerebellar slice culture, 2 mM $^{13}$C$_5$-U-glutamine isotope alone or simultaneously with 1,2,3,4-$^{13}$C$_4$-α-KG (Cambridge Isotope, 6363-53-7) for 2 h prior harvesting. To ensure steady-state tracing, all tracing analyses were conducted for durations exceeding the suggested timeframes elucidated by Jang and colleagues, whereby the isotopic-steady state occurs within approximately 10 min for glycolysis and around 2 h for the tricarboxylic acid (TCA) cycle in in vitro[168].

At harvesting, cells or tissues were then washed twice with 10 mL of 5% mannitol aqueous solution, and subsequently incubated with 1 mL of methanol containing 25 μm internal standards (methionine sulfone, 2-(N-morpholino) -ethanesulfonic acid (MES) and D-camphor-10-sulfonic acid) for 10 min. Four hundred microliters of the extracts were mixed with 200 μL Milli-Q water and 400 μL chloroform and centrifuged at 10,000 g for 3 min at 4 °C. Subsequently, 400 μL of the aqueous solution was centrifugally filtered through a 5-kDa cut-off filter to remove proteins. The filtrate was centrifugally concentrated and dissolved in 50 μL of Milli-Q water that contained reference compounds (200 μm each of 3-aminopyrrolidine and trimesate) immediately before metabolome analysis.

The relative concentrations of all the charged metabolites in samples were measured by CE-TOFMS, following the methods as previously reported[169]. In brief, a fused silica capillary (50 μm internal diameter × 100 cm) was used with 1 M formic acid as the electrolyte. Methanol: water (50% v/v) containing 0.1 μm hexakis (2,2-difluoroethoxy) phosphazene was delivered as the sheath liquid at 10 μL min −1. Electrospray ionization (ESI)-TOFMS was performed in positive-ion mode, and the capillary voltage was set to 4 kV. Automatic recalibration of each acquired spectrum was achieved using the masses of the reference standards. [(13 C isotopic ion of a protonated methanol dimer (2 MeOH + H)]+, m/z 66.0632) and ([hexakis (2,2-difluoroethoxy) phosphazene + H]+, m/z 622.0290). Quantification was performed by comparing peak areas to calibration curves generated using internal standardization techniques with methionine sulfone[169]. To analyze anionic metabolites, a commercially available COSMO(+)

(chemically coated with cationic polymer) capillary (50 μm internal diameter × 105 cm) (Nacalai Tesque) was used with a 50 mm ammonium acetate solution (pH 8.5) as the electrolyte. Methanol; 5 mm ammonium acetate (50% v/v) containing 0.1 μm hexakis (2,2-difluoroethoxy) phosphazene was delivered as the sheath liquid at 10 μL min −1. ESI-TOFMS was performed in negative ion mode, and the capillary voltage was set to 3.5 kV. For anion analysis, trimesate and CAS were used as the reference and the internal standards, respectively[170]. Mole percent enrichment of isotopes, an index of isotopic enrichment of metabolites, was calculated as the percent of all atoms within the metabolite pool that are labelled according to the established formula[171].

### Metabolic fuel flux assays

The Mito Fuel Flex Tests were performed on Seahorse XFe24 Bioanalyzer (Agilent). All assays were performed following manufacturer's protocols. In brief, the test inhibits import of three major metabolic substrates (pyruvate, fatty acids, and/or glutamine) with mitochondrial pyruvate carrier inhibitor UK5099 (2 μM), carnitine palmitoyltransferase 1 A inhibitor etomoxir (4 μM), or glutaminase inhibitor BPTES (3 μM). This test determines cellular dependence on each of the metabolites to fuel mitochondrial metabolism by inhibiting the individual substrate import. Baseline OCR was monitored for 18 min followed by sequential inhibitor injections (i.e., Treatment 1 or Treatment 2) with OCR reading for 1 h following each treatment. The inhibitor treatment combinations and calculations are shown as below:

| Metabolite test | Treatment 1 | Treatment 2 |
|---|---|---|
| Glucose/Pyruvate dependence | UK5099 | Etomoxir + BPTES |
| Glutamine dependence | BPTES | Etomoxir + UK5099 |
| Fatty acid dependence | Etomoxir | BPTES + UK5099 |

$$\text{Dependency}(\%) = ([\text{Baseline OCR} - \text{Target inhibitor OCR}] / [\text{Baseline OCR} - \text{All inhibitors OCR}]) \times 100\%.$$

### Glycolysis stress test

The XF Glycolysis Stress Test (Seahorse Bioscience, Agilent) was used to assess glycolysis function in cells, which was conducted using the XFe24 Analyzer. By directly measuring ECAR, the kit provided a standard method to assess the following key parameters of glycolysis flux: glycolysis, glycolytic capacity and glycolytic reserve, in addition to non-glycolytic acidification. Cells were seeded at a density of 45,000 cells/well. Cells were first incubated in pyruvate-free glycolytic assay medium for 1 h prior to the first injection of a saturated concentration of glucose (final concentration: 10 mM). The cells catabolize glucose into pyruvate via the glycolysis pathway, producing ATP, nicotinamide-adenine dinucleotide (reduced form), water and protons. The discharge of protons into surrounding medium leads to a sudden increase in ECAR, which was used to define the basal glycolytic capacity. The second injection was oligomycin (final concentration: 1 μM), which may divert energy production to glycolysis by restricting mitochondrial ATP production. Consequently, the sharp increase in ECAR indicates the level of glycolytic capacity. The final injection was 2-deoxy-glucose [2-DG; final concentration: 50 mM], which is a glucose analogue that inhibits glycolysis through competitive binding to glucose hexokinase; the first enzyme in the glycolytic pathway. The resulting decrease in ECAR confirmed that the ECAR produced in the experiment was caused by glycolysis. The gap between glycolytic capacity and glycolysis was defined as the glycolytic reserve. The ECAR prior to glucose injection is referred to as non-glycolytic acidification and may occur due to additional processes in the cell.

### Quantification procedures and statistical analyses

For each experiment, no statistical methods were used to predetermine sample sizes, but our sample sizes were similar to those reported in recent publications. An assessment of the normality of all datasets was conducted, with this analysis documented in the source data files. In instances where datasets exhibited non-normal distributions, non-parametric statistical tests were employed for further statistical evaluations. All samples were analyzed and data collected were blinded to the experimental conditions. All experiments were performed in at least three independent occasions. N-numbers presented represent biological replicates. Differences between two groups were analyzed using two-tailed unpaired Student's t-test (parametric) or two-tailed Mann-Whitney test (nonparametric). Differences between three groups were analyzed using one-way ANOVA (parametric) or Kruskal-Wallis test (nonparametric). Two-way ANOVA was used to determine the effect of two nominal predictor variables. The potential association between variants was analyzed using Pearson correlation coefficient. Except omics-related analyses, all statistical analyses were performed using GraphPad Prism 10 software package. Exact P-value is provided in each figure panel except when it is smaller than 1e-15 (i.e., the lower limit of the software", then "<1e-15" is labelled instead. $P < 0.05$ was considered to indicate statistical significance.

### Reporting summary

Further information on research design is available in the Nature Portfolio Reporting Summary linked to this article.

## Data availability

The single-nucleus RNA-sequencing dataset obtained from the human cerebellum in Ataxia Telangiectasia can be found under the accession code SCP1300 from the Single Cell Portal provided by The Broad Institute [https://singlecell.broadinstitute.org/single_cell/study/SCP1300/single-cell-atlas-of-the-human-cerebellum-in-ataxia-telangiectasia][76]. The snRNA-seq of the mouse cerebellar cortex can be found under the accession code SCP795 from the Single Cell Portal [https://singlecell.broadinstitute.org/single_cell/study/SCP795/a-transcriptomic-atlas-of-the-mouse-cerebellum][163]. The cerebellar cortex expression in ataxia-telangiectasia patients and normal controls from expression profiling by array can be found in GEO Omnibus under the accession code GSE61019[162]. The bulk RNA-seq data generated in this study have been deposited in GEO Omnibus under the accession code GSE222655. The phospho-proteomics/proteomics data have been deposited in PRIDE database under the accession code PXD062018. The metabolomics/lipidomics raw data generated in this study have been included in Source Data file. The processed "Minimum dataset" data that are necessary to interpret, verify and extend the research in the article, are in "Supplementary Dataset" files". Source data are provided with this paper.

## Code availability

All original codes for each individual figure are available at the following Zenodo website: https://zenodo.org/records/16777519.

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

## Acknowledgements

The authors acknowledge the Laboratory Animal Services Centre (LASEC) of the Chinese University of Hong Kong for their animal husbandry assistance as well as the University Research Facility in Chemical

and Environmental Analysis (UCEA) of the Hong Kong Polytechnic University for technical assistance in phosphoproteomic analysis. The work was supported, in part, by grants from the following: the National Natural Science Foundation-Young Scientist Fund 2023 (Ref.:32300643) and Excellent Young Scientists Fund 2020 (Ref.: 32022087); the Hong Kong Research Grants Council (RGC)-General Research Fund (GRF) (PI: ECS24107121 and PI: 11410824) and the RGC-Collaborative Research Fund (CRF) (Co-I: C4033-19EF); CUHK-Improvement on Competitiveness in Hiring New Faculties Funding Scheme (PI: ref. 132.), CUHK-Gerald Choa Neuroscience Institute (PI: Ref. 8425032) and School of Life Sciences Start-up funding to H.-M.C.

## Author contributions

H.-M.C. conceptualized the entire study. H.-M.C. acquired the funding. H-M.C., K.H. and R.P.H. supervised the study. H.-M.C. (all aspects), J.K.-L.S. (in vivo behavioral experiments, proteomics analyses), G.C.-N.W. (immunohistochemistry, in silico docking analysis), W.D. (transcriptomics analyses) and A.Z.P. (immunoblotting assays) devised the methodology, carried out the experiments and analyzed the data. K-M.K. offered reagents and advice on technical issues on immunohistochemistry. H.-M.C. wrote the original draft. H.-M.C., K.H. and R.P.H. reviewed the draft. All authors read and agreed to the final version of the manuscript.

## Competing interests

The authors declare no competing interests.
