## [Transparent Peer Review file · Nature Communications]

Alpha-ketoglutarate mitigates insulin resistance and metabolic inflexibility in a mouse model of Ataxia-Telangiectasia

Corresponding Author: Professor Kim Chow

Version 0:

Reviewer comments:

Reviewer #1

(Remarks to the Author)

Sun, Wong, and colleagues investigated the pathways involved in the metabolic flexibility induced by insulin and they found that ATM kinase plays a key role by phosphorylating PKM2 and HIF1a. This leads to their nuclear localization to promote biomass synthesis (via PKM2) and lipogenesis (via HIF1a). Mice lacking ATM -hence mimicking ataxia-telangiectasia (A-T), showed insulin signaling disrupted leading to metabolic inflexibility that results in glucose incompetence and glutamine dependence. The authors gave alpha-ketoglutarate (KG) (to bypass toxic ammonium production of glutamine) in the diet and found the rescue of both glucose incompetence and glutamine dependence. The experiments were performed in-depth, using a wide range of in silico, in vitro, and in vivo approaches to give rise to a dense and substantiated set of data to cover the vast majority of the signaling steps described. The main conclusion is potentially useful, as the A-T is intractable hence the authors suggest this work might open a novel avenue to palliate this devastating disease. Nevertheless, this reviewer has some comments and suggestions that the authors should pay attention to.

Comments

1. As the authors are aware, KG is required for HIF1a hydroxylation and destabilization (PMID: 17925579; PMID: 12432100), so one wonders to which extent an excess KG by the long-term dietary supplementation in ATM-deficient mice (or patients in A-T in a possible future trial) would worsen, rather than improve, the disease progression. Have the authors tested HIF1a activity in ATM-KO mice fed with KG?
2. A-T presents serious swallowing problems, so the potential use of a dietary strategy to treat A-T might represent a drawback. Have the authors confirmed KG benefits using a different administration route, such as i.p. or i.v.?
3. Regarding the use of both sexes, the authors indicate that behavioral studies were performed only in mice to increase consistency, but in in vitro experiments, both sexes were used, assumingly without segregation hence it is not possible to ascertain whether there are sex differences in the mechanisms and treatment efficacy. Could the authors show key experiments to disregard sex differences?
4. On occasions, cerebellar tissue cultures were used but in other experiments primary cortical neurons, cerebellar granule cells or neuronal-like cell lines were employed. This variety of biological models might distort some of the conclusions. For instance, the data shown in Fig. 2 were generated by analyzing cerebellar tissue lysates or primary cortical neurons, in spite that the data interpretation was used indistinctly. It is well known that different brain areas show metabolic heterogeneity (PMID: 20837536), as it happens with cerebellar and cortical cells (e.g., PMID: 36676133 for astrocytes; PMID: 26723542 for cerebellar granule neurons; PMID: 19448625 for cortical neurons). Whilst the authors clustered the cell types by annotation in the RNA-seq experiments, the rest of the experiments were performed in heterogeneous systems. Whilst this drawback does not invalidate the study, it weakens the preciseness and therefore the authors should declare this potential limitation of the study unless the key experiments are validated using the same biological system.

5. The involvement of NOX4 is shown exclusively by the evidence of the use of one antagonist drug, the efficacy and selectivity of which were not confirmed. The evidence for the ROS involvement in the insulin-ATM axis is also weak, as no experimental data directly showing ROS (e.g., fluorescent probes) are shown. The use of scavengers supports ROS involvement, but they do not prove it.

Reviewer #2

(Remarks to the Author)

In this manuscript, the authors presented how ATM kinase as an important mediator in Ataxia Telangiectasia. Importantly, they illustrated the mechanisms of how it is stimulated by insulin, which influences whether the cell undergoes catabolism or anabolism. Understanding the metabolism through isotope tracing, they implemented an alpha-ketoglutarate supplementation which was able to reverse the metabolic inflexibility and rescue the degenerating and ataxic cells.

Metabolomics and isotope tracing was convincingly used in 3 parts to illustrate the energy fuel inflexibility and switching:

1. ¹³C6-U-glucose tracer was used in ex vivo cerebellar culture of Atm^{+/+} and Atm^{-/-} mice. With glargine, Atm^{-/-} cells was seen to be unresponsive to stimulation, unlike the WT.
2. ¹³C6-U-glutamine tracer was used to show energy fuel switching to glutamine by observing increase in tracers in OAA. The number of tracers elegantly showed the extent of catabolism vs anabolism.
3. The experiments led to the use of AKG tracer supplementation, which effectively showed cells switching to using AKG as compared to glutamine.

Overall the metabolomics/tracing methods are sound and supports the conclusions of the paper.

Minor comments

1. Figure 1's resolution is quite poor, the labeling ^{+/+} and ^{-/-} cannot be seen clearly. Other figures like supplementary fig3a,d are also poor in resolution—the words in figures are unclear.
2. What is meant by P30 and P49 in figure 1? The other section I can find P49 is in the methods section, line 1357-9, but there was no mention of P30.
3. Please state the long form of Q in captions of Fig 6c and d. The many abbreviations make the report difficult to read.
4. Line 1677 "precipitation" should be "precipitate".
5. Please clarify what is "far filtrated" line 1690.
6. Line 1713: Change to "the limit of peak detection was determined by a signal-noise ratio of at least 3".

Comments:

1. Categorization of compounds need to be revisited: for eg Fig4a and 5l: In the figure, glycosylceramide, phytosphingosine, sphingamine, sphingolmyelin and sphingosine were categorized under ceramide. The broad categorization should be sphingolipids and not ceramide, as ceramide is a specific class of lipids under sphingolipids. Serine is an amino acid and should not be under PC/PE/PL. These are only some examples, and authors should check through thoroughly all compounds under the 3 categories: FA, PC/PE/PL and ceramide.
2. Can authors please put in a brief explanation of why n+3 OAA (Fig 3e) is anabolic rather than catabolic. How is the 3 labelled carbon OAA generated from citrate? This will enable readers to grasp the idea more easily.
3. With reference to Fig 6 and line 384, why is it that there is a preference to use AKG over glutamine when both were used in the treatment?
4. Methods of Stable isotope labelled glucose, glutamine and competitive alpha-ketoglutarate metabolite tracing (line 1764)—was steady state of metabolite flux ensured prior to harvesting of cells?
5. It is well known that in diabetic patients, branched chain amino acids are elevated. ATN deficiency is also reported in diabetes and insulin resistance. Is there any indication of the changes in BCAA utilisation in the mouse model? This question will be interesting for many metabolic disease researchers.

Reviewer #3

(Remarks to the Author)

In their study, "α-ketoglutarate corrects metabolic inflexibility in Ataxia-Telangiectasia", Sun et al. state to have discovered the role of ATM, a kinase in mediating the metabolic switch from catabolism to anabolism downstream of insulin. They state that this finding explains the metabolic phenotype observed in patients with Ataxia telangiectasia (AT). However, the finding that ATM acts downstream of insulin signaling and causes the metabolic phenotype of AT has already been reported (Kanan et al., Nat. Cell Bio., 2000) and also the role of ATM in phosphorylating and controlling a translation initiation factor has been nicely described, the data presented in this study does not give any insights into the role of ATM in

the signaling pathway.

Overall, the study contains an accumulation of confusing conclusions that are not supported by the data, which lack quality. In the abstract, the authors state that they define the role of ATM kinase insulin signaling and in mediating a metabolic switch. However, I did not find a single piece of data in the paper that would tell anything about how ATM acts in insulin signaling or how it functions at the molecular level.

Overall, this study lacks any structure and argumentation lines. Even for an expert in metabolism and phosphoproteomics, it is impossible to follow the argument. Conclusions are drawn from experiments with poor data quality, which are not supported by the data and are connected with the strange interpretations of other studies. The connections between the experiments and the rationales behind the experiments are unclear. Poor language and grammatical errors make the reading difficult. The worst thing is that the authors use exaggerated language to oversell each finding. Partial sentences or titles have no logical or scientific meaning (for example, "Insulin-induced unique ATM glycolytic-centric phospho-proteome"). "Insulin-induced unique ATM glycolytic-centric phospho-proteome").

Main points:

It is not clear why the authors focused on the cerebellum to study the metabolic consequences of ATM mutations in a mouse model. Even if insulin signaling is impaired in this brain region, it is unclear whether this would have any consequences on whole-body energy metabolism. There is no indication that the neuronal and metabolic phenotypes are connected. It is much more likely that the impairment of insulin signaling in the liver or adipose tissue leads to a metabolic phenotype. In the phosphoproteomic paragraph, it was first described that the cerebellum is insulin responsive, then that insulin levels in ATM-KO mice are elevated, and finally, they test a H₂O₂ scavenger. Despite reading this passage ten times, it was impossible to understand the rationale behind the conducted experiments, how they are connected, and what should be the outcome or conclusion.

Overall, the phosphoproteomic data did not meet any quality criteria of the field. The authors did not provide any quality assessments, information about sample correlations, PCA, number of identified sites, or reproducibility. The text does not contain information about data filtering, normalization procedures, statistical testings, or FDR. The fact that only ~100 regulated sites were found indicates that the workflow was of very poor quality, as many other studies have mapped thousands of regulated sites upon insulin stimulation. It is very surprising that the authors select a 6h treatment time point. Normally, insulin treatment reaches a maximum of signaling intensity 1-5 minutes. After 6 h of treatment, changes in the total proteome are induced, which required normalization of phosphorylation to protein levels. A closer examination of the supplemental tables further shows that the authors did not detect most canonical insulin targets. Therefore, pathway analysis or information gain based on these data is not possible. Moreover, it is very strange that, upon KD of a kinase, the number of regulated sites decreases, since it would be expected the other way around. This further points to problems with the data quality. The authors did not conduct any analysis steps, which are normally performed in the proteomics field, such as an analysis of regulatory sites, principal component analyses, or hierarchical clustering.

The authors concluded that any change in phosphorylation in ATM-KD cells is a direct target of ATM. However, this conclusion is not valid since indirect effects can also lead to phosphorylation changes. The authors further modeled in silico protein interactions between phosphorylated PKM2 and an importin and stated that this proves that PKM2 is shuttled to the nucleus induced by ATM phosphorylation. The microscopy images that were supposed to prove this showed a single cell and no image-based quantification. Why were the proteins overexpressed instead of immunostaining for endogenous protein levels?

The analysis of transcriptomic and metabolomics data remains very superficial, and no integration of the two levels occurs. It is only stated that the data suggest some metabolic dysregulation". However, what this really means remains unclear.

Overall, the manuscript would strongly profit from a profound restructuring with emphasis on key findings and messages and more quality control of data.

Minor comments:

- The abstract should be intensively edited since no key message becomes clear
- L46. the statement that insulin induces a Warburg effect should be revised since this refers to the state in cancer cells and not the physiological insulin action
- L48 if PKM2 moves to the nucleus upon phosphorylation, this is not "spontaneous"

Reviewer #4

(Remarks to the Author)

Title: α -Ketoglutarate corrects metabolic inflexibility in Ataxia-Telangiectasia

Authors: Jacquelyne Ka-Li SUN et al.

ATM is a multitasking protein that regulates many cellular processes. Its main function is the regulation of the DNA damage response, but ATM is also involved in mitochondrial as well cytoplasmic processes. In the current manuscript, Sun et al. showed that insulin can stimulate the activity of Atm and thereby the activation of a protein network that promotes the Warburg effect. The mechanism involves nuclear localization of PKM2 and HIF1 α , which initiates a cascade of events that regulate the cellular metabolic state. Furthermore, the authors showed that supplementation of Atm-deficient mice with α -Ketoglutarate markedly extended the lifespan of the mutant mice and improved their motor skills. In its current form, the manuscript does not merit publication in Nature Communications. The concerns detailed below must be fully addressed prior to consideration of publishing the revised version of this manuscript.

Major points:

1. General comments: It is known that the Atm deficiency has little effect on the morphology and the functionality of the cerebral cortex. Atm's effects are specific to certain cells. I recommended that the authors focus on those cells that are highly dependent on Atm. A-T is a progressive disease; thus, it is highly important to follow the metabolic and behavioral changes in young, adult, and aged mice. There is a difference between Atm^{-/-} and Atm inhibitor-treated cells. I highly recommended

the use of *Atm*^{-/-} cells. Inhibitors can have off-target effects that are not detected in *Atm*^{-/-} cells. Cerebral cortical neurons should be used as controls to compare changes caused by *Atm* deficiency in cerebellar versus cerebral cells. This comparison can hint at the causality of the metabolites, the pathways, and the treatments. It is thought that A-T has a developmental component. Thus, the use of *Atm* inhibitor in healthy cells is problematic.

2. The normal range of RQ is 0.7-1. Thus, the *Atm*^{-/-} mice are within the normal range.

3. Lines 128-138: It seems that 6-month-old *Atm*^{-/-} +STZ and Glargine behave very similarly to WT mice under the same treatment. This has to be discussed.

4. Figure 2e: There are signs of pAtm and Atm in *Atm*-KO mice. This is rather unusual!

5. Line 167: If *Atm* is involved in oxidative stress and affected by it, then the authors must show that the *Atm* is not phosphorylated on S1981 but rather forms a dimer through a disulfide bond at C2991.

6. The data shown in Figs. 2e and f should be quantified and subjected to statistical analyses.

7. Line 175: "...insulin stimulus in primary neurons harvested from WT and *Atm*-KO mice". What type of neurons were used? 8. Is ATM involved in the nuclear transportation of Hif1 to the nucleus? This has to be addressed.

9. Line 204 and Fig. 5f: The use of the term "equilibrium" when discussing living cells is problematic. All the cellular reactions in the cells are maintained in an energy-dependent steady state, which is a dynamic situation. Equilibrium occurs at steady state and the free energy is zero.

10. Figure 3a (left panel) is not clear and more detailed explanation is needed.

11. Figure 4b shows western blot analysis and does not really show ALDOC in PCs. The data should be quantified and subjected to statistical analyses. There is high variability in the levels of *Atm*^{+/+} GAD67. What happened in 3-month-old mice? It is hard to judge the ALDOC levels in Fig. 4f, and it seems that there is immunoreactivity in the white matter of the cerebellum.

12. Line 317: The rationale of the addition of TEPP-46 should be explained. It would be helpful to show the effect of TEPP-46 on *Atm*^{-/-} mice.

13. Figure 5a-c should include *Atm*^{-/-} alone.

14. Fig. 5g: The western blot should be quantified and subjected to statistical analysis. Surprisingly, *Atm*^{-/-} mice appear to express *Atm*! If glargine can induce the expression of *Atm* in *Atm*^{-/-} mice, it is a very unexpected result! Also, there is a background of pAtm in *Atm*^{-/-} mice. This has to be explained.

15. A-T patients do not display cerebral cortex abnormalities. Thus, if the cerebral cortex is also insulin sensitive, I have doubts about the causality of the insulin effects.

16. Hif1 is upregulated by hypoxia. Is there any connection between glargine and hypoxia?

17. As shown in Fig. 5m, glargin treatments failed to induce any positive effect on open field test results in WT and *Atm*^{-/-} mice! What can we conclude from that?

18. It would be informative to determine the cause of death of the *Atm*^{-/-} mice and to understand the effect of alfa-KG on those pathways that lead to the death of *Atm*^{-/-} mice.

19. Fig. S14 does not provide any data regarding preservation of PCs! It would be very helpful if this could be shown in the main figures.

20. Statistical analyses: The authors assumed that all their data are distributed normally. This must be tested for each experiment, and the data should be analyzed using the proper statistical tests (parametric or non-parametric).

Minor points:

1. Protocols for cerebellar cell cultures are not well presented. It is not clear which kind of cultures were used. This has to be clarified.

2. Figure 1i: The colors are confusing! Sometimes green indicates WT and sometimes KO. Be consistent to prevent confusion.

3. Figs 2i and j are not data but demonstration and can be moved to the supplementary section.

Version 1:

Reviewer comments:

Reviewer #1

(Remarks to the Author)

The authors have successfully and adequately addressed all comments raised by this reviewer and, therefore, the manuscript is now much improved.

Juan P. Bolanos

Reviewer #2

(Remarks to the Author)

All comments have been addressed satisfactorily. Relevant figures and information with regards to metabolomics/lipidomics and tracing experiments have been corrected and are clear. I have no further questions.

Reviewer #3

(Remarks to the Author)

Also the revised form of the manuscripts lacks a clear conceptual framework or logical structure ("red thread") that guides the reader through the rationale, data, and conclusions. As it stands, the manuscript reads as fragmented and somewhat

confusing, even for a specialized audience. For a broader scientific readership, it would be impossible to follow. the use of domain-specific jargon without explanation starting already in the abstract.

I am particularly concerned about the quality and rigor of the statistical analyses, especially concerning the phosphoproteomics data. The manuscript does not clearly state how p-values were corrected for multiple hypothesis testing. From what is presented, it appears that uncorrected p-values were used, which is not acceptable given the large number of comparisons inherent in phosphoproteomic analyses. Moreover, the manuscript lacks a thorough and appropriate pathway analysis. Without this, it is unclear whether any of the reported changes are biologically significant or relevant. Although the authors attempted to address my points of criticism in their rebuttal letter, I find that many of the responses were not convincing. Several core issues, such as the lack of proper statistical handling, weak data interpretation, and unclear manuscript structure, remain unresolved. In summary, I believe the manuscript in its current form is not suitable for publication and requires substantial revisions to address the critical issues outlined above.

Reviewer #4

(Remarks to the Author)

1. The results are important showing that A-T is a metabolic disease and supplementation of alfa-keto-glutarate can alleviate the detrimental symptoms of A-T.
2. Yes. the work provides an important insight into the molecular mechanisms of A-T.
3. Yes. the authors present sufficient data that support their conclusions.
4. I did not find any flaws in data analyses, interpretation and the conclusions.
5. Yes. the methodologies are sound and meet the expected standards in the field of A-T.
6. The authors provide sufficient information regarding the methods for work to be reproduced.

Version 2:

Reviewer comments:

Reviewer #3

(Remarks to the Author)

My concerns have been addressed

POINT-TO-POINT RESPONSE TO REVIEWERS

REVIEWER COMMENTS

Reviewer #1 (Remarks to the Author):

Comment 1—As the authors are aware, KG is required for HIF1 α hydroxylation and destabilization (PMID: 17925579; PMID: 12432100), so one wonders to which extent an excess KG by the long-term dietary supplementation in ATM-deficient mice (or patients in A-T in a possible future trial) would worsen, rather than improve, the disease progression. Have the authors tested HIF1 α activity in ATM-KO mice fed with KG?

Response 1— In response to the aforementioned comment, we would like to first provide a brief summary of our key findings for clarifications, before commenting our interpretations on the HIF1 α biology.

Brief summary of our key findings:

Pyruvate kinase (PKM) assumes a pivotal role in the enzymatic conversion of phosphoenolpyruvate to pyruvic acid during the terminal step of glycolysis, thereby exercising critical regulation over the final fate of glucose carbons (Schormann et al., 2019). In response to an oral or intraperitoneal load of carbohydrates, a physiological duration of insulin response that typically lasts for approximately 2 hours would be initiated (Kim et al., 2006, Savage et al., 1975a, Savage et al., 1975b, Dowse et al., 1993) (**Revised manuscript: Fig.2a**). This activates ATM via an oxidative stress-related mechanism (**Revised manuscript Fig.2e-f, S2d-h**), which subsequently promotes the ATM-mediated phosphorylation and disassociation of PKM2 tetramers back as dimers or monomers (**Revised manuscript: Fig. 2q, S2**). Such quaternary structure disruption diminishes the overall metabolic activity of pyruvate kinase (**Revised manuscript: Fig.2n and 2q**), prompting a diversion of glucose carbons towards biomass generation (**Revised manuscript: Fig. 3a-e, S5**), as also reported by others (Kim et al., 2018). However, in the context of prolonged insulin exposure (i.e., hyperinsulinemia), the sustained activation of ATM further influences the behaviour of PKM2. Acting as a co-activator (Luo et al., 2011), PKM2 in conjunction with delayed accumulation and nuclear translocation of ATM-phosphorylated HIF1 α , reshape a metabolic landscape that favours lipid and biomass biogenesis (**Revised manuscript: Fig.5, S11**). The resulting outcome at the systemic level is a global anabolic response, as reflected by body weight gain and growth. While this phenomenon could be considered as a regular physiological occurrence during periods of adolescence and puberty, attributable to the impact of rapid growth (Caprio et al., 1989, Bloch et al., 1987), yet this could also potentially transit into a pathological condition at other life stages, manifest as excessive fat deposits and becoming overweight. Conversely, the deficiency of ATM, as observed in A-T, impedes the anabolic responses evoked by insulin. ATM deficiency contributes to inherent defects in insulin signalling that leads to an initial hyperinsulinemia as a negative feedback response to ineffective blood glucose homeostasis at the system level (**Revised manuscript: Fig.1**), and gradual progression to type 2 diabetes. These gradual endocrine failure and metabolic inflexibility reflect as stunt growth, lean muscle loss, and unexpectedly cerebellar degeneration in individuals with A-T as they reach early adolescence and puberty ages (Natale et al., 2021).

1. Quote “As the authors are aware, KG is required for HIF1 α hydroxylation and destabilization (PMID: 17925579; PMID: 12432100), so one wonders to which extent an excess KG by the long-term dietary supplementation in ATM-deficient

mice (or patients in A-T in a possible future trial) would worsen, rather than improve, the disease progression.”

We acknowledge the reasoning behind Reviewer #1’s comment. He/she highlighted that AKG-dependent enzymes, such as prolyl hydroxylases (PHDs) and certain asparaginyl hydroxylases (e.g., Factor inhibiting HIF-1, FIH1), could destabilize and inhibit the activities of HIF1 α (Dann et al., 2002). Therefore, he/she argues that the supplementation strategy involving AKG may instead impose a risk of further reducing residual HIF1 α activity and its anabolic impact in conditions characterized by *Atm* deficiency, and hence potentially exacerbating disease progression. We would argue this with the facts that indeed the activities of PHDs/FIH1 are also dependent upon additional cofactors, including Fe (II) ions, oxygen, and ascorbate (Hoang and Joseph, 2020). Furthermore, immediate metabolites downstream of α -KG in the TCA cycle, such as succinate and fumarate, may however inhibit these enzymes. It is also noteworthy that PHD3, in contrast to other PHDs, exhibit the ability to interact with and promote PKM2 in functioning as a HIF1 α coactivator (Luo et al., 2011). These different literature findings suggest an existence of a complex regulatory network (Baksh and Finley, 2021), indicating that the modulation of PHD activities within the cellular milieu is influenced by a multitude of factors rather than a singular determinant, e.g. AKG. In the revised manuscript, not only did we add these information to the discussion section (**Revised manuscript: Lines 602-623**), we also provided additional experimental evidence that long term AKG supplementation did not significantly alter nuclear activities of HIF1 α in cerebellar tissues or human patient fibroblast (**Revised manuscript: Fig.S16h-i**).

2. Quote “Have the authors tested HIF1a activity in ATM-KO mice fed with KG?”

In our working models of ATM deficiency, our findings indicate that:

Evaluation of HIF1 α protein levels and nuclear activities in tissue lysates obtained from experimental animals indicated that chronic AKG supplementation did not result in any notable differences when compared to animals that being fed on vehicle controls (**Revised manuscript: Fig.S16h-i**). This comparable trend was also noted in fibroblasts from A-T patients following chronic AKG treatment (**Revised manuscript: Fig.S16i**).

To explain so, our metabolic tracing data concerning the fate of exogenous AKG indicated that majority of AKG was indeed actively metabolized into GABA or TCA cycle metabolites (such as succinate, citrate, and OAA) in *Atm*-KO samples (**Revised manuscript: Fig.6b-c**). Additionally, our results reveal that exogenous AKG even surpasses the contribution of these metabolites from glutamine when both are concurrently available in the same system (**Revised manuscript: Fig.6b-c**). Notably, when both the isocitrate dehydrogenase-1 (IDH1) (Inhibitor: Compound 13/IDH1i) (Tian et al., 2022) and aspartate aminotransferase (GOT1) (Inhibitor: Compound 2c) (Anglin et al., 2018)—the two pivotal cytosolic enzymes facilitating AKG entry into the metabolic network—were inhibited (**Revised manuscript: Fig.6a, S16i**), a slight reduction in basal HIF1A nuclear activities was detected in both *ex vivo* acute *Atm*-KO cerebellar cultures and human A-T patient fibroblasts (**Revised manuscript: Fig.S16i**). This observation, therefore further emphasize that the majority of the AKG introduced into the system is actively integrated into the fuel metabolic network of these cells,

rather than contributing to the activation of the PHD-mediated HIF1 α degradation pathway, as further elaborated in our response to “Quote 1” of this comment.

Comment 2—A-T presents serious swallowing problems, so the potential use of a dietary strategy to treat A-T might represent a drawback. Have the authors confirmed KG benefits using a different administration route, such as i.p. or i.v.?

Response 2—We thank the reviewer for reminding us the difficult reality encountered by most A-T patients and the valuable suggestions. Herein, we aim to elucidate the rationale behind the adoption of the dietary approach in our study.

Adolescents with A-T commonly encounter feeding and swallowing challenges, with no definitive remedy currently available for the associated swallowing difficulties in A-T. Nonetheless, modifications in mealtime practices can potentially ameliorate these issues, facilitating easier ingestion and reducing the likelihood of choking and aspiration. As delineated in the caregiver resources provided by the A-T Children’s® Project, solid foods can be adapted to softer textures, while liquids can be thickened to a consistency that promotes cohesive retention in the oral cavity, facilitating smoother swallowing for patients.

Given the enduring nature of our metabolic reprogramming objective associated with AKG supplementation, spanning months to potentially years, to counterbalance the inherent metabolic deficiency in A-T, and considering as well previous pharmacokinetic investigations which have highlighted the relatively brief *in vivo* half-life of AKG (approximately 1 hour) (Dabek et al., 2005, Guo et al., 2022b); a sequential supplementation regimen via 24/7-accessible consumables like drinking water or food chow emerges as a more pragmatic approach for animal experimentation (Guo et al., 2022b). Furthermore, with previous reports on successful long-term dietary supplementation of calcium alpha-ketoglutarate (CaAKG) in enhancing overall metabolic health (Yuan et al., 2022, Asadi Shahmirzadi et al., 2020), this further bolsters the rationale for the selected approach. In light of the practical guidance offered by the A-T Children’s® Project and the feasibility concerns pertaining to post-hoc analyses, a decision was made to administer CaAKG supplementation via standard animal chow in the form of soft food pellets, as opposed to water supplementation. This choice was made to preempt potential confounding factors that could arise from liquid thickening agents such as commercial products like Thick-it® or consumables like milkshakes/malts, which may introduce additional nutritional elements complicating the *in vivo* metabolic phenotyping analyses.

Nevertheless, in response to the reviewer's recommendation, additional experiments were added in this revised version to introduce CaAKG into the test subjects via the conventional injection route. The existing literature indicates the potentials of intraperitoneal (i.p.) administration of AKG, as documented by Yamamoto (1992) and Yuan et al. (2022), albeit predominantly in studies characterized by single-dose settings aimed at assessing acute physiological responses (Yuan et al., 2022, Yamamoto, 1992). Notably, recent investigations have also highlighted that mice subjected to i.p. administration of AKG at a dosage of 10 mg/kg exhibited an acute surge in circulating blood levels comparable to those achieved following chronic 2% alpha-ketoglutarate water supplementation (Yuan et al., 2022). Additionally, these AKG-treated mice manifested immediate reductions in blood glucose concentrations at 1, 2, and 3 hours post-injection in contrast to the saline-treated counterparts (Yuan et al., 2022). To align with these findings (Yuan et al., 2022), a regimen involving daily i.p. injections of 10 mg/kg/day CaAKG was implemented in P25 (post-natal day 25) mice over a span of 60 days (**Revised manuscript: Fig.S16a-e**). Comparative analysis revealed mice which was

administered 2% CaAKG consecutively through softened food pellets for the same treatment duration exhibited superior blood glucose control outcomes than those subjected to i.p. injections (**Revised manuscript: Fig.S16b**). Furthermore, metabolic profiling performed at P85 indicated a less prominent improvement in metabolic flexibility in mice receiving i.p. injections compared to those receiving dietary supplementation of 2% CaAKG, as reflected by less amount of changes in average respiratory quotient (RQ) between the dark and light cycles (**Revised manuscript: Fig.S16c**). Notably, parameters such as food intake (**Revised manuscript: Fig.S16d**) and locomotor activity levels remained fairly consistent across all experimental treatment groups (**Revised manuscript: Fig.S16e**).

We predicted that the observed discrepancies are mainly stemming from the abbreviated half-life of free AKG *in vivo*, which compromises the sustained need of metabolic reprogramming objective, particularly in the extended period following i.p. administration. Therefore, the findings from this study supports that a regimen of consecutive supplementation throughout the day represents a more efficacious approach in achieving the desired metabolic outcomes.

Comment 3—Regarding the use of both sexes, the authors indicate that behavioural studies were performed only in mice to increase consistency, but in *in vitro* experiments, both sexes were used, assumingly without segregation hence it is not possible to ascertain whether there are sex differences in the mechanisms and treatment efficacy. Could the authors show key experiments to disregard sex differences?

Response 3—In response to the feedback, we have incorporated the missing behavioural data from female mice into all our experiments, affirming the consistency of the derived conclusions with our previous findings. Total and sex-specific analyses were performed for all the experiments, if applicable, the statistical data can be found in the source data files.

One exception arose in the context of the study involving streptozotocin (STZ) injection (**Revised manuscript: Figure 1i-m, S1c-g**). It is noteworthy that STZ, a pharmacological agent utilized to induce diabetes in laboratory mice, as documented in the literature (Furman, 2015), exhibits differential sensitivity in female mice due to potential protective effects conferred by estrogen, as elucidated by Paik and colleagues (Paik et al., 1982). Consequently, owing to the inherent complexities introduced by the estrous cycle in fertile female mice and the disparate STZ dosages required to elicit pancreatic β -cell toxicity (with female subjects typically needing higher STZ doses than their male counterparts), the existing literature have favoured the utilization of male animals in STZ-induced diabetic mouse studies (Kolb, 1987, Furman, 2021). Considering these reasons, we declare that this set of experiments was conducted only with male subjects, to ensure consistency and comparability with existing literature and methodologies.

Comment 4—On occasions, cerebellar tissue cultures were used but in other experiments primary cortical neurons or neuronal-like cell lines were employed. This variety of biological models might distort some of the conclusions. For instance, the data shown in Fig. 2 were generated by analyzing cerebellar tissue lysates or primary cortical neurons, in spite that the data interpretation was used indistinctly. It is well known that different brain areas show metabolic heterogeneity (PMID: 20837536), as it happens with cerebellar and cortical cells (e.g., PMID: 36676133 for astrocytes; PMID: 26723542 for cerebellar granule neurons; PMID: 19448625 for cortical neurons). Whilst the authors clustered the cell types by annotation in the RNA-seq experiments, the rest of the experiments were performed in heterogeneous systems. Whilst this drawback does not

invalidate the study, it weakens the preciseness and therefore the authors should declare this potential limitation of the study unless the key experiments are validated using the same biological system.

Response 4— We express our appreciation for the valuable recommendations provided by the reviewer. In accordance with the feedback received, we have reintegrated the following statement into the discussion section to declare the limitation (**Revised manuscript: Lines: 681 to 688**).

“It is also noted that while the widespread metabolic perturbation inherent to the disease were consistently observed across various tissues sampled from the Atm-KO mouse model, it is crucial to recognize that such analyses may have limitations in fully elucidating possible metabolic differences that could have pre-existed among distinct cell subtypes (e.g., neurons versus glia) located within different regions of the tissue. Furthermore, the clarity on whether a dosage-dependent effect exists, particularly in patients with mutations leading to residual ATM proteins, remains uncertain. Addressing these nuances will necessitate dedicated investigations in future studies.”

Comment 5—**The involvement of NOX4 is shown exclusively by the evidence of the use of one antagonist drug, the efficacy and selectivity of which were not confirmed. The evidence for the ROS involvement in the insulin-ATM axis is also weak, as no experimental data directly showing ROS (e.g., fluorescent probes) are shown. The use of scavengers supports ROS involvement, but they do not prove it.**

Response 5—In compliance with the reviewer’s requests, we employed the cytosol- targeted redox-sensitive roGFP biosensors with real-time imaging technique (Kim et al., 2017) as well as the CellROX method in primary cortical neurons and human patient fibroblasts. These methodologies were utilized to demonstrate the elicitation of cytosolic reactive oxygen species (ROS) upon acute exposure to glargine (**Revised manuscript: Fig.S2d-e**).

The specific involvement of NOX4 in mediating the impact of glargine within the cerebellum was confirmed through gene knockdown experiments conducted in cerebellar tissues, as illustrated in **Revised manuscript: Fig.S2f-g**.

Reviewer #2 (Remarks to the Author):

Comment 1—Figure 1’s resolution is quite poor, the labelling +/- and -/- cannot be seen clearly. Other figures like supplementary fig3a,d are also poor in resolution—the words in figures are unclear.

Response 1—In response to this comment, the symbols of +/- and -/- have been substituted with "WT" and "KO" correspondingly across all visual representations in the revised manuscript. Enhanced-resolution versions with improved textual clarity have been introduced to the figure panels as mentioned by the Reviewer, which are now located at **Revised manuscript: Fig.S5a-d.**

Comment 2—What is meant by P30 and P49 in figure 1? The other section I can find P49 is in the methods section, line 1357-9, but there was no mention of P30.

Response 2— In acknowledgment of this critique, it is noted that P30 and P49 within **Figure 1** serve as abbreviations denoting the ages of the mice, specifically postnatal day 30 and 49, respectively. To enhance comprehension, the abbreviation labelling indicating "P = postnatal day" has been reintegrated into the corresponding figure panels (**Revised manuscript: Fig.1i-l**).

Comment 3—Please state the long form of Q in captions of Fig 6c and d. The many abbreviations make the report difficult to read.

Response 3—In response to the suggestion, abbreviation labelling, specifically "Q = glutamine," has been reinstated within the relevant figure panels (**Revised manuscript: Fig.6c-d**).

Comment 4—Line 1677 “precipitation” should be “precipitate”.

Response 4—In response to this reminder, we amended the wordings accordingly in the **Methods and Materials** section of “Untargeted metabolome analysis by capillary electrophoresis time-of-flight mass spectrometry (CE-TOFMS) and liquid chromatography (LC)-TOFMS” experiment.

Comment 5—Please clarify what is “far filtrated” line 1690.

Response 5— We express gratitude to the reviewer for highlighting the typographical error; it should read as "further filtrated," and appropriate corrections have been implemented as advised in the **Methods and Materials** section of “Untargeted metabolome analysis by capillary electrophoresis time-of-flight mass spectrometry (CE-TOFMS) and liquid chromatography (LC)-TOFMS” experiment.

Comment 6—Line 1713: Change to” the limit of peak detection was determined by a signal-noise ratio of at least 3”.

Response 6—In response to this suggestion, we have amended the wording accordingly in the **Methods and Materials** section of “Untargeted metabolome analysis by capillary

electrophoresis time-of-flight mass spectrometry (CE-TOFMS) and liquid chromatography (LC)-TOFMS” experiment.

Comment 7—Categorization of compounds need to be revisited: for eg Fig4a and 5l: In the figure, glycosylceramide, phytosphingosine, sphingamine, sphingomyelin and sphingosine were categorized under ceramide. The broad categorization should be sphingolipids and not ceramide, as ceramide is a specific class of lipids under sphingolipids. Serine is an amino acid and should not be under PC/PE/PL. These are only some examples, and authors should check through thoroughly all compounds under the 3 categories: FA, PC/PE/PL and ceramide.

Response 7— We extend our appreciation to the reviewer for bringing this to our attention. In response, we have reclassified the specified group of metabolites in **Figures 4a, 5i, and S11c of the revised manuscript** into eight distinct categories based on their inherent characteristics, as also shown below.

Categories	Metabolites
Neurotransmitter	4-amino-butanoic acid
TCA cycle intermediates	Acetyl-CoA, Succinate acid
Fatty acid	Lauric acid, arachidic acid, arachidonic acid, oleic acid, palmitic acid, palmitoleic acid, stearic acid, myristic acid, pelargonic acid
Phospholipid	Phosphatidylcholine, phosphorylcholine, stearyl ethanolamide, O-phospho-ethanolamine, ethanolamine
Sphingolipid	Sphinganine, sphingomyelin (d18:1/16:0), sphingomyelin (d18:1/18:0), sphingosine (d20:1), phytosphingosine, glucosylceramide (d18:1/24:1))
Amino acid	Serine
Glycolysis intermediates	F2,6P, G3P, DHAP
Choline reaction intermediates	CDP, CMP, acetylcholine

Comment 8—Can authors please put in a brief explanation of why M+3 OAA (Fig 3e) is anabolic rather than catabolic. How is the 3 labelled carbon OAA generated from citrate? This will enable readers to grasp the idea more easily.

Response 8—In response to the comment, it is elaborated in the figure that the M+3 oxaloacetate (OAA) is uniquely derived from cytosolic citrate (**Revised manuscript: Fig.3e**), while those OAA derived from mitochondrial citrate were always heavily labelled with even numbers of heavy carbons (i.e., M+2, M+4) (**Revised manuscript: Fig.3e**). The former occurs in a reaction where each cytosolic M+5 citrate is cleaved into M+2 acetyl-CoA, a pivotal compound for initiating lipid/palmitate biogenesis, signifying its anabolic nature. While the M+3 OAA produced does not directly contribute carbons to palmitate biosynthesis but rather appears as a by-product of the reaction, we have recognized its significance in facilitating the conversion of cytosolic citrate to acetyl-CoA, and hence we previously categorized it as anabolic (A). Nevertheless, since we did not elaborate further on the fate of this M+3 OAA, and to avoid confusion, we removed such label in the revised figures.

Comment 9—With reference to Fig 6 and line 384, why is it that there is a preference to use AKG over glutamine when both were used in the treatment?

Response 9— Drawing from our analysis of data extracted from a published A-T cerebellar transcriptome dataset (**Revised manuscript: Fig.6a**), it is evident that the expression of pivotal genes facilitating exogenous AKG uptake, notably NaDC1/*SLC13A2* and NaDC3/*SLC13A3*, were up-regulated in A-T (**Revised manuscript: Fig.6a, left panel**). Conversely, the cell surface receptor essential for glutamine uptake (*SLC1A5*) did not demonstrate significant alterations in A-T (**Revised manuscript: Fig.6a, right panel**). Given that the subsequent metabolic pathways of AKG and glutamine converge following their common conversion to glutamate (Glu), the heightened expression of cell surface receptors for AKG relative to glutamine in A-T cases implies a preference for AKG over glutamine and that the latter is likely sourced from muscle breakdown in *Atm*-deficient cells/tissues, as outlined in the study by Cruzat and colleagues (Cruzat et al., 2018). These changes therefore likely account for the phenomena (**Revised manuscript: Fig.6b-c**). To enhance clarity on this concept, we have added supplementary labels in the updated figure panel (**Revised manuscript: Fig.6a**) and descriptive text in the revised manuscript.

Comment 10—Methods of Stable isotope labelled glucose, glutamine and competitive alpha-ketoglutarate metabolite tracing (line 1764)—was steady state of metabolite flux ensured prior to harvesting of cells?

Response 10— In the context of isotope tracer investigations, it is often advantageous to minimize metabolic disturbances during tracer introduction, thereby preserving a state of "metabolic steady-state." This objective can be attained by transitioning to media that are otherwise identical but contain specific nutrients in a labelled rather than unlabelled form, as outlined by Jang and colleagues (Jang et al., 2018). Our tracing experiments meticulously adhered to this approach, and additional experimental details have been reintegrated into the **Methods and Materials** section of "Stable isotope labelled glucose, glutamine and competitive alpha-ketoglutarate (α -KG) metabolite tracing" in the revised manuscript.

Moreover, as elucidated in the literature (Jang et al., 2018), the achievement of steady-state labelling, referred to as "isotopic-steady state," typically occurs within approximately 10 minutes for glycolysis and around 2 hours for the tricarboxylic acid (TCA) cycle in *in vitro* settings. To ensure steady-state tracing, all tracing analyses were conducted for durations exceeding these suggested timeframes. These procedural details have also been incorporated into the **Methods and Materials** section of "Stable isotope labelled glucose, glutamine and competitive alpha-ketoglutarate (α -KG) metabolite tracing" in the revised manuscript.

Comment 11—It is well known that in diabetic patients, branched chain amino acids are elevated. ATM deficiency is also reported in diabetes and insulin resistance. Is there any indication of the changes in BCAA utilisation in the mouse model? This question will be interesting for many metabolic disease researchers.

Response 11— In light of the comprehensive global metabolite profiling data analysis, a discernible reduction in valine levels was observed in cerebellar tissues from 6-month-old ATM-KO mice, while the concentrations of leucine and isoleucine exhibited no significant variance between *Atm*-KO and WT samples (**Revised manuscript: Table S7**). Furthermore, a notable elevation in succinic acid levels was detected in the cerebellar tissues of 6-month-old

Atm-KO mice. Succinic acid, a metabolite that could be directly stemming from valine-derived succinyl-CoA in the tricarboxylic acid (TCA) cycle reaction, displayed augmented levels in the *Atm*-KO samples (**Revised manuscript: Table S7**). Although direct experimental validation regarding the carbon source of succinic acid from valine was lacking, this observation provides insight into the unique induction of this TCA cycle intermediate in the examined samples.

Reviewer #3 (Remarks to the Author):

Comment 1—It is not clear why the authors focused on the cerebellum to study the metabolic consequences of ATM mutations in a mouse model. Even if insulin signaling is impaired in this brain region, it is unclear whether this would have any consequences on whole-body energy metabolism. There is no indication that the neuronal and metabolic phenotypes are connected. It is much more likely that the impairment of insulin signaling in the liver or adipose tissue leads to a metabolic phenotype.

Response 1—Please find our responses to the reviewer’s question one statement after another:

- (1) **Quote: “It is not clear why the authors focused on the cerebellum to study the metabolic consequences of ATM mutations in a mouse model.”**

While ATM is traditionally recognized as a pivotal component in the DNA damage response pathway via its role in detecting and signalling double-stranded breaks (Lavin et al., 2006), recent investigations from our laboratory and others have also underscored the critical involvement of the kinase in central carbon metabolism (Dahl and Aird, 2017). This dual role suggests that ATM serves as a significant regulator of metabolic programming in response to various physiological stressors (Aird et al., 2015, Chow et al., 2019a, Schneider et al., 2006).

In the context of A-T, metabolic perturbations manifesting as liver disease (Donath et al., 2019), insulin resistance (Espach et al., 2015), lipid irregularities (Mercer et al., 2010, Paulino et al., 2017) and cardiovascular complications (Andrade et al., 2015) have been documented. An optimal management of these metabolic syndromes correlates not only with clinical amelioration and enhanced survival rates among affected individuals, but also the severity of ataxia (Donath et al., 2019, Paulino et al., 2017). Within the brain where all the cells lack a fully functional ATM, degeneration of Purkinje cells located within the cerebellum are consistently observed among all A-T patients. Our prior investigations have suggested that the selective vulnerability of this specific type of neuron is partly attributed to altered fuel metabolism and energy deficits, as their structural and functional properties render exceptional energy demands as compared to other neuronal cells in the brain, such as granule neurons and pyramidal neurons (Chow et al., 2019a).

These findings have prompted us to delve into the broader metabolic role of ATM under both pathological and physiological conditions, and how deregulations in peripheral metabolic and endocrine status contribute to selective degeneration of Purkinje cells and ataxic phenotype in the disease of A-T. Therefore in this study, our major goal is to elucidate how hypo- or hyper-activity of this kinase could induce systemic metabolic alterations and endocrine changes in the insulin-sensitive tissues, and how that affects the physiology of maturing cerebellum during early childhood

and adolescence. We hoped that via understanding/regulating the overall metabolic status at the peripheral in subjects deficient of ATM could potentially be useful for informing and modulating the health status of cerebellar Purkinje cells.

- (2) **Quote: “Even if insulin signaling is impaired in this brain region, it is unclear whether this would have any consequences on whole-body energy metabolism. There is no indication that the neuronal and metabolic phenotypes are connected. It is much more likely that the impairment of insulin signaling in the liver or adipose tissue leads to a metabolic phenotype.”**

As clarified above, our major goal is indeed to demonstrate that ATM functions as a underappreciated downstream signalling kinase of insulin action that facilitates the rapid anabolic response in cells sensitive towards it. Consequently, the physiological functions of peripheral insulin-sensitive organs, including the adipose tissue, skeletal muscle, and liver; as well as the less acknowledged cerebellum in the brain, are impacted in the absence of ATM. It is essential to clarify that our intention is not to correlate how impaired insulin signalling in the cerebellum affects whole-body energy metabolism. Rather, we emphasize that the loss of ATM in these diverse insulin-sensitive cell types results in a hypercatabolic phenotype that is detected throughout the entire organism.

Comment 2—In the phosphoproteomic paragraph, it was first described that the cerebellum is insulin responsive, then that insulin levels in ATM-KO mice are elevated, and finally, they test a H₂O₂ scavenger. Despite reading this passage ten times, it was impossible to understand the rationale behind the conducted experiments, how they are connected, and what should be the outcome or conclusion.

Response 2—As detailed in **Response 1**, our primary objective was to underscore that ATM is an underappreciated downstream signalling kinase that facilitates a prompt anabolic shift triggered by the extracellular insulin hormone. Consequently, in multiple insulin-sensitive cellular and tissue models related to the disease symptoms, we investigated the molecular mechanism through which ATM becomes activated by insulin-mediated signalling. Drawing insights from the existing literature, it is well-recognized that the kinase itself acts as an oxidative stress sensor within the cytosol (Stagni et al., 2018). We illustrated that insulin stimulation can result in an upsurge of intracellular hydrogen peroxide (H₂O₂) generated by NAD(P)H oxidase (NOX) (**Revised manuscript: Fig.S2d-f**), thereby turning on the downstream signalling cascade of insulin action (Mahadev et al., 2004). Based on these observations, we explored the potential interplay between insulin stimulation, NOX-mediated H₂O₂ production, and subsequent ATM activation (**Revised manuscript: Fig.S2g-h**). The related paragraph has been rewritten and reorganized in the revised manuscript, with updated references as well (**Revised manuscript: Line 207-219**).

Comment 3—Overall, the phosphoproteomic data did not meet any quality criteria of the field. The authors did not provide any quality assessments, information about sample correlations, PCA, number of identified sites, or reproducibility. The text does not contain information about data filtering, normalization procedures, statistical testings, or FDR. The fact that only ~100 regulated sites were found indicates that the workflow was of very poor quality, as many other studies have mapped thousands of regulated sites upon insulin stimulation.

Response 3—In conjunction to criticisms received in **Comment 4** from the same reviewer, we re-performed a new set of phosphoproteomic analysis using the label-free approach to avoid incomplete labelling of non-dividing neurons in the cerebellar tissue culture system with SILAC (Zhang et al., 2014). In this new set of experiment, the updated procedures for protein extraction, digestion, clean up as well as phosphopeptide enrichment can be found in the revised **Materials and Methods** section written in complete details. Both the complete tryptic-digested peptide and the enriched phosphopeptides were introduced into a Vanquish Neo LC system (ThermoFisher) for the purpose of peptide separation. Subsequent to elution, peptides were directly introduced into an Orbitrap Fusion Lumos Mass Spectrometer (ThermoFisher), where mass spectra data were obtained by employing a data-dependent acquisition approach within a scan range of 400-1500 m/z. Detailed specifications of the column, temperature and running conditions of the liquid chromatography step, as well as those related to the Orbitrap performance are now added into the **Materials and Methods** section as well. After the experimental procedures, mass spectrometric data analysis was performed using MaxQuant, a specialized quantitative proteomics software tailored for the analysis of extensive datasets obtained through high-resolution MS, following the previously published workflow and procedures (Tyanova et al., 2016).

The raw data files were analysed using MaxQuant to obtain phosphosite identifications and their respective label-free quantification values using the pre-set standard parameters. Followed by that, the relevant MaxQuant output files, including (1) Phospho(STY)Site.txt and (2) ProteinGroups.txt files were used to run the Phospho-Analyst program: an easy-to-use interactive web-platform that analyses and visualizes phosphoproteomics data pre-processed with MaxQuant (Zhang et al., 2023).

1. For the phosphosite report, the input data were normalized based on the assumption that the majority of phosphosites do not change between the different conditions. Contaminant phosphosites, reverse sequences and phosphosites identified “only by site” were filtered out. In addition, phosphosites with localization probability <0.75 have been removed as well. The phosphosite data was converted to log₂ scale, samples were grouped by conditions and missing values were imputed using “Miss not At Random” (MNAR) method, which uses random draws from a left-shifted Gaussian distribution of 1.8 standard deviation apart with a width of 0.3. Protein-wise linear models combined with empirical Bayes statistics were used for the differential expression analyses. The limma package from R Bioconductor was used to generate a list of differentially expressed phosphosites for each pair-wise comparison. A cutoff of adjusted p-value of 0.05 (Benjamini-Hochberg method) along with a $|\log_2$ fold change| of 1 has been applied to determine significantly regulated phosphosites in each pairwise comparison.
2. For the proteinGroup report, the data from MaxQuant were normalized based on the assumption that the majority of proteins do not change between different conditions. Contaminant proteins, reverse sequences and proteins identified “only by site” were filtered out. In addition, proteins that have been only identified by a single peptide and proteins not identified/quantified consistently in the same condition have been removed as well. The LFQ data was converted to log₂ scale, samples were grouped by conditions and missing values were imputed using the “Miss not At Random” (MNAR) method, as described above. Protein-wise linear models combined with empirical Bayes statistics were used for the differential expression analyses. The limma package from R Bioconductor was used to generate a list of differentially expressed proteins for each

pair-wise comparison. A cutoff of adjusted p-value of 0.05 (Benjamini-Hochberg method) along with a $|\log_2$ fold change| of 1 has been applied to determine significantly regulated phosphosites in each pairwise comparison.

From the analyses, the Phospho-Analyst program revealed that the MaxQuant result output contains 2,320 phosphosites that were reproducibly quantified (**Revised manuscript: Table S1**). Among which, 983 (i.e., 42.3%) phosphosites differ significantly among the treatment groups. Next, as for quality check, the sample coefficient of variation among all treatment groups were found to be similar, with median values ranging between 25-27% (**Revised manuscript: Fig.S3c**). The number of phosphosites quantified in all the samples (except 1) were also similar too (i.e., ~1,500) (**Revised manuscript: Fig.S3b**). Principal component analysis (PCA) revealed that the 4 treatment groups included in our analysis were nicely clustered, having unstimulated samples (i.e., “WT – Glargine” and “KO – Glargine”) being similar to one another while stimulated samples (i.e., “WT + Glargine” and “KO + Glargine”) profiles being more distantly grouped from one another (**Revised manuscript: Fig. 2g**). In the subsequent differential expression analysis, while comparison between unstimulated KO and WT samples (i.e., “KO – Glargine” versus “WT – Glargine”) failed to reveal any meaningful differences in their phosphosite profiles [i.e., only 12 significantly changed (defined as $|\log_2$ fold change| >1 and adjusted p-value <0.05) phosphosites were identified] (**Revised manuscript: Fig.S3f**); the comparison made between insulin-stimulated versus unstimulated samples, regardless of their ATM status, revealed significant differences in their phosphoproteomes (**Revised manuscript: Fig.S3d-e**) (i.e., “KO + Glargine” versus “KO – Glargine”: 648 significantly changed phosphosites; “WT + Glargine” versus “WT – Glargine”: 547 significantly changed phosphosites). Further analyses on the functioning of these hits revealed that proteins derived from upregulated phosphosites that were uniquely found in “WT + Glargine” implicated widely in a set of central carbon metabolic pathways (i.e., glycolysis/gluconeogenesis, pentose phosphate pathway, fructose and mannose metabolism, central carbon metabolism in cancer and HIF-1 signalling pathway) (**Revised manuscript: Fig.S3d, Table S3**). This phenomenon was however not observed among those identified in the “KO + Glargine” group (**Revised manuscript: Fig.S3e, Table S4**). We also conducted direct comparison between phosphoproteome profiles of “KO + Glargine” versus “WT + Glargine” samples, which revealed 74 differentially changed phosphopeptides belonging to 79 proteins (**Revised manuscript: Fig.2h-I, Table S5**). Among which, 54 phosphopeptides which belong to 40 proteins were preferentially enriched in “WT + Glargine” samples, which again implicated glycolysis/gluconeogenesis, central carbon metabolism in cancer, as well as HIF-1 signalling pathway (**Revised manuscript: Fig.2i, Table S5**), that are known drivers of Warburg effect (Courtney et al., 2015, Vander Heiden et al., 2009). In the contrary, those enriched in “KO + Glargine” samples were mostly unrelated to the immediate downstream fate of glucose (**Revised manuscript: Fig.S3i, Table S5**). Next, we also conducted kinase substrate enrichment analysis (KSEA), which revealed that in the absence of ATM (i.e., “KO + Glargine” versus “WT + Glargine”), phosphorylated substrates of kinase downstream to insulin signals, including AKT1 (Kim et al., 1999), MAPK3 and MAPK1 (Lazar et al., 1995) were significantly diminished (**Revised manuscript: Fig.S3h**). More importantly, we also noticed a substantial proportion (i.e., 21/55) of phosphopeptides enriched in “WT + Glargine” were phospho-modified at SQ/TQ residues (**Revised manuscript: Fig.2j**)—the targeted substrate motif of the ATM kinase (Traven and Heierhorst, 2005). These hits included PKM2 and HIF1 α (**Revised manuscript: Fig.2k**), which are the key regulators of the Warburg effect (Yang and Lu, 2013, Courtney et al., 2015).

For the total protein analyses performed in parallel, the Phospho-Analyst program revealed that the MaxQuant result output contains 3,150 proteins that were reproducibly quantified. Among which, only 236 (i.e., 7.5%) proteins differ significantly among the samples (**Revised manuscript: Table S2**). This huge difference from the numbers of significantly changed phosphosites indicated that the majority of changes detected at phosphoproteome level were true and independent from total protein level changes. For quality checking purposes, the sample coefficient of variation among all treatments were again confirmed similar, with median values ranging between 13%-15% (**Revised manuscript: Fig.S4d**). The number of proteins quantified per sample were similar as well (i.e., ~2,500) (**Revised manuscript: Fig.S4c**). PCA revealed that treatment groups were less distinct from one another, in contrast to that pattern observed from phosphosite analyses (**Revised manuscript: Fig.S4a**). In the subsequent differential expression analysis, the number of significantly changed peptides identified from all comparisons in concern were substantially smaller than those obtained from phosphosite analyses (**Revised manuscript: Fig.S4e-f**). Functionally these proteins were not implicated in metabolic pathways and do not interfere with our observations from the phosphoproteome analysis (**Revised manuscript: Fig.S4f**).

Together, these new data provided confirmed the validity of our analyses. All these details are now added to the revised manuscript (**Revised manuscript: Line 221-285**).

Comment 4—It is very surprising that the authors select a 6h treatment time point. Normally, insulin treatment reaches a maximum of singling intensity 1-5 minutes. After 6 h of treatment, changes in the total proteome are induced, which required normalization of phosphorylation to protein levels. A closer examination of the supplemental tables further shows that the authors did not detect most canonical insulin targets. Therefore, pathway analysis or information gain based on these data is not possible. Moreover, it is very strange that, upon KD of a kinase, the number of regulated sites decreases, since it would be expected the other way around. This further points to problems with the data quality. The authors did not conduct any analysis steps, which are normally performed in the proteomics field, such as an analysis of regulatory sites, principal component analyses, or hierarchical clustering.

Response 4—Here below is our response to the reviewer's comment one statement after another:

- (1) Quote: It is very surprising that the authors select a 6h treatment time point. Normally, insulin treatment reaches a maximum of singling intensity 1-5 minutes. After 6 h of treatment, changes in the total proteome are induced, which required normalization of phosphorylation to protein levels.**

We thank the reviewer for this criticism, and we agree with him/her that insulin treatment effect could potentially reach its maximum within minutes. However, considering that upon exposure to any physiological, albeit oral or intraperitoneal load of carbohydrates, a physiological duration of insulin response that typically lasts for around 2 hours would be initiated (Kim et al., 2006, Savage et al., 1975a, Savage et al., 1975b, Dowse et al., 1993). Therefore, it would be imaginable that any insulin-sensitive cells/organs could be exposed to this endocrine hormonal stimulus for this period of time. As we would like to investigate what kind of changes would have occurred after each physiological duration of insulin response in the body, we decided to re-conduct the new label-free phosphoproteomic analysis at 2 hours treatment time point, as

illustrated above in **Response 3**. As elaborated, we also introduced parallel acquisition of the total proteome along with the phosphoproteome as we acknowledge that the treatment time did last longer than the typical time needed for protein expression changes. This arrangement enables assigning changes to the stoichiometry of the phosphorylation site (phosphorylation change relative to protein amount) and the absolute phosphorylation changes resulting from a protein expression change.

- (2) **Quote: A closer examination of the supplemental tables further shows that the authors did not detect most canonical insulin targets. Therefore, pathway analysis or information gain based on these data is not possible. Moreover, it is very strange that, upon KD of a kinase, the number of regulated sites decreases, since it would be expected the other way around. This further points to problems with the data quality.**

We appreciate the reviewer's criticism and suggestions. As we responded above, a new set of phosphoproteomic analyses is now incorporated in the revised manuscript. Our new dataset revealed enhanced phosphorylation of canonical insulin targets when tissues were treated with insulin for 2 hours, these included increased levels of pTSC2 at S939 (Inoki et al., 2002); pARAF at S580 (Kovacina et al., 1990); pPFKL at S775 (Yugi et al., 2014); pACLY at S455 (Yugi et al., 2014) and pMDH1 at S241 (Yugi et al., 2014) (**Revised manuscript: Figure S3g**), and that in reverse an expected reduction in pPDHA1 at S293 after insulin treatment was also observed as well (Hossain et al., 2022). The quality of the data set was further validated now as described in details in **Response 3** already.

Regarding the statement “**it is very strange that, upon KD of a kinase, the number of regulated sites decreases, since it would be expected the other way around**”, we have a different view point that this may not necessarily be the case at all times as it also depends on the specific cellular context and the downstream signalling network of the kinase in concern. Nevertheless, as in the way the reviewer mentioned, our new dataset did reveal that under ATM knockout conditions, insulin treatment resulted in a larger number of phosphorylated peptides detected, as compared to the WT controls (i.e., “KO + Glargine” versus “KO – Glargine”: 648 significantly changed phosphosites; “WT + Glargine” versus “WT – Glargine”: 547 significantly changed phosphosites).

- (3) **Quote: The authors did not conduct any analysis steps, which are normally performed in the proteomics field, such as an analysis of regulatory sites, principal component analyses, or hierarchical clustering.**

As elaborated in **Response 3**, we have already added back these analyses to the revised manuscript.

Comment 5—The authors concluded that any change in phosphorylation in ATM-KD cells is a direct target of ATM. However, this conclusion is not valid since indirect effects can also lead to phosphorylation changes. The authors further modeled in silico protein interactions between phosphorylated PKM2 and an importin and stated that this proves that PKM2 is shuttled to the nucleus induced by ATM phosphorylation. The microscopy images that were supposed to prove this showed a single cell and no image-based quantification. Why were the proteins overexpressed instead of immunostaining for endogenous protein levels?

Response 5— In this context, we aim to respond to each statement in the comment individually:

(1) Quote: “The authors concluded that any change in phosphorylation in ATM-KD cells is a direct target of ATM. However, this conclusion is not valid since indirect effects can also lead to phosphorylation changes.”

We acknowledge the perspective that alterations observed in *Atm-KO* cells, in contrast to *Atm-WT*, may potentially arise from secondary changes in the activities of other kinases downstream to the effect of ATM as well. We adopted the existing knowledge that ATM belongs to the family of phosphatidylinositol kinase-like kinase (PI3KK's) that are serine-threonine kinases that phosphorylate their substrates on SQ or TQ motifs (Traven and Heierhorst, 2005). We also noticed a substantial proportion (i.e., 20/55) of phosphopeptides enriched in “WT + Glargine” were modified at pSQ/TQ residues (**Revised manuscript: Fig.2j**) These hits included PKM2 and HIF1 α (**Revised manuscript: Fig.2k**) implicated commonly in the Warburg effects (Yang and Lu, 2013, Courtney et al., 2015).

Within the mammalian genome, the only other recognized SQ/TQ targeting kinases besides ATM are ATR and DNA-PKcs (Traven and Heierhorst, 2005, Nagasawa et al., 2011), we also added immunoblot analyses, demonstrating that the expression levels of ATR and DNA-PKcs remained unchanged upon exposure to insulin (**Revised manuscript: Fig.S2h, Lanes 1-2**). Subsequently, knockdown of ATM, ATR or DNA-PKcs one at a time in the same experimental system further confirmed that insulin-induced SQ/TQ phosphorylation and nuclear location of PKM2 and HIF α were ATM-specific (**Revised manuscript: Fig.S2h, Lanes 3-5**). This validation was specifically conducted for PKM2 and HIF1 α as they are the only two pSQ/TQ phosphopeptides implicated in “Central carbon metabolism in cancer”, “glycolysis/gluconeogenesis”, HIF-1 signalling pathway” (**Revised manuscript: Fig.2i**).

(2) Quote: The authors further modelled in silico protein interactions between phosphorylated PKM2 and an importin and stated that this proves that PKM2 is shuttled to the nucleus induced by ATM phosphorylation.

This *in silico* finding was validated through the gene knockdown assay as mentioned above (**Revised manuscript: Fig.S2h**). Beyond the phosphorylation at pSQ/TQ sites, it was observed that the transition from tetrameric to dimeric/monomeric protein quaternary structure and subsequent nuclear translocation of PKM2 were distinctly induced upon glargine/insulin-induced ATM activation, rather than relying on other SQ/TQ-targeting kinases (**Revised manuscript: Fig.S2h, Lanes 3-5**).

(3) Quote: The microscopy images that were supposed to prove this showed a single cell and no image-based quantification.

In the revised manuscript, quantifications of the microscopic images in the original Figure 2k have been incorporated (**Revised manuscript: Fig.2l**). In response to "Quote (4)" below, images of endogenous PKM2 along with quantifications of its subcellular localization in Purkinje neurons of cerebellar tissue slices culture are now included in the revised manuscript as well (**Revised manuscript: Fig.2m**).

(4) Quote: Why were the proteins overexpressed instead of immunostaining for endogenous protein levels?

Our initial rationale for employing the overexpression system was to investigate potential disparities in the temporal dynamics required to induce nuclear translocation of PKM2 and HIF1 α (**Revised manuscript: Fig.2l**), as also hinted by the findings from the PKM activity assay and the HIF-response element (HRE) binding assay (**Revised manuscript: Fig.2n-o**). The delay in the appearance of EGFP-HIF1 α signals can be potentially explained by the fact that this protein undergoes regular degradation in the absence of external stimuli, hence it takes more time for it to buildup (Bagnall et al., 2014). Nevertheless, as also a response to the reviewer's request, alterations in the endogenous levels, spatial distribution of PKM2 and HIF1 α , alongside the quantification of subcellular localization signals (**Revised manuscript: Fig.2p-q, S2h**), and immunostaining for endogenous PKM2 in cerebellar slice culture (**Revised manuscript: Fig.2m**) are now added into the revised manuscript.

Comment 6—The analysis of transcriptomic and metabolomics data remains very superficial, and no integration of the two levels occurs. It is only stated that the data suggest some metabolic dysregulation". However, what this really means remains unclear.

Response 6—In response to the reviewer's inquiry, both the metabolomics (**Revised manuscript: Fig.3a, S5a-c, Table S7**) and transcriptomics datasets (**Revised manuscript: Fig.S5d-f, Table S8**) underwent individual re-analysis, followed by a comprehensive dual omics integrated investigation (**Revised manuscript: Fig.3b, Table S8**). Notably, these different analytical approaches yielded the same findings and conclusions as follows:

Global metabolic profiling analysis of mouse cerebellar tissues collected at 3 and 6 months of age unveiled a common set of perturbed metabolites (**Revised manuscript: Fig.S5a-b, Table S7**). Those related to "glycine, serine, and methionine metabolism" were consistently elevated in the absence of ATM (**Revised manuscript: Fig.S5c**), while the majority of metabolites detected were reduced instead (**Revised manuscript: Fig.3b, Table S7**). These downregulated metabolites were implicated in the central glycolytic network (e.g., Warburg effect, gluconeogenesis, glycolysis, pentose phosphate pathway, fructose and mannose degradation), suggestive of a reduced glycolytic flux. The quantities of metabolites indicated in multiple lipid biosynthesis pathways (e.g., phospholipid biosynthesis, *de novo* triacylglycerol biosynthesis, glycerolipid metabolism, α -linoleic acid and linoleic acid biosynthesis, glycerol phosphate shuttle, cardiolipin biosynthesis, phosphatidylethanolamine biosynthesis) which reflects the anabolic capacity of lipid macromolecules were also diminished. Also, those in part of ammonia partitioning (e.g., urea cycle, glutamate metabolism, and ammonia recycling) and the TCA cycle anaplerosis networks (e.g., transfer of acetyl groups into mitochondria, citric acid cycle, malate-aspartate shuttle, mitochondrial electron transport chain, ketone body metabolism, carnitine synthesis, nicotinate and nicotinamide metabolism) were also reduced in the absence of ATM (**Revised manuscript: Fig.3a**). Matching with these, bulk transcriptomic profiling of cerebellar tissues harvested from 6-month-old mice suggested that differentially downregulated genes were involved in glycolysis, adipogenesis (i.e., lipid anabolism), and myogenesis (i.e., protein anabolism) (**Revised manuscript: Fig.S5d-f, Table S8**).

Finally, the gene-metabolite interaction network analysis was conducted on the MetaboAnalyst platform (Pang et al., 2022), using the differentially expressed metabolites as well as the metabolic genes identified from cerebellar tissues of 6 month old KO mice (**Revised**

manuscript: Fig.3b). The results again consistently highlighted a downregulated anabolic network, including the syntheses of amino acids, lipids and lipoproteins, as well as the pyruvate (i.e., the last product of glycolysis) metabolism and TCA cycle.

These consistent findings derived from different datasets and analytical methods collectively hinted at a failure in utilizing glucose as a fuel source at the cellular level, and potentially suggested defects in macromolecule biosynthesis, including adipogenesis and myogenesis at the organ level. These hypotheses were supported by subsequent mitochondrial fuel-dependence tests (**Revised manuscript: Fig.3c and S5g**); and fate tracing analyses using labelled glucose and glutamine (**Revised manuscript: Fig.3d-e**) in *ex vivo* cerebellar slice culture system. These changes, consequently compromised the lipid profile of the cerebellum of the KO mice (**Revised manuscript: Fig.4a**), so as the number of *ALDOC*⁺ Purkinje cells that are inherently more active on the glycolytic network (**Revised manuscript: Fig.4g-n**). The loss of ATM also introduce systemic changes at the peripheral, including reduced wet weights of various fat pads (**Revised manuscript: Fig.1f**), skeletal muscles (**Revised manuscript: Fig.1g**), stunted growth (**Revised manuscript: Fig.1a-b**), as well as compromised blood glucose homeostasis and HOMA-IR status (**Revised manuscript: Fig.1c-d**).

Comment 7—The abstract should be intensively edited since no key message becomes clear

Response 7—The abstract is entirely rewritten and language checking service was used before the resubmission of the revised manuscript.

Comment 8—L46. the statement that insulin induces a Warburg effect should be revised since this refers to the state in cancer cells and not the physiological insulin action.

Response 8— We do agree with the reviewer that the Warburg effect is traditionally characterized by heightened glucose uptake and the conversion of glucose to lactate, even in the presence of functional mitochondria (referred to as aerobic glycolysis) as initially described by Warburg in 1956. (Warburg, 1956). This phenomenon is recognized as one of the six fundamental hallmarks of cancer, as highlighted previously (Vander Heiden et al., 2009, Liberti and Locasale, 2016, Schwartz et al., 2017). However, recent investigations have unveiled additional dimensions of the Warburg effect beyond cancer, encompassing various physiological realms such as immunity, angiogenesis, and pluripotency (Abdel-Haleem et al., 2017, Burns and Manda, 2017). Consequently, drawing from our findings concerning metabolic alterations in response to insulin, while at the same time to avoid confusion, we categorized these changes as “Warburg-like” effects in the revised manuscript.

Comment 9—L.48 if PKM2 moves to the nucleus upon phosphorylation, this is not “spontaneous”

Response 9— The term "spontaneous" is now omitted in the revised manuscript.

Reviewer #4 (Remarks to the Author):

Comment 1—General comments: It is known that the Atm deficiency has little effect on the morphology and the functionality of the cerebral cortex. Atm's effects are specific to certain cells. I recommended that the authors focus on those cells that are highly dependent on Atm. A-T is a progressive disease; thus, it is highly important to follow the metabolic and behavioural changes in young, adult, and aged mice. There is a difference between Atm^{-/-} and Atm inhibitor-treated cells. I highly recommended the use of Atm^{-/-} cells. Inhibitors can have off-target effects that are not detected in Atm^{-/-} cells. Cerebral cortical neurons should be used as controls to compare changes caused by Atm deficiency in cerebellar versus cerebral cells. This comparison can hint at the causality of the metabolites, the pathways, and the treatments. It is thought that A-T has a developmental component. Thus, the use of Atm inhibitor in healthy cells is problematic.

Response 1— Here below is our response to the reviewer's question one statement after another:

- (1) Quote “It is known that the Atm deficiency has little effect on the morphology and the functionality of the cerebral cortex. Atm's effects are specific to certain cells. I recommended that the authors focus on those cells that are highly dependent on Atm” and “Cerebral cortical neurons should be used as controls to compare changes caused by Atm deficiency in cerebellar versus cerebral cells. This comparison can hint at the causality of the metabolites, the pathways, and the treatments.”

Consistent with prior research (Deacon et al., 2023), ATM expression levels throughout the human lifespan revealed heightened levels within the cerebellar cortex in comparison to other brain regions, starting from late embryonic stage (i.e., 200 days of life) (**Revised manuscript: Fig.S2a**).

The observed distinction in ATM expression levels, which indicates a heightened sensitivity of cerebellar cells to ATM loss, may partially account for the pronounced effects within the cerebellum. However, it is important to note that ATM is also expressed in neurons within other regions of the brain, such as cortical neurons (Li et al., 2012, Cheng et al., 2018). This broader distribution of ATM expression across brain regions may elucidate the varying phenotypic manifestations observed in milder forms of A-T, such as the manifestations of cognitive impairments during adulthood (Volkow et al., 2014, Pizzamiglio et al., 2016).

In our manuscript, emphasis was placed on the insulin sensitivity of Purkinje neurons (**Revised manuscript: Fig.S2b**); however, previous research has also underscored the insulin sensitivity of cortical neurons (Chow et al., 2019b). Methodologically, the establishment of cortical neuronal cultures is notably more robust than that of Purkinje neuronal cultures. Consequently, cortical neuronal cultures were employed as the primary cellular model, with cerebellar tissues serving as our *ex vivo* model in our work. Nevertheless, as suggested by **Reviewer 1 (Comment 4)**, we acknowledged in the discussion section that this choice represents a potential limitation of the study.

- (2) **Quote: “A-T is a progressive disease; thus, it is highly important to follow the metabolic and behavioural changes in young, adult, and aged mice.”**

In this investigation, our primary focus was directed towards Young (1-3 months) and Adult (6 months) mice as key subjects for elucidating the impact of ATM on insulin signalling pathways. While we acknowledge the rationale underlying the recommendation put forth by the reviewer, we deliberately excluded aged mice, characterized as those aged between 14 and 24 months (Radulescu et al., 2021). This is due to potential confounding variables associated with advanced age. These factors encompass age-induced hyperinsulinemia and insulin resistance (Chow et al., 2019b), the onset of diverse metabolic perturbations (Houtkooper et al., 2011), as well as diminished ATM functionality in wild-type (WT) animals (Shen et al., 2016). Moreover, the heightened susceptibility to malignancies and other comorbidities in older ATM-deficient mice (Xu et al., 1996) poses further complexities that could obscure our investigation.

- (3) **Quote: “There is a difference between *Atm*^{-/-} and *Atm* inhibitor-treated cells. I highly recommended the use of *Atm*^{-/-} cells. Inhibitors can have off-target effects that are not detected in *Atm*^{-/-} cells” and “Thus, the use of *Atm* inhibitor in healthy cells is problematic.”**

We have gone through the entire manuscript again, it was confirmed that only one experiment involving the ATM inhibitor (KU-60019) was conducted. Consequently, this specific panel has been excluded from the revised submission, and be replaced by gene knockdown experiments (**Revised manuscript: Fig.S2h**).

Comment 2—The normal range of RQ is 0.7-1. Thus, the *Atm*^{-/-} mice are within the normal range.

Response 2— In response to the reviewer's feedback, we wish to underscore the fundamental feature of metabolic flexibility, which is the dynamic modulation of the Respiratory Quotient (RQ) in response to food ingestion.

Mitochondria play a pivotal role in utilizing oxygen to metabolize carbon intermediates obtained from three primary nutrients: glucose, fatty acids, and amino acids (Muoio, 2014). These substrates are eventually broken down to acetyl-CoA, a universal precursor that fuels the Tricarboxylic Acid (TCA) cycle, generating NADH and FADH₂ alongside carbon dioxide (CO₂) for subsequent oxidative phosphorylation reactions (Muoio, 2014). The RQ, denoting the ratio of CO₂ production to oxygen consumption at a cellular level, typically falls within the range of 0.7 to 1.0, as noted by the reviewer, representing an estimate of mitochondrial fuel utilization under standard conditions where amino acids (RQ = 0.8) should play a minor role as oxidative substrates. A high RQ value (~1.0) signifies heightened glucose oxidation, while a low RQ value (~0.7) indicates predominant fat oxidation (Muoio, 2014).

In healthy physiology, there are natural fluctuations in whole-body RQ throughout the day, reflecting a metabolically flexible state where mitochondria adeptly switches between substrates based on systemic nutritional and physiological cues (Muoio, 2014, McGarry, 2002). Conversely, individuals afflicted by metabolic disorders like obesity (Prior et al., 2014), diabetes (Ukropcova et al., 2007) and cardiovascular ailments (Turer et al., 2010) exhibit metabolic inflexibility, marked by impaired fuel switching ability. Thus, the ability to diurnally

shift and variations in whole-body RQ (i.e., the capacity to adjust fuel utilization) in response to dietary patterns and physical activity alterations are regarded as key indicators of metabolic disease status (Muio, 2014), rather than a static RQ value at any single time point.

In our investigation, the KO mice, particularly the 6-month-old cohort, displayed a lack of diurnal RQ oscillation patterns, suggestive of metabolic inflexibility and dysfunction. To elucidate this concept further, relevant figures and accompanying text have been revised in the updated manuscript (**Revised manuscript: Fig.1e, 1j, 5c, 6i, S1c-d, S13d-e, S16c-g**).

Comment 3—Lines 128-138: It seems that 6-month-old *Atm*^{-/-} +STZ and Glargine behave very similarly to WT mice under the same treatment. This has to be discussed.

Response 3— Upon thorough verification of the text and associated figures (**Revised manuscript: Fig.1i-m**), it was ascertained that mice belonging to the KO + STZ + Glargine group (indicated by BROWN labelling in the corresponding figures) exhibited distinct behaviours compared to WT mice subjected to the same treatment regimen (identified by PURPLE labelling in the corresponding figures).

The initial administration of STZ pre-treatment aimed to establish a uniform insulin-deficient baseline across all subjects within the four test groups. By inducing this consistent insulin deficiency with STZ and subsequent administration of an equivalent dosage of glargine, a fair assessment of insulin sensitivity among all test animals could be conducted without the influence of endogenous insulin.

Following an extended exposure to glargine over a 14-day period (spanning from Postnatal day 35 to 49), alterations in the diurnal oscillation patterns of respiratory quotients (**Revised manuscript: Figure 1j**), responses in Intraperitoneal Glucose Tolerance Test (IGTT) (**Revised manuscript: Figure 1k**) and levels of serum triglycerides in the fed state (**Revised manuscript: Figure 1l**) confirmed the inherent condition of insulin resistance in the KO mice. Furthermore, it was observed that the effects of glargine administration in these mice did not mirror the responses observed in WT mice that underwent the same STZ and glargine treatments (PURPLE labelling group). Instead, the responses of the KO + STZ + Glargine group (BROWN labelling group) more closely resembled those of the STZ-treated mice groups (both WT and KO) who did not receive glargine supplementation (ORANGE and CYAN labelling).

To enhance clarity and avoid confusion, the original labelling and colour codes in these figures has been updated as elaborated above in the revised manuscript.

Comment 4—Figure 2e: There are signs of pAtm and Atm in *Atm*-KO mice. This is rather unusual!

Response 4— We regret the oversight and suspect that it may have stemmed from sample contamination during the tissue homogenization process performed by one of our junior trainees. To address this issue, we meticulously prepared new samples and repeated the blotting experiments in the revised manuscript (**Revised manuscript: Fig.2d-e**).

Comment 5—Line 167: If *Atm* is involved in oxidative stress and affected by it, then the authors must show that the *Atm* is not phosphorylated on S1981 but rather forms a dimer through a disulfide bond at C2991.

Response 5— We appreciate the reviewer's comment. In the updated manuscript, examination of ATM via Western Blotting under non-reducing/non-heated denaturing conditions unveiled reduced levels of monomeric ATM following glargine exposure. Additionally, the analysis on the same blot showcased the existence of SDS-resistant ATM dimers under similar treatment conditions (**Revised manuscript: Fig.2e-f; 5g; S2g**).

Comment 6—The data shown in Figs. 2e and f should be quantified and subjected to statistical analyses.

Response 6— Quantitative analysis and statistical evaluations have been reintegrated into the revised manuscript (**Revised manuscript: Fig.2e and 2f**).

Comment 7—Line 175: “...insulin stimulus in primary neurons harvested from WT and Atm-KO mice”. What type of neurons were used?

Response 7— All *in vitro* neuronal experiments were specifically carried out using cerebral cortical neurons. This information has been reinstated in the revised manuscript. Moreover, in alignment with the response to **Comment 1** from the same reviewer, we have also deliberated on the related constraints within the discussion section.

Comment 8—Is ATM involved in the nuclear transportation of Hif1 to the nucleus? This has to be addressed.

Response 8— We are confident in this scenario. In WT conditions, upon insulin/glargine-exposure, ATM likely initiates the phosphorylation of HIF1A to enhance its protein stability (**Revised manuscript: Fig.2l, 5g-h**). Subsequently, the stabilized HIF1 α interacts with the non-tetrameric PKM2 (induced by ATM-mediated phosphorylation of the protein in the absence of the tetrameric protein quaternary structure stabilizer TEPP-46) (Anastasiou et al., 2012) (**Revised manuscript: 5g-h**), which further facilitates HIF1 α translocation to the nucleus (**Revised manuscript: 5g-h**). In the absence of ATM (i.e., KO), these observations were notably absent, underscoring the pivotal role of this kinase in the cascade (**Revised manuscript: Fig.2l, 5g-h**). Additionally, the involvement of the non-tetrameric form of PKM2 in mediating the interaction and nuclear translocation of HIF1 α was accentuated through rescue experiments via the co-treatment with TEPP-46 (Anastasiou et al., 2012) (**Revised manuscript: 5g-h**).

Comment 9—Line 204 and Fig. 5f: The use of the term “equilibrium” when discussing living cells is problematic. All the cellular reactions in the cells are maintained in an energy-dependent steady state, which is a dynamic situation. Equilibrium occurs at steady state and the free energy is zero.

Response 9— We appreciate the reminder from the Reviewer. The relevant text has been removed in the revised manuscript.

Comment 10—Figure 3a (left panel) is not clear and more detailed explanation is needed.

Response 10— Figure 3a and Figure S5a-c present the outcomes of metabolic set enrichment analyses (MSEA) conducted on the commonly up-regulated and down-regulated metabolites identified in cerebellar tissues obtained from KO mice aged 3 and 6 months, after comparing

to their corresponding WT control samples (**Revised manuscript: Fig.S5a-b**). While the metabolites that are commonly upregulated exhibited significant (FDR<0.01) enrichment only in methionine metabolism (**Revised manuscript: Fig.S5c**); the commonly downregulated metabolites displayed enrichment across a set of diverse metabolic pathways (**Revised manuscript: Fig.3a**). These pathways encompassed the central glycolytic network (e.g., Warburg effect, gluconeogenesis, glycolysis, pentose phosphate pathway, fructose and mannose degradation), suggestive of a reduced glycolytic flux. Metabolites indicated in multiple lipid biosynthesis pathways (e.g., phospholipid biosynthesis, *de novo* triacylglycerol biosynthesis, glycerolipid metabolism, α -linoleic acid and linoleic acid biosynthesis, glycerol phosphate shuttle, cardiolipin biosynthesis, phosphatidylethanolamine biosynthesis) were also reduced, indicating diminished anabolic capacity of lipid macromolecules. Also, those involved ammonia partitioning network (e.g., urea cycle, glutamate metabolism, and ammonia recycling) were diminished as well. Finally, those derived-metabolites related to TCA cycle anaplerosis (e.g., transfer of acetyl groups into mitochondria, citric acid cycle, malate-aspartate shuttle, mitochondrial electron transport chain, ketone body metabolism, carnitine synthesis, nicotinate and nicotinamide metabolism) were also reduced under *Atm*-KO conditions (**Revised manuscript: Fig.3a**).

Similarly, analysis of bulk transcriptomic data from cerebellar tissues of 6-month-old mice suggested that differentially downregulated genes were implicated to glycolysis, adipogenesis (the process of fat tissue formation dependent on lipid anabolism), and myogenesis (the formation of skeletal tissues dependent on protein anabolism) (**Revised manuscript: Fig.S5d-f**).

Finally, the gene-metabolite interaction network analysis was conducted on the MetaboAnalyst platform (Pang et al., 2022), using the differentially expressed metabolites as well as the metabolic genes identified from cerebellar tissues of 6 month old KO mice (**Revised manuscript: Fig.3b**). The results again consistently highlighted a downregulated anabolic network, including the syntheses of amino acids, lipids and lipoproteins, as well as the pyruvate (i.e., the last product of glycolysis) metabolism and TCA cycle.

These consistent findings derived from different datasets and analytical methods collectively hinted at a failure in utilizing glucose as a fuel source at the cellular level, and potentially suggested defects in macromolecule biosynthesis, including adipogenesis and myogenesis at the organ level. These hypotheses were supported by subsequent mitochondrial fuel-dependence tests (**Revised manuscript: Fig.3c**); and fate tracing analyses using labelled glucose and glutamine (**Revised manuscript: Fig.3d-e**) in *ex vivo* cerebellar slice culture system. These changes, consequently compromised the lipid profile of the cerebellum of the KO mice (**Revised manuscript: Fig.4a**), so as the number of *ALDOC*⁺ Purkinje cells that are inherently more dependent on the glycolytic network (**Revised manuscript: Fig.4g-n**). The loss of ATM also introduces systemic changes at the peripheral level, including reduced wet weights of various fat pads (**Revised manuscript: Fig.1f**), skeletal muscles (**Revised manuscript: Fig.1g**), impaired growth (**Revised manuscript: Fig.1a-b**), as well as compromised blood glucose homeostasis and HOMA-IR status (**Revised manuscript: Fig.1c-d**) were in the KO mice. These detailed findings have been reinstated in the revised manuscript.

Comment 11—Figure 4b shows western blot analysis and does not really show ALDOC in PCs. The data should be quantified and subjected to statistical analyses. There is high variability in the levels of *Atm*^{+/+} GAD67. What happened in 3-month-old mice? It is

hard to judge the ALDOC levels in Fig. 4f, and it seems that there is immunoreactivity in the white matter of the cerebellum.

Response 11: In accordance with the recommendations of the reviewer, the Western blotting data presented in Fig.4b have now been subjected to quantification (**Revised manuscript: Fig.S8a**). Notably, a comparable trend of alterations in ALDOC and GAD67 was observed among 6-month-old mice, albeit to a lesser extent than the 3-month-old ones (**Response: Fig.4b, S8a**). Regarding immunostaining, we conducted a repeat of the staining procedures and verified that ALDOC signals were localized robustly in Purkinje cells, with a notable reduction observed in the number of these cells in KO animals (**Revised manuscript Fig. 4e, S9**). As highlighted by the reviewer, there was a slight detection of ALDOC immunoreactivity in the white matter regions of the cerebellum, which was also diminished (**Revised manuscript Fig. 4e, S9**). According to existing literature (Sillitoe et al., 2003), these observations are likely partially attributed to the axons of ALDOC-positive Purkinje cells.

Comment 12—Line 317: The rationale of the addition of TEPP-46 should be explained. It would be helpful to show the effect of TEPP-46 on *Atm*^{-/-} mice.

Response 12— According to the published literature (Anastasiou et al., 2012), TEPP-46 (also known as ML265, PubChem CID: 44246499) belongs to the thieno[3,2-b]pyrrole[3,2-d]pyridazinones class of selective PKM2 activators. In a manner akin to fructose-1,6-bisphosphate (FBP), the endogenous allosteric activator known to stabilize PKM2 in the tetrameric state (Lv et al., 2013, Dombrauckas et al., 2005), TEPP-46 decreases the K_m of PKM2 for PEP without affecting the K_m for ADP. This suggests that TEPP-46 activates PKM2 through a mechanism similar to that of FBP (Anastasiou et al., 2012, Jiang et al., 2010). TEPP-46 has pharmacokinetic properties conducive to experiments in mice, with mouse plasma drug concentration measured over time after a single intravenous, intraperitoneal, or oral dose showing relatively low clearance and a long half-life (Anastasiou et al., 2012).

In the revised manuscript, we have included data obtained from KO mice treated with different regimens [i.e., “KO + vehicle (s.c.daily) + vehicle (i.p./daily)”]; “KO + Glargine (s.c./daily) + vehicle (i.p./daily)” or “KO + Glargine (s.c./daily) + TEPP46 (i.p.)”] (**Revised manuscript: Fig.5**). In summary, the body weight of KO animals generally did not respond to the glargine (s.c./daily) stimulus, unlike the weight gain observed in WT animals rescued with glargine [WT + Glargine (s.c./daily)] (**Revised manuscript: Fig.5b**). This suggests a crucial role of ATM in mediating the anabolic effects of insulin (glargine). Consequently, the counteractive effect of TEPP-46 on glargine observed in WT mice was not observed in the KO group either (**Revised manuscript: Fig.5b**).

To assess their metabolic flexibility status, we calculated the dynamics of respiratory quotients (RQs) in response to changing energy demand and nutrient supply, a measure that may indicate inflexibility under conditions of insulin resistance or hyperinsulinemia (Galgani et al., 2008) (**For details please refer to Response 2 again**). As expected, while WT subjects on glargine showed consistent alternations in RQs during both light and dark cycles, all respiratory quotients values recorded from KO subjects however remained similar, indicating metabolic inflexibility (**Revised manuscript: Fig.5c**). The alternating RQs of WT subjects on glargine indicated shifting usage of carbohydrate and fat (RQ = 0.8-0.9); whilst the average respiratory quotients of KO subjects remained stagnant at most time, indicating a sustained usage of a mix of fats and proteins instead (RQ = 0.7-0.8) (Muoio, 2014) (**Revised manuscript: Fig.5c**).

Comment 13—Figure 5a-c should include Atm-/- alone.

Response 13— In line with our reply to **Comment 12** from the same reviewer above, we have incorporated investigation concerning the KO group alone [i.e., KO + vehicle (s.c./daily) + vehicle (i.p./daily)] in the revised manuscript (**Revised manuscript: Fig.5**).

Comment 14—Fig. 5g: The western blot should be quantified and subjected to statistical analysis. Surprisingly, Atm-/- mice appear to express Atm! If glargine can induce the expression of Atm in Atm-/- mice, it is a very unexpected result! Also, there is a background of pAtm in Atm-/- mice. This has to be explained.

Response 14—In a manner akin to our handling of **Comment 4** from the same reviewer, we acknowledge and regret the oversight made, attributing it potentially to sample contamination during the tissue homogenization process carried out by one of our junior trainees. To address this issue, we meticulously prepared fresh samples with enhanced care and repeated the blotting procedures in the revised manuscript. Subsequently, all blots were quantified and analysed statistically as indicated (**Revised manuscript: Fig.S13f**).

Comment 15—A-T patients do not display cerebral cortex abnormalities. Thus, if the cerebral cortex is also insulin sensitive, I have doubts about the causality of the insulin effects.

Response 15— This comment parallels the one expressed in **Comment 1** from the same reviewer. As detailed in **Response 1**, although cerebellar neurons exhibit heightened sensitivity to the absence of ATM, it is worth noting that ATM is present in neurons across various brain regions, including cortical neurons (Li et al., 2012, Cheng et al., 2018). This broader neuronal distribution of ATM may elucidate the variable phenotypic manifestations observed in milder forms of A-T concerning cognitive impairments during adulthood (Volkow et al., 2014, Pizzamiglio et al., 2016).

While our current manuscript underscores the insulin sensitivity of Purkinje neurons (**Revised manuscript: Fig.S2a-c**), our prior research has also demonstrated the insulin sensitivity of cortical neurons (Chow et al., 2019b). We hypothesize that there might exist varying degrees of resilience against insulin resistance between cerebral cortex neurons and those in the cerebellum, attributed to their reliance on insulin receptor-mediated signalling pathways. Notably, the spatial and temporal expression levels of the insulin receptor (*INSR*) in the human brain (Kang et al., 2011) reveal significantly higher levels in the cerebellar cortex (CBC) postnatally as compared to other brain regions (**Revised manuscript: Fig.S2a**). This finding implies the critical role of *INSR*-mediated insulin signalling in the maturation of the cerebellar cortex. Given that ATM expression mirrors the trend of *INSR* expression (**Revised manuscript: Fig.S2a**), these factors combined may contribute to the specific susceptibility of the cerebellum to disruptions in both ATM and insulin signalling in the context of A-T.

Comment 16—Hif1 is upregulated by hypoxia. Is there any connection between glargine and hypoxia?

Response 16— Several studies in the literature have indicated that HIF-1 plays a role in biological processes that necessitate its activation even under normoxic conditions (Dery et al., 2005). Notably, concerning insulin, specifically glargine, various investigations conducted by different research groups (Feldser et al., 1999, Zelzer et al., 1998) have shown that insulin can

elevate HIF1 α protein levels, enhance DNA binding activities, and induce the expression of its target genes, including those responsible for glucose transporters and glycolytic enzymes, without the induction of hypoxia (Feldser et al., 1999, Zelzer et al., 1998).

While the precise mechanisms underlying these effects have not been extensively explored in these studies (with our work proposing a potential involvement of ATM), other research focusing on factors that induce HIF1 α in the absence of hypoxia (such as mersalyl) has suggested the potential involvement of certain cytosolic kinases, notably mitogen-activated protein kinase (MAPK) activity (Agani and Semenza, 1998), a potential downstream target of ATM (Kim et al., 2007), as also hinted from our phosphoproteomic analyses that insulin-induced activation of MAPK was lost in KO condition (**Revised manuscript: Fig.S3h**). Similarly, our observations regarding insulin-triggered ATM activation, subsequent phosphorylation, and resultant HIF1 α activation provide further insight into the regulatory influence of “cytosolic kinases” on HIF1 α stability and function within a physiological context (**Revised manuscript: Fig.2l, 5g-h**).

Comment 17—As shown in Fig. 5m, glargine treatments failed to induce any positive effect on open field test results in WT and *Atm*^{-/-} mice! What can we conclude from that?

Response 17—In response to the aforementioned critique, we seek to expound further on the aims of the treatments in question. Glargine, being a stabilized insulin analogue characterized by a duration of action spanning up to 24 hours without pronounced peaks or oscillations (Hilgenfeld et al., 2014), is integral to the discussion. Prolonged exposure to an excess of this agent in mice subjected to a standard chow diet serves to replicate the state of hyperinsulinemia, a precursor to diabetes (Yang et al., 2014, Chow et al., 2019b, Sarmiento-Ortega et al., 2023).

Figure 5 of our study conveys a crucial message regarding the significance of a flexibly on-and-off status of insulin-ATM activity cycle in upholding metabolic homeostasis. To elaborate, aberrations in this flexibility in the insulin-ATM signalling axis, such as those observed in *Atm*-KOs (i.e. a persistently “signal off” status) or in models of hyperinsulinemia in wild-type (WT) mice (i.e., a persistently “signal on” status), present an inflexible manifestation. These deviations respectively correlate with suboptimal anabolic state that results in stunt growth versus excessive anabolism that leads to excessive body weight gain and obesity.

Our research posits that, under normal circumstances, a transient postprandial surge in insulin triggers ATM activation, leading to the phosphorylation of key catabolic-and-anabolic regulators like PKM2 and HIF1 α (**Revised manuscript: Fig.2g-m**). This physiological period of insulin effect (~2 hours) (Kim et al., 2006, Savage et al., 1975a, Savage et al., 1975b, Dowse et al., 1993), promptly inhibits PKM2's metabolic actions by disrupting its tetrameric structure (**Revised manuscript: Fig.2q, S2h**), thereby impeding the predominant contribution of glucose carbons to pyruvate and their subsequent entry into the TCA cycle (**Revised manuscript: Fig.3a-e**). Consequently, these diverted carbons of glucose contributes to *de novo* lipogenesis that could be diverted to along the glycolytic pathway (**Revised manuscript: Fig.S5h**). Within this physiological time frame (~2 hours) of insulin action, HIF1 α 's nuclear activity remains modest during this physiological context (**Revised manuscript: Fig.2l, 2o**). However, in pathological scenarios like hyperinsulinemia, prolonged insulin exposure to the cells leads to sustained activation of ATM, not only this results in changes to the PKM2 tetrameric status as mentioned above, but also allows time for the monomeric version of the protein to induce a “nuclear moonlighting effect”. This effect, in turn, co-activates HIF1 α 's nuclear activities, fostering a pro-lipogenic cascade via upregulating the expression levels of

multiple lipogenic enzymes and hence lipid biosynthesis (**Revised manuscript: Fig.5f-m**). The requirement for PKM2's monomeric moonlighting effect on HIF1 α 's nuclear activities, incited by chronic insulin (glargine) stimuli, was mitigated by TEPP-46—a small molecule that promotes PKM2 tetramer stability and its metabolic activities (Witney et al., 2015, Yi et al., 2020) (**Revised manuscript: Fig.5**) (**For more information of TEPP-46, please refer to Response to Comment 12**). This intervention curtailed the nuclear moonlighting and HIF1 co-activator role of PKM2, which aligned with prior findings (Yi et al., 2020). Notably, in the KO mice, these ATM-mediated effect on PKM2 and HIF1 α were virtually absent, rendering treatments comprising glargine alone or in combination with TEPP-46 indistinguishable from vehicle-treated animals (**Revised manuscript: Fig.5**).

Indeed, the total PKM metabolic activities in KO subjects were substantially lower (**Revised manuscript: Fig.5i**) as the metabolic kinase predominantly existed in the dimeric form (**Revised manuscript: Fig.3f, 5g**). Despite so, administration of TEPP-46 failed to boost the total PKM metabolic activities in these animals (**Revised manuscript: Fig.5i**) or rescue cerebellar metabolite profiles (**Revised manuscript: Fig.5l-m**). We reasoned that this could be in part attributed to the “cytosolic moonlighting effect” of the PKM2 dimer (**Revised manuscript: Fig.3h-k**), where it interacts with c-Myc IRES-RNA to facilitate IRES-dependent translation to promote glutaminolysis and glutamine dependence in ATM-deficient cells (**Revised manuscript: Fig.3h-k**) (Li et al., 2020). Taken together, these observations elucidate why treatments involving glargine alone or in conjunction with TEPP-46 elicited no discernible effects on the motor and coordination functions of KO mice (**Revised manuscript: Fig.5n**).

In the WT mice, prolonged exposure to continuous insulin through glargine administration resulted in robust pro-lipogenic reprogramming (**Revised manuscript: Fig.5d-n**), metabolic inflexibility (**Revised manuscript: Fig.5c**), and body weight gain (**Revised manuscript: Fig.5b**). However, no significant correlations were found between such phenomenon with cerebellar atrophy or Purkinje cell loss. Conversely, several studies conducted by independent research teams have instead highlighted a link between low body mass index and underweight conditions with the severity and progression of cerebellar ataxia in genetic contexts (Diallo et al., 2017, Guo et al., 2022a, Almaguer-Mederos et al., 2021, Ronnefarth et al., 2020, Yang et al., 2018). These findings supported that the cerebellum is particularly more responsive to lipid insufficiency, which could be resulted from impaired insulin signaling, rather than the converse scenario of sustained insulin signalling activation.

Comment 18—It would be informative to determine the cause of death of the *Atm*^{-/-} mice and to understand the effect of alfa-KG on those pathways that lead to the death of *Atm*^{-/-} mice.

Response 18—In accordance with existing literature, respiratory ailments stand out as a leading cause of morbidity and mortality in A-T patients, who are susceptible to recurrent sinopulmonary infections, bronchiectasis, pulmonary fibrosis, and pulmonary failure (Yeo et al., 2019). The primary driver of such outcomes is often immune deficiency stemming from the absence of ATM (Yeo et al., 2019). Additionally, coexisting conditions like underweight status and malnutrition have been identified as significant risk factors for infections in A-T patients (Dobner and Kaser, 2018, Moser et al., 2019).

While our research team lacks specific expertise in monitoring pulmonary infections, we noted a strong association between low body weight gain and premature mortality in *Atm*-KO animals that were fed a standard chow diet (**Revised manuscript: Fig.6f-g**). Conversely, within our

randomized cohort of KO mice (designated as "KO + vehicle" and "KO + 2% CaAKG") with comparable baselines in terms of pre-treatment blood glucose responses and body weights (**Revised manuscript: Fig.S16a-b**), those receiving α -KG supplementation exhibited enhanced overall weight gain, leading to reduced occurrences of premature mortality or sudden death (**Revised manuscript: Fig.6f-g**).

Comment 19—Fig. S14 does not provide any data regarding preservation of PCs! It would be very helpful if this could be shown in the main figures.

Response 19— The suggested modifications have been incorporated by reintegrating the relevant data into the primary and supplementary figures of the revised manuscript (**Revised manuscript: Fig.6n, S17**). Analysis of the staining data also demonstrated superior preservation of IP3R1-labeled Purkinje cells (PCs) and VGlut2 densities (**Revised manuscript: Fig.S18**) in KO mice subjected to 2% CaAKG treatment compared to those receiving vehicle controls.

Comment 20—Statistical analyses: The authors assumed that all their data are distributed normally. This must be tested for each experiment, and the data should be analysed using the proper statistical tests (parametric or non-parametric).

Response 20— Within the revised manuscript, an assessment of the normality of all datasets was conducted, with this analysis documented in the source data file. In instances where datasets exhibited non-normal distributions, non-parametric statistical tests were employed for further statistical evaluations. Details regarding these analyses were revised and included in the corresponding figure legends, source data files and the "Quantification procedures and statistical analyses" section within the **Methods and Materials** section.

Comment 21—Protocols for cerebellar cell cultures are not well presented. It is not clear which kind of cultures were used. This has to be clarified.

Response 21— In response to the aforementioned feedback, we initially presumed that the reviewer's reference was to the "acute cerebellar organotypic brain slice culture," for which we now meticulously outlined a protocol in the **Methods and Materials** section.

Should our interpretation have been erroneous (specifically, if the inquiry is related to a cerebellar neuronal culture protocol), we wish to clarify that we did not conduct any experiments involving primary neuronal cultures derived from cerebellar tissues. Rather, in experiments necessitating primary neurons, these cells were sourced from E16 embryonic cortical tissues, a decision previously expounded upon (**refer to Reviewer #1 Comment 1; Reviewer #4 Comments 1 and 15**). Concurrently, as suggested by **Reviewer #1**, we have also acknowledged and addressed the potential constraints associated with utilizing heterogeneous systems in the revised iteration of the manuscript.

Comment 22—Figure 1i: The colours are confusing! Sometimes green indicates WT and sometimes KO. Be consistent to prevent confusion.

Response 22— In addressing this feedback, modifications have been made to the colour codes in **Figure 1i**, along with similar adjustments in **Figures 5a and 6e**. Furthermore, additional labelling has been incorporated on the figures to specify the corresponding figure panels to

which the respective colour scheme pertains. These enhancements aim to mitigate any potential confusion among readers.

Comment 23—Figs 2i and j are not data but demonstration and can be moved to the supplementary section.

Response 23—Both figure panels are now re-allocated to the supplementary section in the revised manuscript (**Revised manuscript: Fig. S2i-j**).

Reference list

- ABDEL-HALEEM, A. M., LEWIS, N. E., JAMSHIDI, N., MINETA, K., GAO, X. & GOJOBORI, T. 2017. The Emerging Facets of Non-Cancerous Warburg Effect. *Front Endocrinol (Lausanne)*, 8, 279.
- AGANI, F. & SEMENZA, G. L. 1998. Mersalyl is a novel inducer of vascular endothelial growth factor gene expression and hypoxia-inducible factor 1 activity. *Mol Pharmacol*, 54, 749-54.
- AIRD, K. M., WORTH, A. J., SNYDER, N. W., LEE, J. V., SIVANAND, S., LIU, Q., BLAIR, I. A., WELLEN, K. E. & ZHANG, R. 2015. ATM couples replication stress and metabolic reprogramming during cellular senescence. *Cell Rep*, 11, 893-901.
- ALMAGUER-MEDEROS, L. E., PEREZ-AVILA, I., AGUILERA-RODRIGUEZ, R., VELAZQUEZ-GARCES, M., ALMAGUER-GOTAY, D., HECHAVARRIA-PUPO, R., RODRIGUEZ-ESTUPINAN, A. & AUBURGER, G. 2021. Body Mass Index Is Significantly Associated With Disease Severity in Spinocerebellar Ataxia Type 2 Patients. *Mov Disord*, 36, 1372-1380.
- ANASTASIOU, D., YU, Y., ISRAELSEN, W. J., JIANG, J. K., BOXER, M. B., HONG, B. S., TEMPEL, W., DIMOV, S., SHEN, M., JHA, A., YANG, H., MATTAINI, K. R., METALLO, C. M., FISKE, B. P., COURTNEY, K. D., MALSTROM, S., KHAN, T. M., KUNG, C., SKOUMBOURDIS, A. P., VEITH, H., SOUTHALL, N., WALSH, M. J., BRIMACOMBE, K. R., LEISTER, W., LUNT, S. Y., JOHNSON, Z. R., YEN, K. E., KUNII, K., DAVIDSON, S. M., CHRISTOFK, H. R., AUSTIN, C. P., INGLESE, J., HARRIS, M. H., ASARA, J. M., STEPHANOPOULOS, G., SALITURO, F. G., JIN, S., DANG, L., AULD, D. S., PARK, H. W., CANTLEY, L. C., THOMAS, C. J. & VANDER HEIDEN, M. G. 2012. Pyruvate kinase M2 activators promote tetramer formation and suppress tumorigenesis. *Nat Chem Biol*, 8, 839-47.
- ANDRADE, I. G. A., COSTA-CARVALHO, B. T., DA SILVA, R., HIX, S., KOCHI, C., SUANO-SOUZA, F. I. & SARNI, R. O. S. 2015. Risk of Atherosclerosis in Patients with Ataxia Telangiectasia. *Ann Nutr Metab*, 66, 196-201.
- ANGLIN, J., ZAVAREH, R. B., SANDER, P. N., HALDAR, D., MULLARKY, E., CANTLEY, L. C., KIMMELMAN, A. C., LYSSIOTIS, C. A. & LAIRSON, L. L. 2018. Discovery and optimization of aspartate aminotransferase 1 inhibitors to target redox balance in pancreatic ductal adenocarcinoma. *Bioorg Med Chem Lett*, 28, 2675-2678.
- ASADI SHAHMIRZADI, A., EDGAR, D., LIAO, C. Y., HSU, Y. M., LUCANIC, M., ASADI SHAHMIRZADI, A., WILEY, C. D., GAN, G., KIM, D. E., KASLER, H. G., KUEHNEMANN, C., KAPLOWITZ, B., BHAUMIK, D., RILEY, R. R., KENNEDY, B. K. & LITHGOW, G. J. 2020. Alpha-Ketoglutarate, an Endogenous Metabolite, Extends Lifespan and Compresses Morbidity in Aging Mice. *Cell Metab*, 32, 447-456 e6.
- BAGNALL, J., LEEDALE, J., TAYLOR, S. E., SPILLER, D. G., WHITE, M. R., SHARKEY, K. J., BEARON, R. N. & SEE, V. 2014. Tight control of hypoxia-inducible factor- α transient dynamics is essential for cell survival in hypoxia. *J Biol Chem*, 289, 5549-64.

- BAKSH, S. C. & FINLEY, L. W. S. 2021. Metabolic Coordination of Cell Fate by alpha-Ketoglutarate-Dependent Dioxygenases. *Trends Cell Biol*, 31, 24-36.
- BLOCH, C. A., CLEMONS, P. & SPERLING, M. A. 1987. Puberty decreases insulin sensitivity. *J Pediatr*, 110, 481-7.
- BURNS, J. S. & MANDA, G. 2017. Metabolic Pathways of the Warburg Effect in Health and Disease: Perspectives of Choice, Chain or Chance. *Int J Mol Sci*, 18.
- CAPRIO, S., PLEWE, G., DIAMOND, M. P., SIMONSON, D. C., BOULWARE, S. D., SHERWIN, R. S. & TAMBORLANE, W. V. 1989. Increased insulin secretion in puberty: a compensatory response to reductions in insulin sensitivity. *J Pediatr*, 114, 963-7.
- CHENG, A., ZHAO, T., TSE, K. H., CHOW, H. M., CUI, Y., JIANG, L., DU, S., LOY, M. M. T. & HERRUP, K. 2018. ATM and ATR play complementary roles in the behavior of excitatory and inhibitory vesicle populations. *Proc Natl Acad Sci U S A*, 115, E292-E301.
- CHOW, H. M., CHENG, A., SONG, X., SWERDEL, M. R., HART, R. P. & HERRUP, K. 2019a. ATM is activated by ATP depletion and modulates mitochondrial function through NRF1. *J Cell Biol*, 218, 909-928.
- CHOW, H. M., SHI, M., CHENG, A., GAO, Y., CHEN, G., SONG, X., SO, R. W. L., ZHANG, J. & HERRUP, K. 2019b. Age-related hyperinsulinemia leads to insulin resistance in neurons and cell-cycle-induced senescence. *Nat Neurosci*, 22, 1806-1819.
- COURTNAY, R., NGO, D. C., MALIK, N., VERVERIS, K., TORTORELLA, S. M. & KARAGIANNIS, T. C. 2015. Cancer metabolism and the Warburg effect: the role of HIF-1 and PI3K. *Mol Biol Rep*, 42, 841-51.
- CRUZAT, V., MACEDO ROGERO, M., NOEL KEANE, K., CURI, R. & NEWSHOLME, P. 2018. Glutamine: Metabolism and Immune Function, Supplementation and Clinical Translation. *Nutrients*, 10.
- DABEK, M., KRUSZEWSKA, D., FILIP, R., HOTOWY, A., PIERZYNOWSKI, L., WOJTASZ-PAJAK, A., SZYMANCZYK, S., VALVERDE PIEDRA, J. L., WERPACHOWSKA, E. & PIERZYNOWSKI, S. G. 2005. alpha-Ketoglutarate (AKG) absorption from pig intestine and plasma pharmacokinetics. *J Anim Physiol Anim Nutr (Berl)*, 89, 419-26.
- DAHL, E. S. & AIRD, K. M. 2017. Ataxia-Telangiectasia Mutated Modulation of Carbon Metabolism in Cancer. *Front Oncol*, 7, 291.
- DANN, C. E., 3RD, BRUICK, R. K. & DEISENHOFER, J. 2002. Structure of factor-inhibiting hypoxia-inducible factor 1: An asparaginyl hydroxylase involved in the hypoxic response pathway. *Proc Natl Acad Sci U S A*, 99, 15351-6.
- DEACON, S., DALLEYWATER, W., PEAT, C., PAINE, S. M. L. & DINEEN, R. A. 2023. Disproportionate Expression of ATM in Cerebellar Cortex During Human Neurodevelopment. *Cerebellum*.
- DERY, M. A., MICHAUD, M. D. & RICHARD, D. E. 2005. Hypoxia-inducible factor 1: regulation by hypoxic and non-hypoxic activators. *Int J Biochem Cell Biol*, 37, 535-40.
- DIALLO, A., JACOBI, H., SCHMITZ-HUBSCH, T., COOK, A., LABRUM, R., DURR, A., BRICE, A., CHARLES, P., MARELLI, C., MARIOTTI, C., NANETTI, L., PANZERI, M., RAKOWICZ, M., SOBANSKA, A., SULEK, A., SCHOLS, L., HENGEL, H., MELEGH, B., FILLA, A., ANTENORA, A., INFANTE, J., BERCIANO, J., VAN DE WARRENBURG, B. P., TIMMANN, D., BOESCH, S., PANDOLFO, M., SCHULZ, J. B., BAUER, P., GIUNTI, P., BALIKO, L., PARKINSON, M. H., KANG, J. S., KLOCKGETHER, T. & TEZENAS DU MONTCEL, S. 2017. Body Mass Index Decline Is Related to Spinocerebellar Ataxia Disease Progression. *Mov Disord Clin Pract*, 4, 689-697.

- DOBNER, J. & KASER, S. 2018. Body mass index and the risk of infection - from underweight to obesity. *Clin Microbiol Infect*, 24, 24-28.
- DOMBRAUCKAS, J. D., SANTARSIERO, B. D. & MESECAR, A. D. 2005. Structural basis for tumor pyruvate kinase M2 allosteric regulation and catalysis. *Biochemistry*, 44, 9417-29.
- DONATH, H., WOELKE, S., THEIS, M., HESS, U., KNOP, V., HERRMANN, E., KRAUSKOPF, D., KIESLICH, M., SCHUBERT, R. & ZIELEN, S. 2019. Progressive Liver Disease in Patients With Ataxia Telangiectasia. *Front Pediatr*, 7, 458.
- DOWSE, G. K., ZIMMET, P. Z., ALBERTI, K. G., BRIGHAM, L., CARLIN, J. B., TUOMILEHTO, J., KNIGHT, L. T. & GAREEBOO, H. 1993. Serum insulin distributions and reproducibility of the relationship between 2-hour insulin and plasma glucose levels in Asian Indian, Creole, and Chinese Mauritians. Mauritius NCD Study Group. *Metabolism*, 42, 1232-41.
- ESPACH, Y., LOCHNER, A., STRIJDOM, H. & HUISAMEN, B. 2015. ATM protein kinase signaling, type 2 diabetes and cardiovascular disease. *Cardiovasc Drugs Ther*, 29, 51-8.
- FELDSER, D., AGANI, F., IYER, N. V., PAK, B., FERREIRA, G. & SEMENZA, G. L. 1999. Reciprocal positive regulation of hypoxia-inducible factor 1 α and insulin-like growth factor 2. *Cancer Res*, 59, 3915-8.
- FURMAN, B. L. 2015. Streptozotocin-Induced Diabetic Models in Mice and Rats. *Curr Protoc Pharmacol*, 70, 5 47 1-5 47 20.
- FURMAN, B. L. 2021. Streptozotocin-Induced Diabetic Models in Mice and Rats. *Curr Protoc*, 1, e78.
- GALGANI, J. E., MORO, C. & RAVUSSIN, E. 2008. Metabolic flexibility and insulin resistance. *Am J Physiol Endocrinol Metab*, 295, E1009-17.
- GUO, J., JIANG, Z., BISWAL, B. B., ZHOU, B., XIE, D., GAO, Q., SHENG, W., CHEN, H., ZHANG, Y., FAN, Y., WANG, J., LIU, C. & CHEN, H. 2022a. Hypothalamic Atrophy, Expanded CAG Repeat, and Low Body Mass Index in Spinocerebellar Ataxia Type 3. *Mov Disord*, 37, 1541-1546.
- GUO, L., CHEN, S., OU, L., LI, S., YE, Z. N. & LIU, H. F. 2022b. Disrupted Alpha-Ketoglutarate Homeostasis: Understanding Kidney Diseases from the View of Metabolism and Beyond. *Diabetes Metab Syndr Obes*, 15, 1961-1974.
- HILGENFELD, R., SEIPKE, G., BERCHTOLD, H. & OWENS, D. R. 2014. The evolution of insulin glargine and its continuing contribution to diabetes care. *Drugs*, 74, 911-27.
- HOANG, M. & JOSEPH, J. W. 2020. The role of alpha-ketoglutarate and the hypoxia sensing pathway in the regulation of pancreatic beta-cell function. *Islets*, 12, 108-119.
- HOSSAIN, A. J., ISLAM, R., KIM, J. G., DOGSOM, O., CAP, K. C. & PARK, J. B. 2022. Pyruvate Dehydrogenase A1 Phosphorylated by Insulin Associates with Pyruvate Kinase M2 and Induces LINC00273 through Histone Acetylation. *Biomedicines*, 10.
- HOUTKOOPER, R. H., ARGMAN, C., HOUTEN, S. M., CANTO, C., JENINGA, E. H., ANDREUX, P. A., THOMAS, C., DOENLEN, R., SCHOONJANS, K. & AUWERX, J. 2011. The metabolic footprint of aging in mice. *Sci Rep*, 1, 134.
- INOKI, K., LI, Y., ZHU, T., WU, J. & GUAN, K. L. 2002. TSC2 is phosphorylated and inhibited by Akt and suppresses mTOR signalling. *Nat Cell Biol*, 4, 648-57.
- JANG, C., CHEN, L. & RABINOWITZ, J. D. 2018. Metabolomics and Isotope Tracing. *Cell*, 173, 822-837.

- JIANG, J. K., BOXER, M. B., VANDER HEIDEN, M. G., SHEN, M., SKOUMBOURDIS, A. P., SOUTHALL, N., VEITH, H., LEISTER, W., AUSTIN, C. P., PARK, H. W., INGLESE, J., CANTLEY, L. C., AULD, D. S. & THOMAS, C. J. 2010. Evaluation of thieno[3,2-b]pyrrole[3,2-d]pyridazinones as activators of the tumor cell specific M2 isoform of pyruvate kinase. *Bioorg Med Chem Lett*, 20, 3387-93.
- KANG, H. J., KAWASAWA, Y. I., CHENG, F., ZHU, Y., XU, X., LI, M., SOUSA, A. M., PLETIKOS, M., MEYER, K. A., SEDMAK, G., GUENNEL, T., SHIN, Y., JOHNSON, M. B., KRŠNIK, Z., MAYER, S., FERTUZHOS, S., UMLAUF, S., LISGO, S. N., VORTMEYER, A., WEINBERGER, D. R., MANE, S., HYDE, T. M., HUTTNER, A., REIMERS, M., KLEINMAN, J. E. & SESTAN, N. 2011. Spatio-temporal transcriptome of the human brain. *Nature*, 478, 483-9.
- KIM, B., JANG, C., DHARANEESWARAN, H., LI, J., BHIDE, M., YANG, S., LI, K. & ARANY, Z. 2018. Endothelial pyruvate kinase M2 maintains vascular integrity. *J Clin Invest*, 128, 4543-4556.
- KIM, J. A., MONTAGNANI, M., KOH, K. K. & QUON, M. J. 2006. Reciprocal relationships between insulin resistance and endothelial dysfunction: molecular and pathophysiological mechanisms. *Circulation*, 113, 1888-904.
- KIM, W. J., RAJASEKARAN, B. & BROWN, K. D. 2007. MLH1- and ATM-dependent MAPK signaling is activated through c-Abl in response to the alkylator N-methyl-N'-nitro-N'-nitrosoguanidine. *J Biol Chem*, 282, 32021-31.
- KIM, Y. B., NIKOULINA, S. E., CIARALDI, T. P., HENRY, R. R. & KAHN, B. B. 1999. Normal insulin-dependent activation of Akt/protein kinase B, with diminished activation of phosphoinositide 3-kinase, in muscle in type 2 diabetes. *J Clin Invest*, 104, 733-41.
- KIM, Y. M., KIM, S. J., TATSUNAMI, R., YAMAMURA, H., FUKAI, T. & USHIO-FUKAI, M. 2017. ROS-induced ROS release orchestrated by Nox4, Nox2, and mitochondria in VEGF signaling and angiogenesis. *Am J Physiol Cell Physiol*, 312, C749-C764.
- KOLB, H. 1987. Mouse models of insulin dependent diabetes: low-dose streptozocin-induced diabetes and nonobese diabetic (NOD) mice. *Diabetes Metab Rev*, 3, 751-78.
- KOVACINA, K. S., YONEZAWA, K., BRAUTIGAN, D. L., TONKS, N. K., RAPP, U. R. & ROTH, R. A. 1990. Insulin activates the kinase activity of the Raf-1 proto-oncogene by increasing its serine phosphorylation. *J Biol Chem*, 265, 12115-8.
- LAVIN, M. F., DELIA, D. & CHESSA, L. 2006. ATM and the DNA damage response. Workshop on ataxia-telangiectasia and related syndromes. *EMBO Rep*, 7, 154-60.
- LAZAR, D. F., WIESE, R. J., BRADY, M. J., MASTICK, C. C., WATERS, S. B., YAMAUCHI, K., PESSIN, J. E., CUATRECASAS, P. & SALTIEL, A. R. 1995. Mitogen-activated protein kinase kinase inhibition does not block the stimulation of glucose utilization by insulin. *J Biol Chem*, 270, 20801-7.
- LI, J., CHEN, J., RICUPERO, C. L., HART, R. P., SCHWARTZ, M. S., KUSNECOV, A. & HERRUP, K. 2012. Nuclear accumulation of HDAC4 in ATM deficiency promotes neurodegeneration in ataxia telangiectasia. *Nat Med*, 18, 783-90.
- LI, L., PENG, G., LIU, X., ZHANG, Y., HAN, H. & LIU, Z. R. 2020. Pyruvate Kinase M2 Coordinates Metabolism Switch between Glycolysis and Glutaminolysis in Cancer Cells. *iScience*, 23, 101684.
- LIBERTI, M. V. & LOCASALE, J. W. 2016. The Warburg Effect: How Does it Benefit Cancer Cells? *Trends Biochem Sci*, 41, 211-218.

- LUO, W., HU, H., CHANG, R., ZHONG, J., KNABEL, M., O'MEALLY, R., COLE, R. N., PANDEY, A. & SEMENZA, G. L. 2011. Pyruvate kinase M2 is a PHD3-stimulated coactivator for hypoxia-inducible factor 1. *Cell*, 145, 732-44.
- LV, L., XU, Y. P., ZHAO, D., LI, F. L., WANG, W., SASAKI, N., JIANG, Y., ZHOU, X., LI, T. T., GUAN, K. L., LEI, Q. Y. & XIONG, Y. 2013. Mitogenic and oncogenic stimulation of K433 acetylation promotes PKM2 protein kinase activity and nuclear localization. *Mol Cell*, 52, 340-52.
- MAHADEV, K., MOTOSHIMA, H., WU, X., RUDDY, J. M., ARNOLD, R. S., CHENG, G., LAMBETH, J. D. & GOLDSTEIN, B. J. 2004. The NAD(P)H oxidase homolog Nox4 modulates insulin-stimulated generation of H₂O₂ and plays an integral role in insulin signal transduction. *Mol Cell Biol*, 24, 1844-54.
- MCGARRY, J. D. 2002. Banting lecture 2001: dysregulation of fatty acid metabolism in the etiology of type 2 diabetes. *Diabetes*, 51, 7-18.
- MERCER, J. R., CHENG, K. K., FIGG, N., GORENNE, I., MAHMOUDI, M., GRIFFIN, J., VIDAL-PUIG, A., LOGAN, A., MURPHY, M. P. & BENNETT, M. 2010. DNA damage links mitochondrial dysfunction to atherosclerosis and the metabolic syndrome. *Circ Res*, 107, 1021-31.
- MOSER, J. S., GALINDO-FRAGA, A., ORTIZ-HERNANDEZ, A. A., GU, W., HUNSBERGER, S., GALAN-HERRERA, J. F., GUERRERO, M. L., RUIZ-PALACIOS, G. M., BEIGEL, J. H. & LA RED, I. L. I. S. G. 2019. Underweight, overweight, and obesity as independent risk factors for hospitalization in adults and children from influenza and other respiratory viruses. *Influenza Other Respir Viruses*, 13, 3-9.
- MUOIO, D. M. 2014. Metabolic inflexibility: when mitochondrial indecision leads to metabolic gridlock. *Cell*, 159, 1253-62.
- NAGASAWA, H., LITTLE, J. B., LIN, Y. F., SO, S., KURIMASA, A., PENG, Y., BROGAN, J. R., CHEN, D. J., BEDFORD, J. S. & CHEN, B. P. 2011. Differential role of DNA-PKcs phosphorylations and kinase activity in radiosensitivity and chromosomal instability. *Radiat Res*, 175, 83-9.
- NATALE, V. A. I., COLE, T. J., ROTHBLUM-OVIATT, C., WRIGHT, J., CRAWFORD, T. O., LEFTON-GREIF, M. A., MCGRATH-MORROW, S. A., SCHLECHTER, H. & LEDERMAN, H. M. 2021. Growth in ataxia telangiectasia. *Orphanet J Rare Dis*, 16, 123.
- PAIK, S. G., MICHELIS, M. A., KIM, Y. T. & SHIN, S. 1982. Induction of insulin-dependent diabetes by streptozotocin. Inhibition by estrogens and potentiation by androgens. *Diabetes*, 31, 724-9.
- PANG, Z., ZHOU, G., EWALD, J., CHANG, L., HACARIZ, O., BASU, N. & XIA, J. 2022. Using MetaboAnalyst 5.0 for LC-HRMS spectra processing, multi-omics integration and covariate adjustment of global metabolomics data. *Nat Protoc*, 17, 1735-1761.
- PAULINO, T. L., RAFAEL, M. N., HIX, S., SHIGUEOKA, D. C., AJZEN, S. A., KOCHI, C., SUANO-SOUZA, F. I., DA SILVA, R., COSTA-CARVALHO, B. T. & SARNI, R. O. S. 2017. Is age a risk factor for liver disease and metabolic alterations in ataxia Telangiectasia patients? *Orphanet J Rare Dis*, 12, 136.
- PIZZAMIGLIO, L., FOCCHI, E., MURRU, L., TAMBORINI, M., PASSAFARO, M., MENNA, E., MATTEOLI, M. & ANTONUCCI, F. 2016. New Role of ATM in Controlling GABAergic Tone During Development. *Cereb Cortex*, 26, 3879-88.
- PRIOR, S. J., RYAN, A. S., STEVENSON, T. G. & GOLDBERG, A. P. 2014. Metabolic inflexibility during submaximal aerobic exercise is associated with glucose intolerance in obese older adults. *Obesity (Silver Spring)*, 22, 451-7.

- RADULESCU, C. I., CERAR, V., HASLEHURST, P., KOPANITSA, M. & BARNES, S. J. 2021. The aging mouse brain: cognition, connectivity and calcium. *Cell Calcium*, 94, 102358.
- RONNEFARTH, M., HANISCH, N., BRANDT, A. U., MAHLER, A., ENDRES, M., PAUL, F. & DOSS, S. 2020. Dysphagia Affecting Quality of Life in Cerebellar Ataxia-a Large Survey. *Cerebellum*, 19, 437-445.
- SARMIENTO-ORTEGA, V. E., MORONI-GONZALEZ, D., DIAZ, A., GARCIA-GONZALEZ, M. A., BRAMBILA, E. & TREVINO, S. 2023. Hepatic Insulin Resistance Model in the Male Wistar Rat Using Exogenous Insulin Glargine Administration. *Metabolites*, 13.
- SAVAGE, P. J., DIPPE, S. E., BENNETT, P. H., GORDEN, P., ROTH, J., RUSHFORTH, N. B. & MILLER, M. 1975a. Hyperinsulinemia and hypoinsulinemia. Insulin responses to oral carbohydrate over a wide spectrum of glucose tolerance. *Diabetes*, 24, 362-8.
- SAVAGE, P. J., GORDEN, P., BENNETT, P. H. & MILLER, M. 1975b. Insulin responses to oral carbohydrate in true prediabetics and matched controls. *Lancet*, 1, 300-2.
- SCHNEIDER, J. G., FINCK, B. N., REN, J., STANDLEY, K. N., TAKAGI, M., MACLEAN, K. H., BERNAL-MIZRACHI, C., MUSLIN, A. J., KASTAN, M. B. & SEMENKOVICH, C. F. 2006. ATM-dependent suppression of stress signaling reduces vascular disease in metabolic syndrome. *Cell Metab*, 4, 377-89.
- SCHORMANN, N., HAYDEN, K. L., LEE, P., BANERJEE, S. & CHATTOPADHYAY, D. 2019. An overview of structure, function, and regulation of pyruvate kinases. *Protein Sci*, 28, 1771-1784.
- SCHWARTZ, L., SUPURAN, C. T. & ALFAROUK, K. O. 2017. The Warburg Effect and the Hallmarks of Cancer. *Anticancer Agents Med Chem*, 17, 164-170.
- SHEN, X., CHEN, J., LI, J., KOFLER, J. & HERRUP, K. 2016. Neurons in Vulnerable Regions of the Alzheimer's Disease Brain Display Reduced ATM Signaling. *eNeuro*, 3.
- SILLITOE, R. V., KUNZLE, H. & HAWKES, R. 2003. Zebrin II compartmentation of the cerebellum in a basal insectivore, the Madagascan hedgehog tenrec *Echinops telfairi*. *J Anat*, 203, 283-96.
- STAGNI, V., CIROTTI, C. & BARILA, D. 2018. Ataxia-Telangiectasia Mutated Kinase in the Control of Oxidative Stress, Mitochondria, and Autophagy in Cancer: A Maestro With a Large Orchestra. *Front Oncol*, 8, 73.
- TIAN, W., ZHANG, W., WANG, Y., JIN, R., WANG, Y., GUO, H., TANG, Y. & YAO, X. 2022. Recent advances of IDH1 mutant inhibitor in cancer therapy. *Front Pharmacol*, 13, 982424.
- TRAVEN, A. & HEIERHORST, J. 2005. SQ/TQ cluster domains: concentrated ATM/ATR kinase phosphorylation site regions in DNA-damage-response proteins. *Bioessays*, 27, 397-407.
- TURER, A. T., MALLOY, C. R., NEWGARD, C. B. & PODGOREANU, M. V. 2010. Energetics and metabolism in the failing heart: important but poorly understood. *Curr Opin Clin Nutr Metab Care*, 13, 458-65.
- TYANOVA, S., TEMU, T. & COX, J. 2016. The MaxQuant computational platform for mass spectrometry-based shotgun proteomics. *Nat Protoc*, 11, 2301-2319.
- UKROPCOVA, B., SEREDA, O., DE JONGE, L., BOGACKA, I., NGUYEN, T., XIE, H., BRAY, G. A. & SMITH, S. R. 2007. Family history of diabetes links impaired substrate switching and reduced mitochondrial content in skeletal muscle. *Diabetes*, 56, 720-7.
- VANDER HEIDEN, M. G., CANTLEY, L. C. & THOMPSON, C. B. 2009. Understanding the Warburg effect: the metabolic requirements of cell proliferation. *Science*, 324, 1029-33.

- VOLKOW, N. D., TOMASI, D., WANG, G. J., STUDENTSOVA, Y., MARGUS, B. & CRAWFORD, T. O. 2014. Brain glucose metabolism in adults with ataxia-telangiectasia and their asymptomatic relatives. *Brain*, 137, 1753-61.
- WARBURG, O. 1956. On the origin of cancer cells. *Science*, 123, 309-14.
- WITNEY, T. H., JAMES, M. L., SHEN, B., CHANG, E., POHLING, C., ARKSEY, N., HOEHNE, A., SHUHENDLER, A., PARK, J. H., BODAPATI, D., WEBER, J., GOWRISHANKAR, G., RAO, J., CHIN, F. T. & GAMBHIR, S. S. 2015. PET imaging of tumor glycolysis downstream of hexokinase through noninvasive measurement of pyruvate kinase M2. *Sci Transl Med*, 7, 310ra169.
- XU, Y., ASHLEY, T., BRAINERD, E. E., BRONSON, R. T., MEYN, M. S. & BALTIMORE, D. 1996. Targeted disruption of ATM leads to growth retardation, chromosomal fragmentation during meiosis, immune defects, and thymic lymphoma. *Genes Dev*, 10, 2411-22.
- YAMAMOTO, H. A. 1992. Nitroprusside intoxication: protection of alpha-ketoglutarate and thiosulphate. *Food Chem Toxicol*, 30, 887-90.
- YANG, J. S., CHEN, P. P., LIN, M. T., QIAN, M. Z., LIN, H. X., CHEN, X. P., SHANG, X. J., WANG, D. N., CHEN, Y. C., JIANG, B., CHEN, Y. J., WANG, N., CHEN, W. J. & GAN, S. R. 2018. Association Between Body Mass Index and Disease Severity in Chinese Spinocerebellar Ataxia Type 3 Patients. *Cerebellum*, 17, 494-498.
- YANG, W. & LU, Z. 2013. Nuclear PKM2 regulates the Warburg effect. *Cell Cycle*, 12, 3154-8.
- YANG, X., MEI, S., GU, H., GUO, H., ZHA, L., CAI, J., LI, X., LIU, Z. & CAO, W. 2014. Exposure to excess insulin (glargine) induces type 2 diabetes mellitus in mice fed on a chow diet. *J Endocrinol*, 221, 469-80.
- YEO, A. J., HENNINGHAM, A., FANTINO, E., GALBRAITH, S., KRAUSE, L., WAINWRIGHT, C. E., SLY, P. D. & LAVIN, M. F. 2019. Increased susceptibility of airway epithelial cells from ataxia-telangiectasia to *S. pneumoniae* infection due to oxidative damage and impaired innate immunity. *Sci Rep*, 9, 2627.
- YI, Z., WU, Y., ZHANG, W., WANG, T., GONG, J., CHENG, Y. & MIAO, C. 2020. Activator-Mediated Pyruvate Kinase M2 Activation Contributes to Endotoxin Tolerance by Promoting Mitochondrial Biogenesis. *Front Immunol*, 11, 595316.
- YUAN, Y., ZHU, C., WANG, Y., SUN, J., FENG, J., MA, Z., LI, P., PENG, W., YIN, C., XU, G., XU, P., JIANG, Y., JIANG, Q. & SHU, G. 2022. alpha-Ketoglutaric acid ameliorates hyperglycemia in diabetes by inhibiting hepatic gluconeogenesis via serpin1e signaling. *Sci Adv*, 8, eabn2879.
- YUGI, K., KUBOTA, H., TOYOSHIMA, Y., NOGUCHI, R., KAWATA, K., KOMORI, Y., UDA, S., KUNIDA, K., TOMIZAWA, Y., FUNATO, Y., MIKI, H., MATSUMOTO, M., NAKAYAMA, K. I., KASHIKURA, K., ENDO, K., IKEDA, K., SOGA, T. & KURODA, S. 2014. Reconstruction of insulin signal flow from phosphoproteome and metabolome data. *Cell Rep*, 8, 1171-83.
- ZELZER, E., LEVY, Y., KAHANA, C., SHILO, B. Z., RUBINSTEIN, M. & COHEN, B. 1998. Insulin induces transcription of target genes through the hypoxia-inducible factor HIF-1alpha/ARNT. *EMBO J*, 17, 5085-94.
- ZHANG, G., DEINHARDT, K. & NEUBERT, T. A. 2014. Stable isotope labeling by amino acids in cultured primary neurons. *Methods Mol Biol*, 1188, 57-64.
- ZHANG, H., STEELE, J. R., KAHROOD, H. V., LUCAS, D. D., SHAH, A. D. & SCHITTENHELM, R. B. 2023. Phospho-Analyst: An Interactive, Easy-to-Use Web Platform To Analyze Quantitative Phosphoproteomics Data. *J Proteome Res*, 22, 2890-2899.

POINT-TO-POINT RESPONSE TO REVIEWERS' COMMENT

Reviewer #1 (Remarks to the Author):

Comment 1—The authors have successfully and adequately addressed all comments raised by this reviewer and, therefore, the manuscript is now much improved.

Response 1—We are grateful to learn that reviewer #1 is satisfied with our revision and justification.

Reviewer #2 (Remarks to the Author):

Comment 1—All comments have been addressed satisfactorily. Relevant figures and information with regards to metabolomics/lipidomics and tracing experiments have been corrected and are clear. I have no further questions.

Response 1—We are grateful to learn that reviewer #2 is satisfied with our revision and justification.

Reviewer #3 (Remarks to the Author):

Comment 1—The revised form of the manuscripts lacks a clear conceptual framework or logical structure (“red thread”) that guides the reader through the rationale, data, and conclusions. As it stands, the manuscript reads as fragmented and somewhat confusing, even for a specialized audience. For a broader scientific readership, it would be impossible to follow. The use of domain-specific jargon without explanation starting already in the abstract.

Response 1—We agreed with the reviewer that we need to make substantial improvements in the manuscript writing. Below is our response and actions taken:

Conceptual framework: We have clarified the scope and focus of the study by restructuring the introductory section. The revised manuscript shall now present the key ideas and their relationships more clearly and coherently. Additionally, we have included the central scientific question addressed in this study, i.e.,

1. Whether metabolic disruptions originating in the periphery can have lasting effects on long-term brain function through alterations in the profiles of endocrine signals derived from the peripheral system.
2. The precise mechanism by which insulin activates ATM and whether ATM plays a role in coordinating flexible metabolic transitions between catabolic and anabolic states in insulin-sensitive cells.
3. How such metabolic disturbances observed account for the ataxic symptoms in cerebellar degeneration of A-T

We have also included a concise summary of the findings that provide answers to these questions.

Logical structure: Drawing from the outlined conceptual framework above, we enhanced the coherence and logical flow of our results by introducing or refining their respective subheadings. The structure is as follows:

Result section order:

1. ATM deficiency leads to systemic insulin resistance and metabolic inflexibility
2. Cerebellar Purkinje cells are insulin sensitive and functionally associated with peripheral insulin resistance
3. Insulin-activated ATM modulates key regulators of aerobic glycolysis in the cerebellum
4. Defective insulin-ATM signaling results in glycolytic insufficiency and a shift towards glutamine dependence in the cerebellum
5. Zebrin-II/ALDOC-positive Purkinje cells, reliant on glycolysis, are more susceptible to metabolic reprogramming in the absence of ATM
6. Nuclear-localized PKM2 co-activates aerobic glycolysis regulator HIF1 α to reshape a metabolic landscape that favors glucose anabolic fate towards lipid biogenesis
7. Supplementation of α -ketoglutarate (α -KG), the α -keto acid of glutamine, mitigates metabolic challenges associated with ATM deficiency

Our narrative adheres to the "inverted pyramid" structure, commencing with systemic pathological and metabolic observations (Result section #1; Fig.1) before delving into localized effects in the cerebellum (Fig.2-4). We then explore how ATM activity is regulated by systemic signals like insulin (Result section #2; Fig.2), its role in mediating insulin effects under normal conditions (Result section #3; Fig.2), and the repercussions of ATM loss on local cellular integrity and physiology (Result section #4-5; Fig.3-4). Additionally, opposite outcomes of sustained insulin-ATM activity are presented as a way to reversely validate the findings in ATM-deficiency (Result section #6; Fig.5). Ultimately, based on these novel insights, we propose a tailored strategy to mitigate the disrupted metabolic phenotype stemming from ATM deficiency (Result section #7; Fig.6).

Conclusion— In the discussion and conclusion section, we have introduced significant modifications to enhance coherence and connectivity within each paragraph. Initially, we provided an overview of the study background and findings, followed by an elaborate examination of ATM's distinct regulatory impact on PKM2 and HIF1 α , shedding light on how this translates into both physiological and pathological effects of insulin. Subsequently, we delved into a detailed discussion on the selective susceptibility of Zebrin-II/ALDOC-positive neurons, which results in specific anatomical degeneration within the cerebellum leading to the motor deficit phenotype frequently observed in A-T. Additionally, with reference to the systemic-central nervous system crosstalk established by the insulin-ATM signaling, we elucidated the rationale behind α -KG

supplementation with relevance to the metabolic reprogramming changes observed in A-T. Finally, we summarized all our findings, with also a reflection on the study's limitations.

Abstract: We have revised the abstract to focus solely on the key conceptual ideas and findings, along with the concluding remark. All technical jargon has been removed or simplified, with complex terms explained in a way that is more accessible to a general audience. The rewritten abstract is now designed to be more understandable for readers without specialized knowledge in the subject.

Comment 2—I am particularly concerned about the quality and rigor of the statistical analyses, especially concerning the phosphoproteomics data. The manuscript does not clearly state how p-values were corrected for multiple hypothesis testing. From what is presented, it appears that uncorrected p-values were used, which is not acceptable given the large number of comparisons inherent in phosphoproteomic analyses.

Response 2— Following the reviewer's feedback, we conducted a comprehensive review of our data and manuscript text. This review confirmed that all outcomes, including the quantities and identities of differentially expressed phosphorylated and total proteins in each pairwise analysis, were determined using unpaired t-tests adjusted by the Benjamini-Hochberg Procedures to manage multiple comparisons (**Revised manuscript: Supplementary Tables 1-6**). We utilized a threshold of an adjusted p-value of 0.05 (using the Benjamini-Hochberg method) along with a $|\log_2$ fold change| of 1 to identify significantly altered phosphosites (or total proteins) in each pairwise comparison.

We admitted that we mistakenly used p-values on the y-axis when plotting the volcano plots, leading to a misinterpretation by the reviewer. Consequently, in this revised version, we have rectified this mistake by plotting $\log_{10}(\text{adjusted p-value})$ on the y-axis in the volcano plots (**Revised manuscript: Fig.2h, S3d-f, S4e-f**). Furthermore, we have included a description in their figure legends elucidating how differentially expressed hits were identified - specifically, by employing an adjusted p-value of 0.05 using the Benjamini-Hochberg method in conjunction with a $|\log_2$ fold change| of 1.

Comment 3—Moreover, the manuscript lacks a thorough and appropriate pathway analysis. Without this, it is unclear whether any of the reported changes are biologically significant or relevant.

Response 3— In response to the reviewer's feedback, we have made significant updates to our manuscript. Initially, we have revised the related pathway analysis plots to showcase the significantly enriched KEGG pathways based on adjusted p-values instead of p-values (**Revised manuscript: Fig.2j, Supplementary Fig.3d-f, 3h, 4f**).

Subsequently, we have included a more in-depth analysis focusing on the differentially downregulated phosphopeptides enriched from the comparison between the “KO + Glargine” and “WT + Glargine” groups (**Revised manuscript: Fig.2h-n**). Our preliminary investigation using the STRING database for cellular component analysis has revealed that the majority of phosphopeptides are uniquely located in various compartments of neuronal structures (**Revised manuscript: Fig.2i**). This finding not only supports the notion that phosphorylation events primarily occur in neurons within the brain but also suggests a preference for subcellular enrichment in "neuronal cytoplasm" and "axonal cytoplasm," indicating that the cytoplasm is a key site for phosphorylation events. This observation aligns with our finding regarding the oxidative stress mediated dimerization activation mechanism of ATM by insulin, which previous literature also indicated this predominantly takes place in the cytoplasm¹ (**Revised manuscript: Fig.2e-f**).

Following this, our KEGG pathway enrichment analysis has identified two distinct categories of pathways: one associated with aerobic glycolysis ("HIF1 signaling pathway, Glycolysis/Gluconeogenesis," and "Metabolism of cancer") and another linked to RNA splicing ("Spliceosome" and "RNA degradation") (**Revised manuscript: Fig.2j, Supplementary Table 5**). Despite the apparent lack of overlap in biological functions between these two sets of pathways, a more in-depth examination of the protein-protein association network using STRING has revealed interconnectedness among proteins enriched in these pathways (**Revised manuscript: Fig.2k**). Of particular note is the discovery of the protein HNRNPA1 (Heterogeneous nuclear ribonucleoprotein A1), a key RNA-binding protein that regulates the alternative splicing of PKM pre-mRNA, thereby favoring the formation of PKM2 mRNA over that of PKM1^{2,3}. This suggests that the enrichment of phosphoproteins related to RNA splicing may further regulate the effectiveness and capacity of aerobic glycolysis in WT neurons exposed to glargine (i.e., insulin).

Furthermore, we have conducted kinase enrichment analysis⁴ and compared the results between the “KO + Glargine” and “WT + Glargine” groups (**Revised manuscript: Fig.2l, Supplementary Table 5**). In the “WT + Glargine” group, ATM emerged the top kinases predicted, exhibiting the lowest adjusted p-values and displaying interrelation with a group of kinases associated with insulin signaling (CSNK2A1⁵, PRKCB⁶, CSNK1D⁷, PRKDC⁸, CSNK1E⁹) (**Revised manuscript: Fig.2l, Supplementary Table 5**). This pattern was not observed with phosphopeptides enriched from the “KO + Glargine” group (**Revised manuscript: Supplementary Fig.3h**). Moreover, the total number of kinases enriched in the “WT + Glargine” group exceeded that in the “KO + Glargine” group, with the majority (18/23) of uniquely identified kinases in the “WT + Glargine” group playing roles in insulin signaling (CSNK2A1⁵, PRKCB⁶, MAPK14¹⁰, MAPK12¹¹, MAPK10¹², RPS6KA3¹³, MAPK9¹⁴, MAPK8¹⁵, MAPK7¹⁶, MKNK1¹⁷, MAPK1¹⁸, SGK1¹⁹, PRKACB²⁰). Consequently, these findings lay the groundwork for our subsequent focus on insulin-sensitive neurons within cerebellar tissue (**Revised manuscript: Fig.2i, Supplementary**

Fig.2b), further substantiating the role of ATM in facilitating the downstream effects of insulin on cellular metabolism and aerobic glycolysis.

Overall comment—Although the authors attempted to address my points of criticism in their rebuttal letter, I find that many of the responses were not convincing. Several core issues, such as the lack of proper statistical handling, weak data interpretation, and unclear manuscript structure, remain unresolved. In summary, I believe the manuscript in its current form is not suitable for publication and requires substantial revisions to address the critical issues outlined above.

Overall response— We sincerely hope that our comprehensive responses to the comments provided in points 1-3 have addressed the reviewer's concerns and that the latest revisions meet his/her expectations.

Reviewer #4 (Remarks to the Author):

Overall comment

1. The results are important showing that A-T is a metabolic disease and supplementation of alpha-keto-glutarate can alleviate the detrimental symptoms of A-T.
2. Yes. the work provides an important insight into the molecular mechanisms of A-T.
3. Yes. the authors present sufficient data that support their conclusions.
4. I did not find any flaws in data analyses, interpretation and the conclusions.
5. Yes. the methodologies are sound and meet the expected standards in the field of A-T.
6. The authors provide sufficient information regarding the methods for work to be reproduced.

Overall Response —We are grateful to learn that reviewer #4 is satisfied with our revision and justification.

References

1. Paull, T.T. Mechanisms of ATM Activation. *Annu Rev Biochem* **84**, 711-738 (2015).
2. Li, Y. *et al.* The Regulatory Network of hnRNPs Underlying Regulating PKM Alternative Splicing in Tumor Progression. *Biomolecules* **14** (2024).
3. Chen, M., Zhang, J. & Manley, J.L. Turning on a fuel switch of cancer: hnRNP proteins regulate alternative splicing of pyruvate kinase mRNA. *Cancer Res* **70**, 8977-8980 (2010).
4. Lachmann, A. & Ma'ayan, A. KEA: kinase enrichment analysis. *Bioinformatics* **25**, 684-686 (2009).
5. Ampofo, E., Nalbach, L., Menger, M.D., Montenarh, M. & Gotz, C. Protein Kinase CK2-A Putative Target for the Therapy of Diabetes Mellitus? *Int J Mol Sci* **20** (2019).
6. Gassaway, B.M. *et al.* PKCepsilon contributes to lipid-induced insulin resistance through cross talk with p70S6K and through previously unknown regulators of insulin signaling. *Proc Natl Acad Sci U S A* **115**, E8996-E9005 (2018).

7. Xu, P. *et al.* Gene expression levels of Casein kinase 1 (CK1) isoforms are correlated to adiponectin levels in adipose tissue of morbid obese patients and site-specific phosphorylation mediated by CK1 influences multimerization of adiponectin. *Mol Cell Endocrinol* **406**, 87-101 (2015).
8. Wong, R.H. *et al.* A role of DNA-PK for the metabolic gene regulation in response to insulin. *Cell* **136**, 1056-1072 (2009).
9. Modak, C. & Bryant, P. Casein Kinase I epsilon positively regulates the Akt pathway in breast cancer cell lines. *Biochem Biophys Res Commun* **368**, 801-807 (2008).
10. Meng, D. *et al.* p38alpha Deficiency in T Cells Ameliorates Diet-Induced Obesity, Insulin Resistance, and Adipose Tissue Senescence. *Diabetes* **71**, 1205-1217 (2022).
11. Sweeney, G. *et al.* An inhibitor of p38 mitogen-activated protein kinase prevents insulin-stimulated glucose transport but not glucose transporter translocation in 3T3-L1 adipocytes and L6 myotubes. *J Biol Chem* **274**, 10071-10078 (1999).
12. Abdelli, S. & Bonny, C. JNK3 maintains expression of the insulin receptor substrate 2 (IRS2) in insulin-secreting cells: functional consequences for insulin signaling. *PLoS One* **7**, e35997 (2012).
13. Brial, F. *et al.* Stimulation of insulin secretion induced by low 4-cresol dose involves the RPS6KA3 signalling pathway. *PLoS One* **19**, e0310370 (2024).
14. Singh, R. *et al.* Differential effects of JNK1 and JNK2 inhibition on murine steatohepatitis and insulin resistance. *Hepatology* **49**, 87-96 (2009).
15. Solinas, G. *et al.* JNK1 in hematopoietically derived cells contributes to diet-induced inflammation and insulin resistance without affecting obesity. *Cell Metab* **6**, 386-397 (2007).
16. Sharma, G. & Goalstone, M.L. Regulation of ERK5 by insulin and angiotensin-II in vascular smooth muscle cells. *Biochem Biophys Res Commun* **354**, 1078-1083 (2007).
17. Moore, C.E. *et al.* MNK1 and MNK2 mediate adverse effects of high-fat feeding in distinct ways. *Sci Rep* **6**, 23476 (2016).
18. Khoo, S. *et al.* Regulation of insulin gene transcription by ERK1 and ERK2 in pancreatic beta cells. *J Biol Chem* **278**, 32969-32977 (2003).
19. Yang, C. *et al.* The functional duality of SGK1 in the regulation of hyperglycemia. *Endocr Connect* **9**, R187-R194 (2020).
20. Enns, L.C. *et al.* Attenuation of age-related metabolic dysfunction in mice with a targeted disruption of the Cbeta subunit of protein kinase A. *J Gerontol A Biol Sci Med Sci* **64**, 1221-1231 (2009).